# Linked and fully-coupled 3D earthquake dynamic rupture and tsunami modeling for the Húsavík-Flatey Fault Zone in North Iceland

Fabian Kutschera[1,2], Alice-Agnes Gabriel[2,1], Sara Aniko Wirp[3,1], Bo Li[4,1], Thomas Ulrich[1], Claudia Abril[5], and Benedikt Halldórsson[6,7]

[1]Institute of Geophysics, Department of Earth and Environmental Sciences, Ludwig-Maximilians-University, Munich, Germany
[2]Institute of Geophysics and Planetary Physics, Scripps Institution of Oceanography, UC San Diego, California, USA
[3]Division of Physical Sciences and Engineering, King Abdullah University of Science and Technology, Thuwal, Saudi Arabia
[4]GEOMAR, Helmholtz Centre for Ocean Research, Kiel, Germany
[5]Barcelona Supercomputing Center, Barcelona, Spain
[6]Division of Processing and Research, Icelandic Meteorological Office, Reykjavík, Iceland
[7]Faculty of Civil and Environmental Engineering, School of Engineering and Natural Sciences, University of Iceland, Reykjavík, Iceland

**Correspondence:** Fabian Kutschera (fkutschera@ucsd.edu)

**Abstract.**

Tsunamigenic earthquakes pose considerable risks, both economically and socially, yet earthquake and tsunami hazard assessments are typically conducted separately. Earthquakes associated with unexpected tsunamis, such as the 2018 $M_W$ 7.5 strike-slip Sulawesi earthquake, emphasize the need to study the tsunami potential of active submarine faults in different tectonic settings. Here, we investigate physics-based scenarios combining simulations of 3D earthquake dynamic rupture and seismic wave propagation with tsunami generation and propagation. We present time-dependent modeling of one-way linked and 3D fully-coupled earthquakes and tsunamis for the ∼100 km long Húsavík-Flatey Fault Zone in North Iceland. Our analysis shows that the HFFZ has the potential to generate sizeable tsunamis. The six dynamic rupture models sourcing our tsunami scenarios vary regarding hypocenter location, spatio-temporal evolution, fault slip, and fault structure complexity but coincide with historical earthquake magnitudes. Earthquake dynamic rupture scenarios on a less segmented fault system, particularly with a hypocenter location in the eastern part of the fault system, have a larger potential for local tsunami generation. Here, dynamically evolving large shallow fault slip (∼8 m), near-surface rake rotation (±20°), and significant coseismic vertical displacements of the local bathymetry (±1 m) facilitate strike-slip faulting tsunami generation. We model tsunami crest-to-trough differences (total wave heights) of up to ∼0.9 m near the town Ólafsfjörður. In contrast, none of our scenarios endanger the town of Akureyri, which is shielded by multiple reflections within the narrow Eyjafjörður Bay and by Hrísey Island.

We compare the modeled one-way linked tsunami waveforms with simulation results using a 3D fully-coupled approach. We find good agreement in the tsunami arrival times and location of maximum tsunami heights. While seismic waves result in transient motions of the sea surface and affect the ocean response, they do not appear to contribute to tsunami generation. However, complex source effects arise in the fully-coupled simulations, such as tsunami dispersion effects and complex super-

position of seismic and acoustic waves within the shallow continental shelf of North Iceland. We find that the vertical velocity amplitudes of near-source acoustic waves are unexpectedly high – larger than those corresponding to the actual tsunami – which may serve as a rapid indicator of surface dynamic rupture. Our results have important implications for understanding the tsunamigenic potential of strike-slip fault systems worldwide and the coseismic acoustic wave excitation during tsunami generation and may help to inform future tsunami early warning systems.

## 1  Introduction

Earthquake-generated tsunamis are generally associated with large submarine events on dip-slip faults, in particular at subduction zone megathrust interfaces (e.g., Bilek and Lay, 2018; Lotto et al., 2018; Melgar and Ruiz-Angulo, 2018; Wirp et al., 2021). The potential generation of a tsunami depends not only on the magnitude of the earthquake, but on the rupture process (e.g., Kanamori, 1972; Ulrich et al., 2022), the geomorphology of the region (e.g., Mori et al., 2022) and secondary effects such as landsliding or mass slumping (Harbitz et al., 2006; Løvholt et al., 2015; Moretti et al., 2020; Poulain et al., 2022). The typically underrepresented tsunami hazard posed by large (partially) submarine strike-slip fault systems has received increasing attention since the unexpected and devastating local tsunami in Palu Bay following the 2018 $M_W$ 7.5 strike-slip Sulawesi earthquake in Indonesia (Ulrich et al., 2019b; Bao et al., 2019; Socquet et al., 2019; Elbanna et al., 2021; Amlani et al., 2022; Ma, 2022). Assessing the tsunamigenic potential of strike-slip fault systems has important implications worldwide, such as for the Dead Sea Transform fault system, the Enriquillo–Plantain Garden fault zone in Haiti, and for northern offshore sections of the San Andreas fault system in California.

Here, we focus on the ∼100 km long Húsavík-Flatey Fault Zone (HFFZ, Fig. 1), the largest strike-slip fault in Iceland, which is part of the Tjörnes Fracture Zone (TFZ). The TFZ is a complex transcurrent fault system composed of three main lineaments. It links the Kolbeinsey Ridge (KR) as part of the Mid-Atlantic Ridge offshore North of Iceland (Eyjafjarðaráll Rift Zone) to its manifestation on land in the Northern Volcanic Zone (NVZ), which is characterized by volcanic systems and extensional faulting (Sæmundsson, 1974; Einarsson, 1991; Geirsson et al., 2006; Einarsson, 2008; Stefansson et al., 2008; Einarsson and Brandsdóttir, 2021). Earthquake faulting in the TFZ is driven by eastward spreading of the Eurasian plate with an average velocity of ∼18 mm/yr relative to the North American plate (Stefansson et al., 2008; Demets et al., 2010). The HFFZ strikes from offshore to onshore and is characterized by right-lateral (dextral) strike-slip faulting, a faulting mechanism which appears frequently subparallel to the adjacent active rift zones of Iceland (Karson et al., 2018). It poses the largest threat to coastline communities such as the town of Húsavík, which is located atop the Húsavík-Flatey Fault Zone at the eastern side of Skjálfandi Bay.

North Iceland has experienced several large earthquakes in the past. Two magnitude 6.5 earthquakes occurred in 1872 and a recent $M_W$ 6 earthquake struck the western end of the HFFZ in 2020 (Fig. 1). The largest $M$7 event in 1755 caused extensive damage and historic reports indicate that a tsunami hit the coastline and overturned boats (Stefansson et al., 2008; Þorgeirsson, 2011; Ruiz-Angulo et al., 2019). Likewise, such reports include records that the events in 1872 caused rapid sea level changes resulting in a series of waves, i.e., a tsunami-like behavior. High-resolution seismic reflection data within Skjálfandi Bay

reveal up to 15 m of accumulated vertical offset during the last ~12,000 years (Magnúsdóttir et al., 2015; Brandsdóttir et al., 2022), indicating possible vertical deformation of the ocean bottom during past earthquakes. This emphasizes the relevance of studying the Húsavík-Flatey Fault Zone from earthquake rupture to its tsunami potential. Metzger and Jónsson (2014) estimate that 30 % to 50 % of the full transform motion is taken up by the HFFZ, corresponding to a geodetic slip rate of 6 to 9 mm yr$^{-1}$. Thus, a locked HFFZ may host potential $M_W$ 6.8±0.1 earthquakes (Metzger et al., 2011, 2013). Although the long-term Holocene slip rate is presumably slower than the present-day geodetic slip rate, it can be used to derive an average recurrence time of 500 to 600 years for a $M_W$ 7 earthquake on the HFFZ (Matrau et al., 2022). De Pascale (2022) calculate a recurrence interval of 32±24 years for a magnitude 6 event. Recent velocities obtained from Global Navigation Satellite System (GNSS) measurements – using more than 100 continuous and campaign-style GNSS stations in total – are close to zero near the fault, indicating that the HFFZ may be fully locked (Barreto et al., 2022).

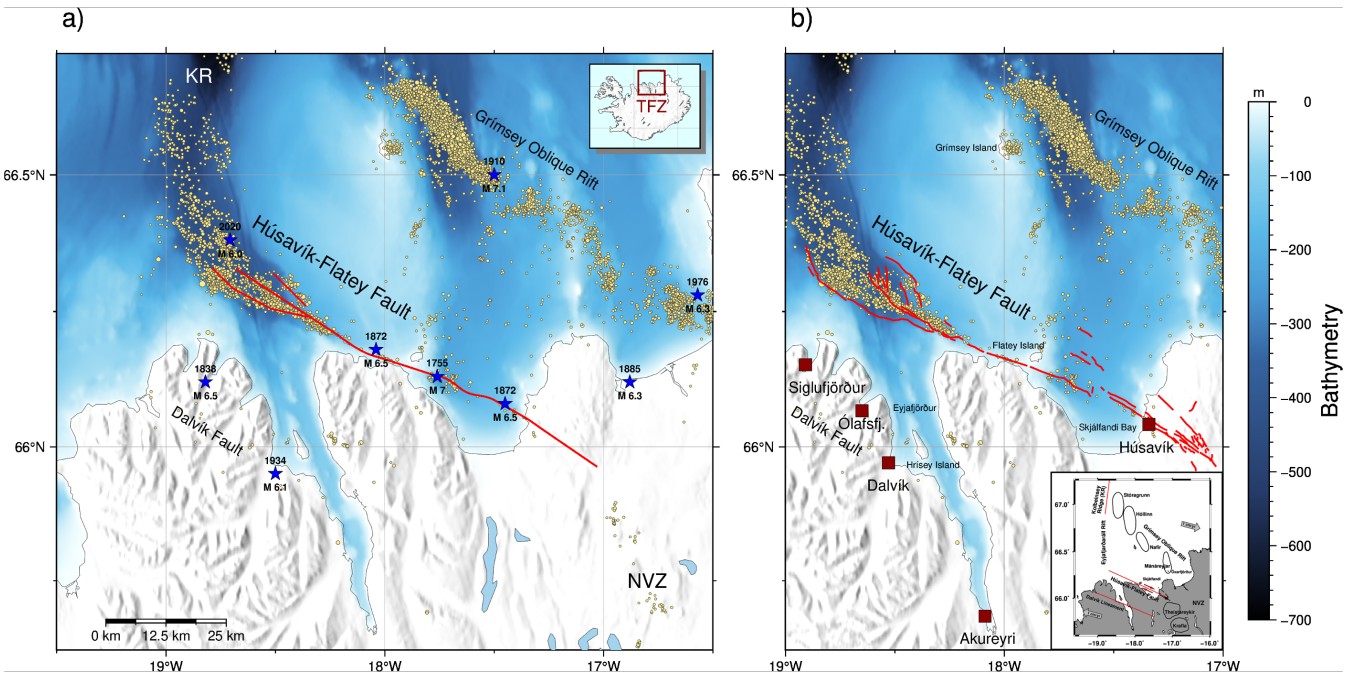

**Figure 1.** Overview of the Tjörnes Fracture Zone (TFZ), which connects the Kolbeinsey Ridge (KR) as part of the Mid-Atlantic Ridge offshore North of Iceland (Eyjafjarðaráll Rift Zone) to its manifestation on land in the Northern Volcanic Zone (NVZ). Yellow circles represent relocated seismicity from 1993 to 2019 (Abril et al., 2018, 2019). a) The here used "simple" fault geometry of the Húsavík-Flatey Fault Zone (HFFZ), which has three segments, is shown as red lines (Li et al., 2023). Historic large earthquakes with $M \geq 6$ are indicated as blue stars (Ambraseys and Sigbjörnsson, 2000; Stefansson et al., 2008; Þorgeirsson, 2011; Jónsson, 2019). b) The used "complex" fault geometry of the HFFZ (Li et al., 2023), which includes 55 fault segments (shown as red lines) together with major towns in the region of Norðurland eystra. The inset at the bottom right shows a schematic tectonic overview of the TFZ with the average plate motion (Stefansson et al., 2008; Demets et al., 2010) and the overall regional tectonic setting in North Iceland, including the location and names of the volcanic centers at the Grímsey Oblique Rift.

A better understanding of the complex interaction between static and time-dependent earthquake displacements, off-fault deformation, and seismic, acoustic, and tsunami amplitudes is now possible using realistic 3D scenarios. Non-linear earthquake dynamic rupture simulations combining coseismic frictional failure on prescribed faults and seismic wave propagation are powerful tools to investigate earthquake dynamics as a consequence of the model's initial conditions (e.g., Aochi and Ulrich, 2015; Wollherr et al., 2019; Ulrich et al., 2019a; Lozos and Harris, 2020; Harris et al., 2021; Taufiqurrahman et al., 2022; Biemiller et al., 2023). Empowered by high-performance computing (Ben-Zion et al., 2022), joint earthquake-tsunami modeling is now becoming applicable for the development of (probabilistic) tsunami forecasting and early warning systems (Yamamoto, 1982; Cecioni et al., 2014; Bernard and Titov, 2015; Mei and Kadri, 2017; Gomez and Kadri, 2021; Selva et al., 2021).

In this study, we investigate the tsunami potential of the HFFZ using two techniques to couple earthquake and tsunami models. First, we apply a one-way linked approach that links the time-dependent seafloor deformation from 3D earthquake dynamic rupture with a subsequent tsunami simulation based on solving the shallow-water equations (Ulrich et al., 2019b; Madden et al., 2020; Wirp et al., 2021; Ulrich et al., 2022; van Zelst et al., 2022). Second, we show 3D fully-coupled earthquake-tsunami models, which simulate seismic (i.e., elastic), ocean gravity (i.e., tsunami), and compressional ocean acoustic waves simultaneously and self-consistently (Lotto and Dunham, 2015; Krenz et al., 2021; Abrahams et al., 2023). We extend six recent dynamic rupture scenarios (Fig. 2) from a suite of physics-based dynamic rupture models (Li et al., 2023). The chosen dynamic rupture models vary in their hypocenter location, spatio-temporal evolution of rupture dynamics, fault slip, and geometric fault system complexity. The simple fault geometry rupture models of Li et al. (2023) coincide with historically and physically plausible earthquake magnitudes, stress drop, rupture speed, and slip distributions, and produce ground motions that have been verified against empirical Ground Motion Models (GMMs) calibrated for Iceland (Kowsari et al., 2020).

We detail the earthquake and tsunami model setups in Sect. 2. Section 3.1 summarizes the six dynamic rupture earthquake scenarios. In Sect. 3.2.1 we investigate physically plausible scenarios of potentially tsunamigenic HFFZ earthquakes by using the one-way linked earthquake-tsunami modeling approach for all six dynamic rupture scenarios. We show that the HFFZ may generate tsunamigenic earthquakes, potentially posing a significant hazard to coastline communities. Based on the results from the one-way linked simulations, we select the three earthquake-tsunami scenarios on the simpler fault geometry causing larger wave heights for the fully-coupled approach to better understand the initial tsunami genesis and complex superposition of seismic, acoustic, and tsunami waves. We compare the results for both earthquake-tsunami modeling techniques in Sect. 3.2.2.

## 2 Model setup

We present one-way linked (cf. Sect. 2.4) and fully-coupled (cf. Sect. 2.5) tsunami models (Abrahams et al., 2023) that are sourced by earthquakes simulated as dynamically propagating shear rupture (Ramos et al., 2022) on seismically locked (Wang and Dixon, 2004) pre-existing faults.

We use six earthquake scenarios based on a suite of 3D spontaneous dynamic rupture simulations developed in Li et al. (2023) that can match local GMMs and reproduce historic earthquake magnitudes. Dynamic rupture modeling includes solving for

the spontaneous frictional failure non-linearly linked to the propagation of seismic waves (Fig. 3) with the purpose of gaining knowledge about the underlying physical processes. Such physically self-consistent descriptions of how faults yield and slide have been developed for complex and/or poorly instrumented earthquakes in various tectonic contexts (e.g., Olsen et al., 1997; Douilly et al., 2015; Kyriakopoulos et al., 2017; Harris et al., 2021; Taufiqurrahman et al., 2023). In contrast to kinematic earthquake source modeling, fault slip is not prescribed, but the rupture dynamics evolve based on an empirical friction law and chosen initial conditions. Here, the initial conditions of the dynamic rupture models, including fault geometries, pre-stress, and fault strength, are constrained by seismic, geodetic, and bathymetry observations as briefly summarized in the following sections. For details and sensitivity analysis of HFFZ dynamic rupture simulations and their initial conditions we refer to Li et al. (2023).

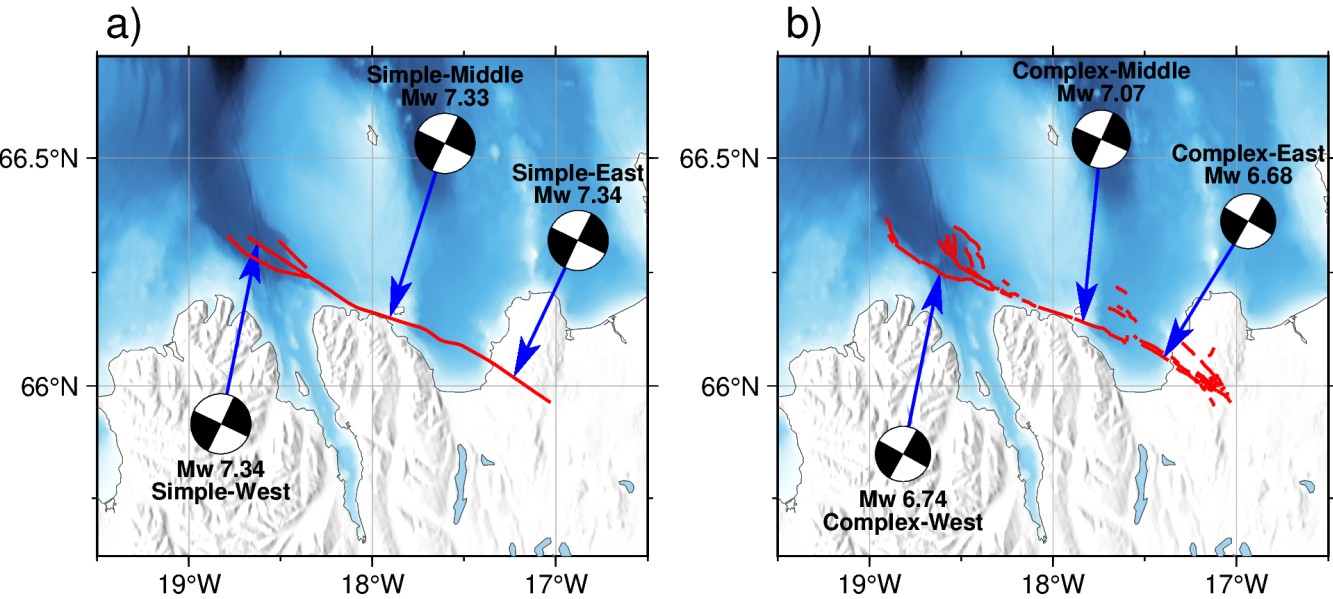

**Figure 2.** Overview over the six 3D dynamic rupture earthquake scenarios based on Li et al. (2023). Arrows indicate the three varied epicenter locations. Each dynamic rupture scenario is nucleated at a hypocentral depth of 7 km. We show the on-fault measured moment magnitude and the equivalent centroid moment tensor solutions (constructed after Ulrich et al. (2022)) representing overall strike-slip faulting mechanisms of the dynamic rupture scenarios. a) shows the three dynamic rupture models on the simple fault system geometry with varying epicentral locations and b) are the three scenarios on the complex fault system geometry.

## 2.1 Fault geometry and subsurface structure

The fault geometry plays an important role in the potential for tsunami generation caused by submarine earthquake rupture. Fault trenching has been conducted for the onshore part of the HFFZ (Harrington et al., 2016; Matrau et al., 2021) and can be used to extrapolate the location of the off-shore fault trace. Recent offshore seismic reflection campaigns in North Iceland and high-resolution bathymetry interpretation (Brandsdóttir et al., 2005; Magnúsdóttir and Brandsdóttir, 2011; Magnúsdóttir

et al., 2015; Hjartardóttir et al., 2016) together with relocated seismicity (Abril et al., 2018, 2019) provide detailed insight on the complexity of the structure of the off-shore fault system. However, it remains challenging to decide which degree of fault system complexity is important for tsunami hazards and to gain direct constraints on the variability of the off-shore geometry of the HFFZ fault system. To capture some of the geometric uncertainty, we consider two proposed fault geometries (Fig. 1, 2) with varying degrees of complexity. The complex fault geometry comprises 55 partially cross-cutting fault segments,

each vertically dipping and intersecting with the complex geomorphology (Li et al., 2023). The simpler fault geometry is composed of one main fault segment with two shorter adjoint fault segments in the West. We assume vertical fault segments that agree with relocated seismicity (Abril et al., 2018, 2019). All faults are embedded in the same recent 3D velocity model (Abril et al., 2021). We subsequently refer to the three earthquake dynamic rupture scenarios on the simpler fault geometry as "Simple-West", "Simple-Middle" and "Simple-East", while the three models on the highly complex fault geometry are called

"Complex-West", "Complex-Middle" and "Complex-East" – the cardinal directions correspond to the epicenter locations with respect to the fault systems as shown in Fig. 2.

## 2.2 Initial stresses and fault friction parametrization

Following Ulrich et al. (2019a), Li et al. (2023) combine Anderson's theory of faulting in combination with Mohr-Coulomb theory of frictional failure (Coulomb, 1776; Anderson, 1905; Célérier, 2008) to define realistic levels of pre-stress for all

125 dynamic rupture simulations. In particular, the intermediate principal stress $\sigma_2$ is assumed to be vertical ($\sigma_1 > \sigma_2 > \sigma_3$). The Icelandic Stress Map from Ziegler et al. (2016) justifies this assumption. Based on the three best quality criteria from the world stress map project (Zoback et al., 1989; Zoback, 1992; Sperner et al., 2003; Heidbach et al., 2007, 2010), they choose the maximum horizontal stress, $SHmax$ (cf. Table 1), to set up a homogeneous regional stress field (Ziegler et al., 2016). This is consistent with previous estimates of $SHmax$ from Angelier et al. (2004) and agrees with the local transtensional deformation

pattern (Garcia and Dhont, 2004).

The stress shape ratio $\nu = (\sigma_2 - \sigma_3)/(\sigma_1 - \sigma_2)$ facilitates the characterization of the stress regime and balances the principal stress amplitudes. Li et al. (2023) select $\nu = 0.5$ corresponding to strike-slip faulting, which is supported by Ziegler et al. (2016) and the analysis of borehole breakouts, earthquake focal mechanism inversions, and geological data. It also agrees with our assumption of a $90°$ dipping fault system.

The dynamic rupture models use a linear slip-weakening (LSW) friction law with frictional cohesion to model frictional yielding and dynamic slip evolution (Ida, 1972; Andrews, 1976). The selected static and dynamic coefficients of friction ($\mu_s$ and $\mu_d$) are consistent with Byerlee's law under the assumption that the increase of rock strength with depth is independent of rock type. The critical slip weakening distance $D_c$ is lower within the nucleation zone for the models with the simpler fault geometry (Table 1).

The relative fault strength is expressed by the maximum pre-stress ratio $R_0$, the ratio of the potential stress drop to the breakdown strength drop (also known as strength excess). $R_0 = (\tau_0 - \mu_d \sigma_n^{'})/((\mu_s - \mu_d)\sigma_n^{'})$, where $\tau_0$ represents the initial shear stress on the fault and $\sigma_n^{'}$ the initial effective normal stress. While in theory $R_0 = 1$ implies critical pre-stress on a virtual optimally orientated plane (Biemiller et al., 2022), $R_0$ falls between 0.9 and 0.55 in our models. In this study, we compare

end-member dynamic rupture scenarios in terms of their generated vertical displacements and, thus, their potential to generate a tsunami. We also require that our comparison includes scenarios with comparable and plausible moment magnitude and dynamic stress drop. Our parameter choices fall within the range of uncertainty and sensitivities of the suite of dynamic rupture scenarios explored in Li et al. (2023). We here choose a slightly higher $R_0 = 0.9$ for all three dynamic rupture simulations on the complex fault geometry in comparison to the scenarios shown by Li et al. (2023) using the complex fault geometry ($R_0 = 0.85$). $R_0$ itself is difficult to directly obtain from observations and we constrain it using a few dynamic rupture trial-and-error simulations (Ulrich et al., 2019a). The change in $R_0$ results in a $\sim$20 % average increase in vertical displacements. Based on the large parameter space explored in the suite of HFFZ dynamic rupture simulations of Li et al. (2023), our chosen models represent end-member earthquake-tsunami scenarios in terms of large uplift. To conserve comparable dynamic stress drops with such increased $R_0$, we prescribe a slightly reduced pore fluid ratio $\gamma = 0.7$ compared to their $\gamma = 0.75$. For the scenarios on the simpler fault geometry, we slightly increase $\mu_s = 0.6$ (cf. $\mu_s = 0.55$ in Li et al. (2023)) which again leads to slightly increased vertical uplifts but still matches local GMMs. All rupture models are initiated smoothly in time and space by gradually reducing the fault strength ($\mu_s$) at a predefined hypocentral location (Harris et al., 2018).

6 to 10 km is the inferred locking depth for the HFFZ (Metzger and Jónsson, 2014), which was estimated by the combined analysis of InSAR time-series and GNSS data and a back-slip model, which describes the interseismic locking by applying continuous slip at depth in reversed slip direction (e.g., Savage, 1983; Metzger et al., 2013; Wang et al., 2015). The locking depth specifies the transition from seismic to aseismic faulting and limits the seismogenic part of a fault system (e.g., Rogers and Nason, 1971). Together with the consideration of the relocated seismicity from Abril et al. (2018, 2019), the nucleation depth of all earthquake dynamic rupture scenarios is chosen to be at 7 km (Li et al., 2023). Our assumed lower limit of the locking depth ($\sim$10 km, see Li et al. (2023)) is shallower in comparison to the locking depths of most continental strike-slip faults (Vernant, 2015). Consequently, this can result in an overshoot of fault length scaling relations (Mai and Beroza, 2000; Shaw, 2013). However, it is in agreement with oceanic transform faults (Abercrombie and Ekström, 2001), where the warmer temperature of the lithosphere at the Mid-Atlantic Ridge controls slip at depth.

## 2.3 Off-fault plastic yielding

All dynamic rupture models incorporate off-fault plasticity (Fig. A1). Accounting for off-fault deformation provides a more realistic representation of rupture dynamics in a fault zone with damaged host rock after the coseismic rupture phase (e.g., Antoine et al., 2022). We use a non-associative Drucker-Prager visco-plastic rheology (Wollherr et al., 2018) requiring assumptions on the bulk cohesion and the bulk friction as governing material parameters. Similarly to the model parameterization in Li et al. (2023), the bulk friction is set to resemble the fault static coefficient of friction ($\mu_s = 0.6$) and assumed to be constant in the elastic solid medium. Bulk cohesion is depth-dependent and varies in dependence of the velocity model. It is calculated as a function of our 3D rigidity model as $C_{plast} = 10^{-4}\mu$ [Pa], which is following the low cohesion model of Roten et al. (2014).

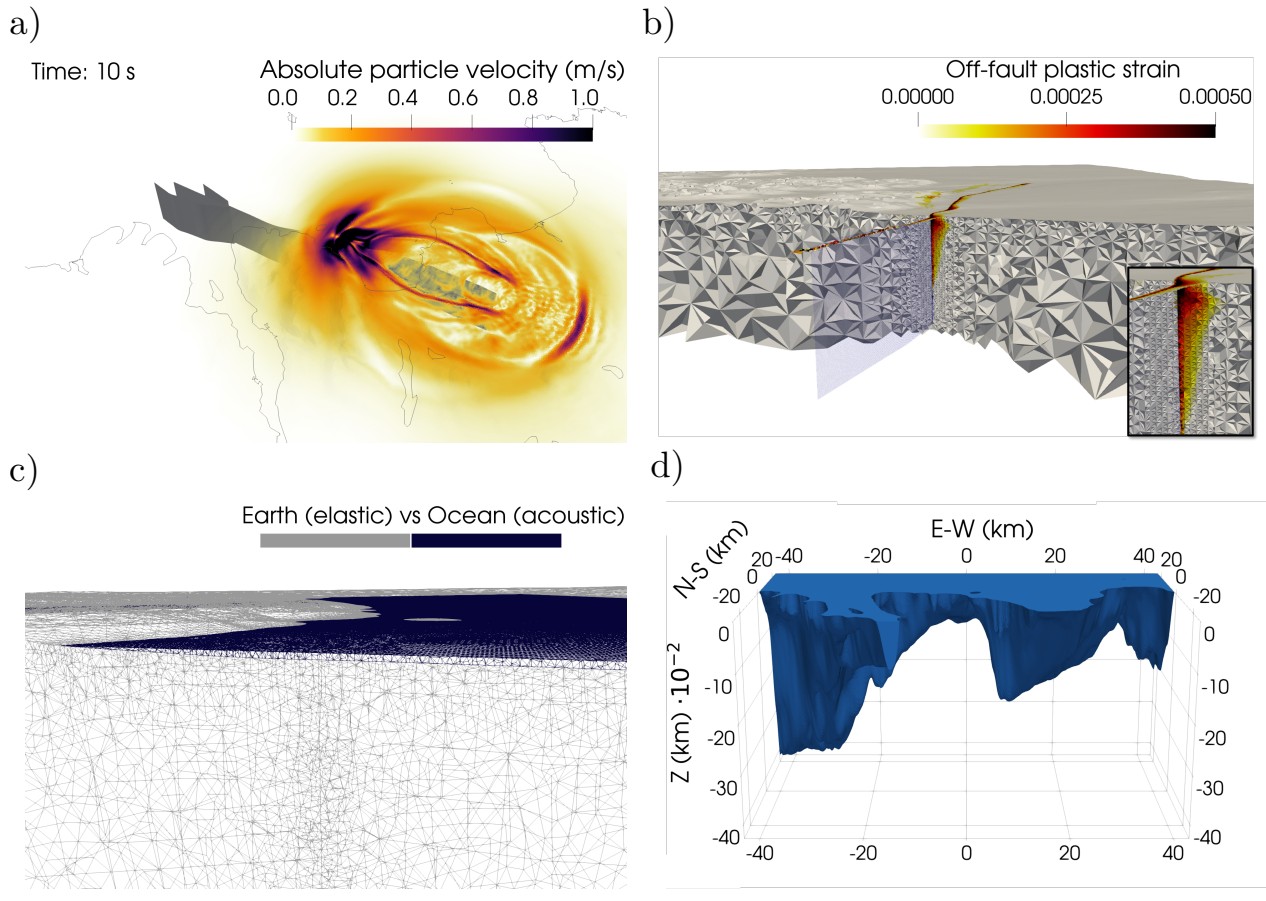

**Figure 3.** a) Snapshot at $t = 10\ s$ of the simulated seismic wavefield for the earthquake dynamic rupture nucleating in the East of the simple fault geometry. b) Accumulated off-fault plastic strain ($\eta$) at the end of simulation Simple-East forming a shallow flower structure. The zoom into the flower structure at the bottom right additionally shows the incorporated static mesh refinement near the fault. c) Mesh of the fully-coupled earthquake-tsunami simulation with the distinction between the elastic medium (Earth) and the acoustic medium (Ocean). d) Vertically exaggerated 3D water layer of the fully-coupled mesh with a maximal length (E-W) of 86 km, maximal width (N-S) of 52 km and maximal depth (Z) of 430 m.

| Parameter | Models with simpler fault geometry | Models with complex fault geometry |
|---|---|---|
| Static friction coefficient ($\mu_\mathrm{s}$) | 0.6 | 0.55 |
| Dynamic friction coefficient ($\mu_\mathrm{d}$) | 0.1 | 0.1 |
| Critical slip distance ($D_\mathrm{c}$) within nucleation area [m] | 0.2 | 0.4 |
| Critical slip distance ($D_\mathrm{c}$) outside nucleation area [m] | 0.5 | 0.4 |
| $SHmax$ [deg] | 155 | 150/155[*] |
| Seismogenic depth [km] | 10 | 10 |
| Nucleation depth [km] | 7 | 7 |
| Maximum pre-stress ratio ($R_0$) | 0.55 | 0.9 |
| Pore fluid ratio ($\gamma$) | 0.6 | 0.7 |
| Stress shape ratio ($\nu$) | 0.5 | 0.5 |
| Nucleation patch radius ($r_{crit}$ [km]) | 1.5 | 1.5 |

**Table 1.** Summary of dynamic rupture parameters chosen by Li et al. (2023) for the models with the simpler and complex fault geometry. [*]The orientation of the maximum horizontal stress is 150° for scenarios Complex-Middle and Complex-East but $SHmax =155°$ for Complex-West.

## 2.4 One-way linked methodology

We use the scientific open-source software package SeisSol (https://github.com/SeisSol/SeisSol, http://www.seissol.org) to simulate six earthquake dynamic rupture scenarios on the HFFZ on two fault system geometries (Sect. 2.1). SeisSol utilizes the arbitrary high-order accurate derivative discontinuous Galerkin method (ADER-DG) (Käser and Dumbser, 2006; Dumbser and Käser, 2006; de la Puente et al., 2009) and has been verified in community benchmarks for dynamic rupture earthquake simulations (Pelties et al., 2014; Harris et al., 2018). SeisSol achieves high-order accuracy in both space and time (Breuer et al., 2015; Uphoff et al., 2017; Krenz et al., 2021) and uses unstructured tetrahedral meshes to incorporate complex 3D bathymetry and topography and the complex fault geometries.

The one-way linked workflow uses the time-dependent seafloor displacement output from SeisSol to initialize sea surface perturbations within sam(oa)$^2$-flash (https://gitlab.lrz.de/samoa/samoa), a dynamically adaptive software for parallel computing (Meister et al., 2016). It solves the non-linear hydrostatic shallow water equations and has been linked to SeisSol in previous work (Ulrich et al., 2019b; Madden et al., 2020; Wirp et al., 2021). We apply the "Tanioka" filter (Tanioka and Satake, 1996), which takes the contribution of the horizontal ground deformation of the realistic bathymetry to the vertical displacement into account. However, the influence of the filter is negligible, likely due to the relatively flat seafloor surrounding the fault system without any large bathymetric gradients. The earthquake dynamic rupture scenarios are simulated for 100 s to ensure that seismic waves have reached the absorbing boundary conditions at the domain edges (Ramos et al., 2022). Each subsequent tsunami is simulated for 40 min which provides sufficient time for the tsunami to reach the coastline.

## 2.5 3D fully-coupled modeling

Traditional earthquake-tsunami modeling is often based on two-step approaches (Abrahams et al., 2023) such as the one-way linked methodology introduced in Sect. 2.4. The fully-coupled method combines earthquake dynamic rupture and tsunami generation into one simulation aiming to capture the full physics of this process (Lotto and Dunham, 2015; Lotto et al., 2018; Wilson and Ma, 2021; Ma, 2022). 3D fully-coupled earthquake-tsunami modeling has recently been implemented in SeisSol, which allows us to account for the generation, propagation, and interaction of 3D elastic, acoustic, and tsunami waves, including dispersion effects, simultaneously (Krenz et al., 2021). The unstructured tetrahedral mesh is extended to include an additional water layer, which is necessary to include both an elastic (Earth) and an acoustic medium (Ocean) (Fig. 3). We incorporate the same resolution bathymetry in the fully-coupled model as used for the one-way linked workflow. The geometric union of fault geometry, subsurface, and the ocean is non-trivial. The higher computational cost associated with adding oceanic acoustic and tsunami wave simulation requires a reduction of the modeling domain, which we achieve by prescribing a water layer that is laterally smaller than the Earth modeling domain (Fig. 3). Within the water layer, we set the rigidity equal to zero ($\mu = 0$) and we prescribe an ocean acoustic wave speed of $\sim$1500 m s$^{-1}$. Acoustic waves (compressive sound waves) are modeled everywhere within the water layer while tsunamis waves, treated as surface gravity waves, are modeled driven by gravity forces acting as restoring forces trying to restore equilibrium at the sea surface (Krenz et al., 2021). The simulated time is 3 min, which allows us to compare the initial tsunami generation and capture the complex superposition of seismic, acoustic, and tsunami waves. Our fully-coupled simulation time is chosen accordingly to avoid waves reaching beyond the edges of the water layer model extent. The spatial discretization within the water layer is 200 m. We use a polynomial order of $p = 4$ (i.e., fifth-order of accuracy in time and space). The on-fault resolution of 200 m is gradually coarsened away from the HFFZ to a maximum size of 5 km at the edges of the elastic medium.

Based on the six scenarios using the one-way linked approach, we analyze the three plausible "worst-case" tsunamigenic scenarios on the simpler fault geometry with the fully-coupled approach, Simple-West, Simple-Middle, and Simple-East. All initial conditions of the dynamic rupture models are kept the same as in the respective linked scenarios.

## 3 Results

### 3.1 Dynamic Rupture

The earthquake dynamic rupture scenarios with nucleation in the West and East of the complex fault geometry yield a significantly smaller moment magnitude than the other four scenarios, which is reflected in the moment rates (Fig. 4). Their rupture fronts propagate only $\sim$30 km due to the high fault segmentation and fault gaps inhibiting dynamic triggering and multiple rupture jumps. While the scenario on the complex fault geometry with the hypocenter in the middle breaks a greater extent of the fault system, its seismic moment is still smaller than all three scenarios on the simpler fault geometry due to reduced maximum fault slip and the smaller ruptured area. All earthquake dynamic ruptures on the simpler fault geometry break over the entire main fault length and generate larger maximum slip (Fig. A2). Scenario Simple-East produces the largest maximum

fault slip localized at the offshore section of the fault system (7.90 m) with an average fault slip of 4.93 m (Table 2). We con-
sider those parts of the fault which experience at least 0.01 m coseismic slip for computing the average fault slip. Furthermore,
the three earthquake dynamic rupture simulations on the simple fault geometry cause significant shallow fault slip resulting in
a negligible shallow slip deficit (SSD, Fig. A3). The SSD ratio is defined as the ratio of near-surface slip to slip at seismogenic
depths (e.g., Fialko et al., 2005; Marchandon et al., 2021). A higher percentage of SSD indicates that fault slip occurring at
depth is larger compared to slip in the uppermost part of the fault. This is the case for all three scenarios on the complex fault
geometry. Our six dynamic rupture simulations show the accumulation of plastic strain surrounding the fault traces (Fig. A1),
where the resulting off-fault plastic strain distribution with depth (Fig. 3) resembles a shallow flower-shape structure enclosing
the fault (e.g., Ben-Zion et al., 2003; Rockwell and Ben-Zion, 2007; Ma, 2008; Ma and Andrews, 2010; Schliwa and Gabriel,
2022).

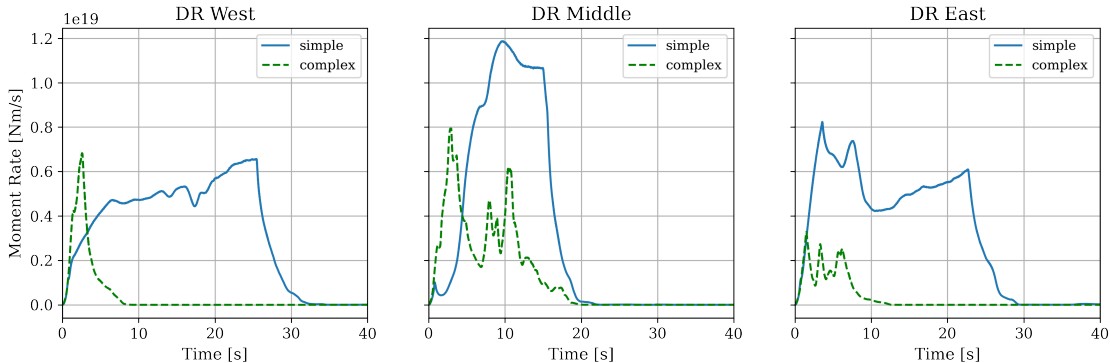

**Figure 4.** Moment release rates for the six earthquake dynamic rupture (DR) simulations (up to $t = 40\ s$). Multiple peaks for Complex-
Middle and Complex-East correspond to the rupture decelerating before jumping to (that is, dynamically triggering) the next fault segment.

### 3.1.1 Seafloor Displacement

The coseismic earthquake displacements reach up to ~1 m of seafloor uplift and up to ~0.8 m of subsidence (Fig. 5, Table
2) for ruptures on the simpler fault geometry with nucleations in the East and Middle of the HFFZ. The earthquake dynamic
rupture simulation with the western hypocenter on the simpler fault geometry reveals that major displacement occurs onshore.
This has a significant impact on seismic hazard assessment, in particular for the town of Húsavík, which is located directly
above the HFFZ (Fig. 1). The tsunami potential of scenarios with Western epicenters is expected to be smaller for Húsavík.
While the maximal coseismic seafloor subsidence of the simulation Complex-Middle is equivalent in size to the maximum
subsidence observed for the scenarios on the simpler fault geometry, the rupture generates only half as much uplift (Table 2).
The total offset of the vertical displacement for the scenario Complex-Middle is ~1.2 m, which matches the vertical offset
for scenario Complex-West. However, the latter offset is restricted to the western end of the HFFZ north of Siglufjörður and
Eyjafjörður Bay. Meanwhile, the displacement pattern of scenario Complex-East mainly affects Skjálfandi Bay and Húsavík.

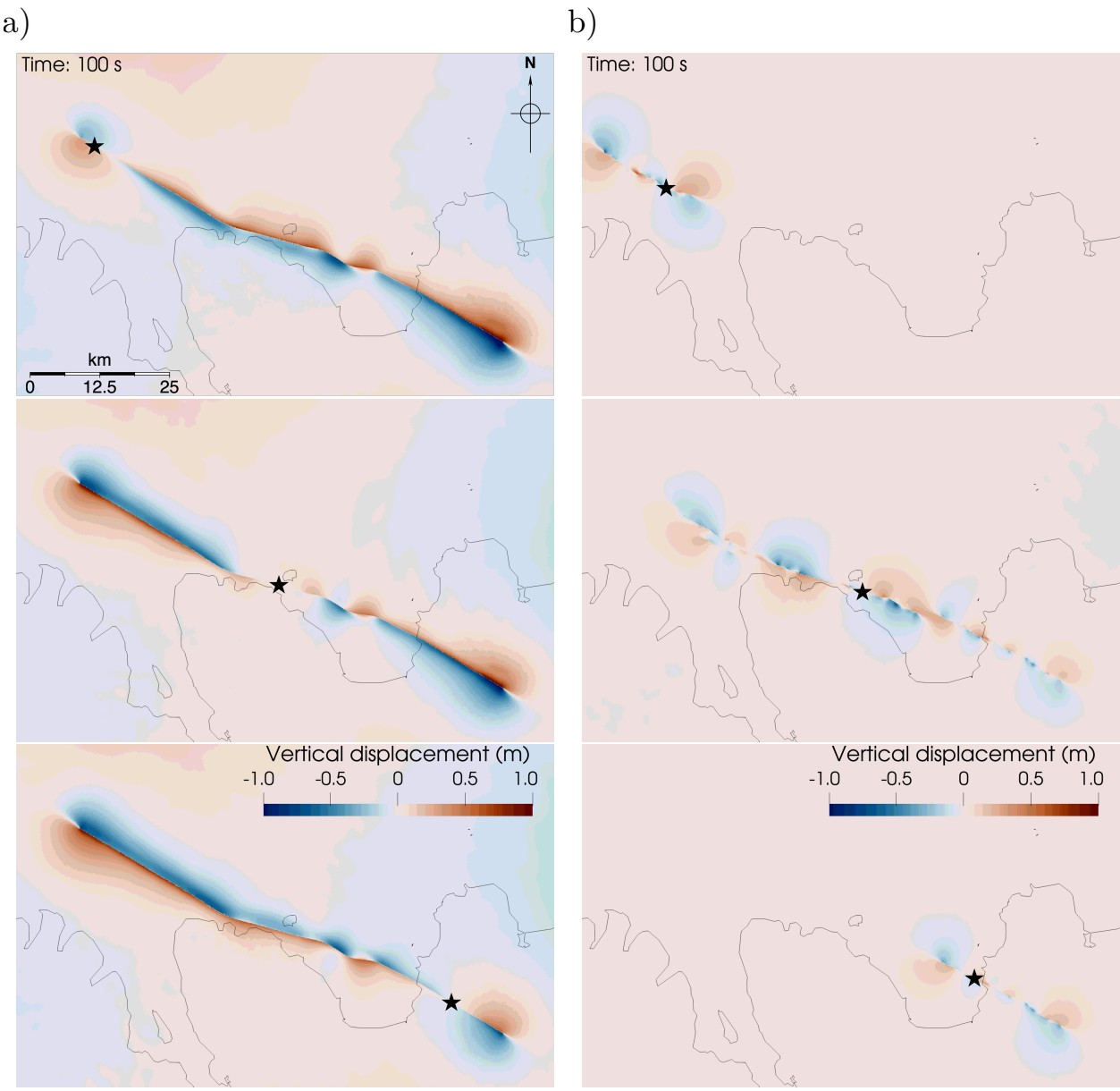

**Figure 5.** Uplift and subsidence from the surface displacements of earthquake dynamic rupture simulations after accounting for local bathymetry using the Tanioka filter (Tanioka and Satake, 1996) on a) the simple fault geometry and b) the complex fault geometry. Black stars mark the epicenter locations.

 **3.1.2 Rake**

Earthquakes on a vertically dipping, right-lateral fault system, such as the HFFZ, predominantly exhibit rake angles of $180°$. However, we observe dynamic rake rotation ($\pm20°$) near the surface during the rupture (Fig. 6). In our models, dynamic rake rotation (interacting with local bathymetry) explains the higher-than-expected vertical seafloor displacements due to the transient changes in slip direction inducing dip-slip components.

| | simple fault geometry | | | complex fault geometry | | |
|---|---|---|---|---|---|---|
| Hypocenter | West | Middle | East | West | Middle | East |
| $M_W$ | 7.34 | 7.33 | 7.34 | 6.74 | 7.07 | 6.68 |
| Avg. fault slip [m] | 5.03 | 4.80 | 4.93 | 2.14 | 1.97 | 1.51 |
| Max. fault slip [m] | 10.34 | 8.11 | 7.90 | 3.50 | 5.23 | 2.74 |
| Max. fault slip offshore [m] | 6.93 | 6.58 | 7.90 | 3.50 | 5.23 | 2.74 |
| Max. peak slip rate [m s$^{-1}$] | 15.05 | 14.93 | 15.14 | 10.44 | 11.59 | 8.66 |
| Max. peak slip rate offshore [m s$^{-1}$] | 13.53 | 12.58 | 15.14 | 10.44 | 11.59 | 8.62 |
| Max. seafloor uplift [m] (after Tanioka filter) | 0.75 | 1.05 | 0.95 | 0.56 | 0.44 | 0.23 |
| Max. seafloor subsidence [m] (after Tanioka filter) | -0.74 | -0.79 | -0.76 | -0.66 | -0.79 | -0.42 |

**Table 2.** Key results of our here considered six earthquake dynamic rupture scenarios. Note that we only report the maximum offshore coseismic vertical displacements (i.e., seafloor offsets) in the table because the onshore vertical displacements do not contribute to the tsunami generation.

 **3.2 Time-dependent tsunami generation**

The coastline of North Iceland includes several smaller islands, like Flatey Island, Grímsey Island, and Hrísey Island (Fig. 1). Furthermore, the region of Norðurland eystra in North Iceland includes steep terrain, elongated fjords such as Eyjafjörður and bays with a shallower shoreline like Skjálfandi Bay. Accounting for these diverse coastal features and the complex bathymetry offshore Northern Iceland, we analyze the one-way linked scenarios with simulation times long enough to compare  the tsunami's impact at synthetic tide gauge stations placed near coastal towns in Sect. 3.2.1. In Sect. 3.2.2, we use fully-coupled scenarios on the simpler fault geometry to study the full dynamics of tsunami generation and earthquake-tsunami interaction.

**3.2.1 One-way linked scenarios**

We define the sea surface height anomaly (ssha) as deviation from the ocean surface at rest. We place six synthetic tide gauge stations offshore, in direct proximity to the towns of Húsavík, Akureyri, Dalvík, Ólafsfjörður, Siglufjörður and Grímsey Island.  Every tsunami is simulated for 40 min.

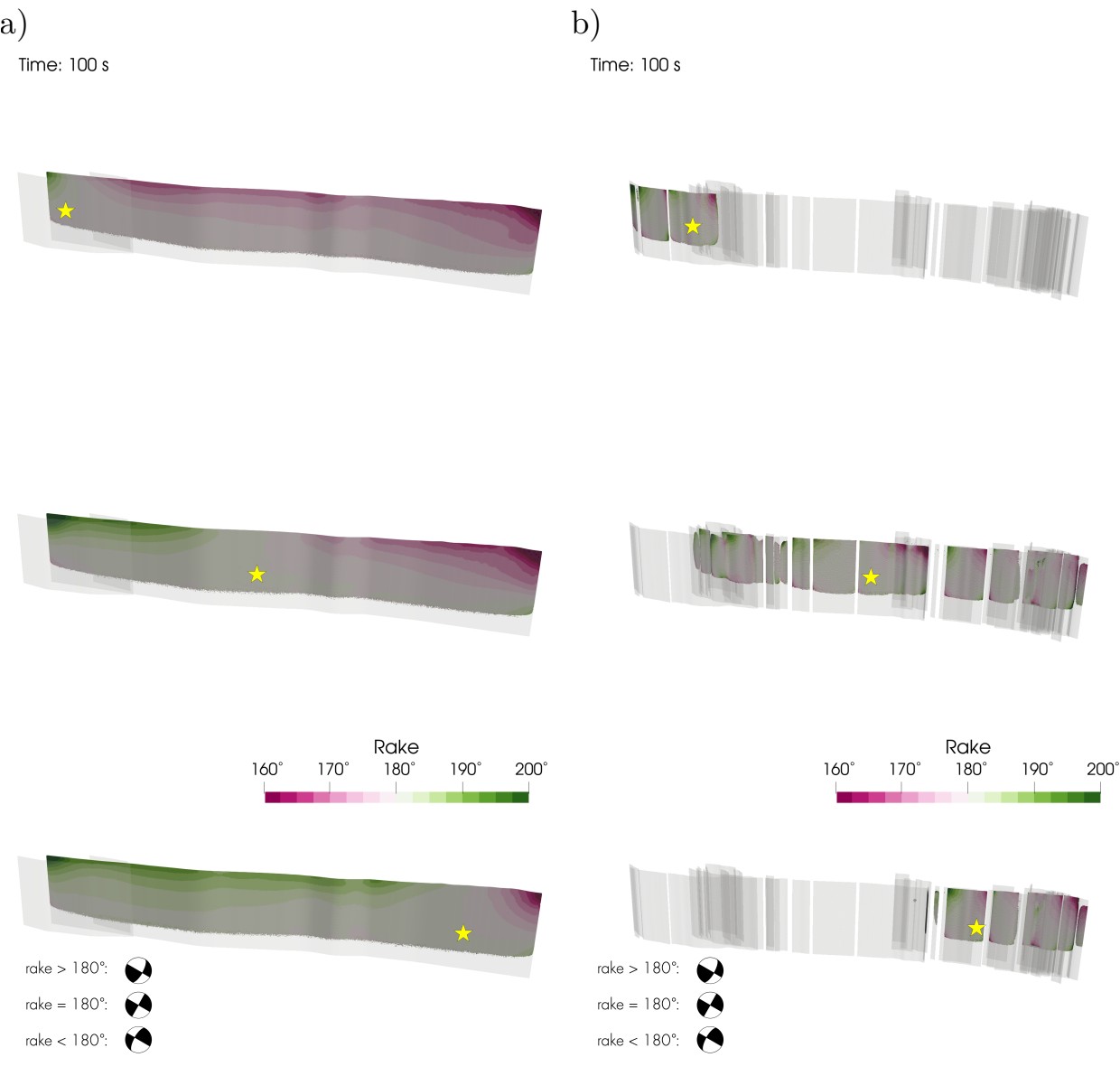

**Figure 6.** Dynamic rake rotation in the dynamic rupture simulations on a) the simple fault geometry and b) the complex fault geometry. Yellow stars mark the hypocenter locations. A rake of 180 degrees indicates pure right-lateral strike-slip faulting.

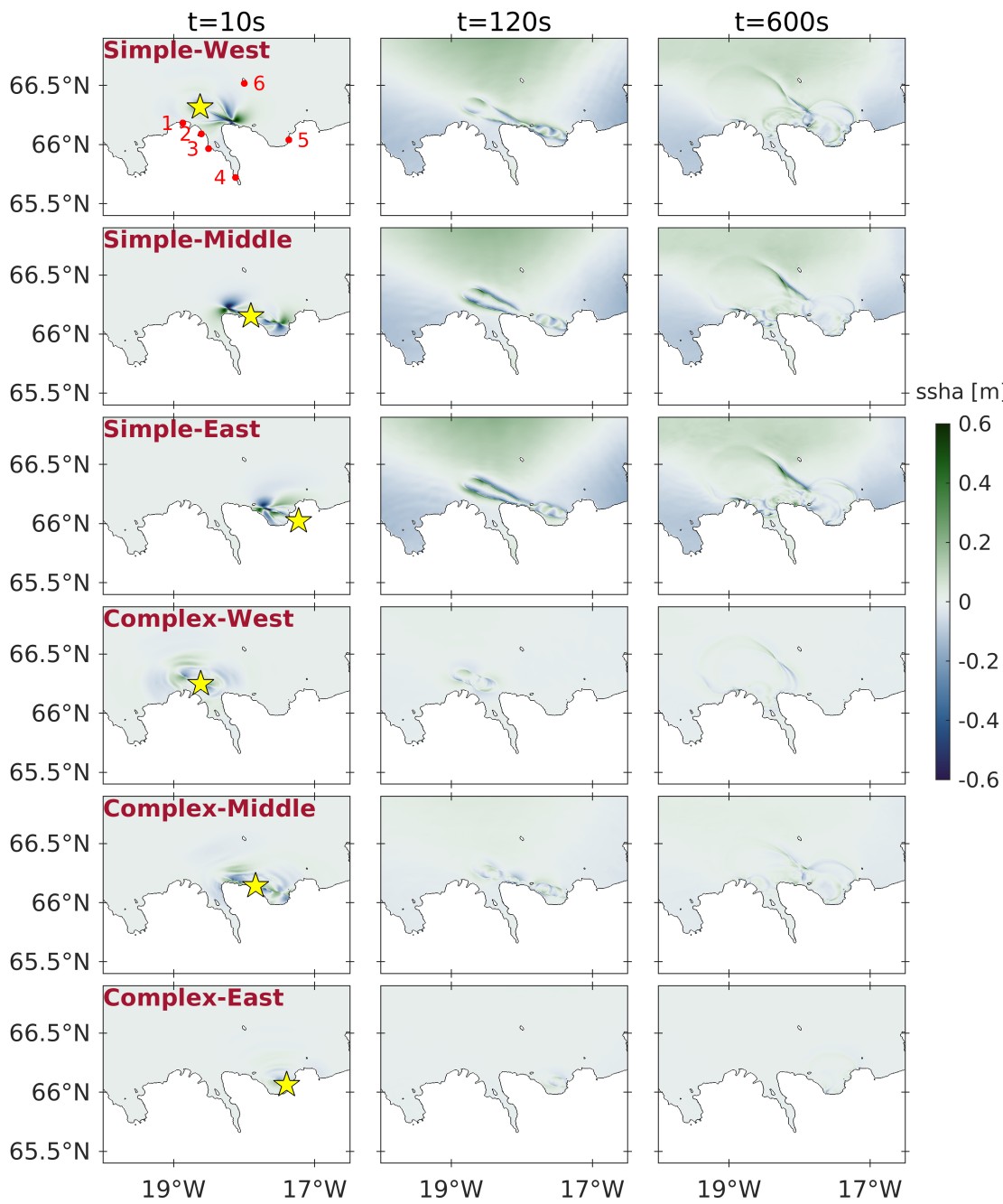

**Figure 7.** Sea surface height anomaly (ssha) of all six one-way linked earthquake-tsunami scenarios at 10 s (first column), 2 min (second column), and 10 min (third column) simulation times. The yellow star in the first column marks the epicenter of each scenario. The red points in the top-left panel indicates the position of synthetic tide gauges near the coastal towns (1) Siglufjörður, (2) Ólafsfjörður, (3) Dalvík, (4) Akureyri, (5) Húsavík (west to east on the mainland) and (6) Grímsey Island. The faint distant blue and green coloring is due to the static vertical displacement as shown in Fig. 5, which is more pronounced for scenarios on the simpler fault geometry.

We show snapshots of tsunami propagation after 120 s and 600 s in Fig. 7. The first column in Fig. 7 shows the complexity of the time-dependent seafloor displacements in all dynamic rupture sources superimposing seismic wave propagation after 10 s.

We first analyze the three one-way-linked tsunami scenarios sourced by dynamic rupture simulations on the simpler fault geometry which cause overall larger wave heights. All dynamic ruptures on the simpler fault geometry are still propagating after 10 s. The corresponding snapshots in Fig. 7 highlight source directivity effects for the simple-fault-geometry scenarios. Earthquake ruptures in the scenarios Simple-Middle and Simple-East arrive at Skjálfandi Bay within the first seconds. The unilateral dynamic rupture in scenario Simple-West requires more time to propagate eastwards towards Húsavík and causes a higher maximum ssha of 27 cm at the corresponding synthetic tide gauge (Fig. 8 and 9).

The initiating tsunami wavefronts from scenarios Simple-East and Simple-Middle evolve similarly, visible in the snapshots at 2 min and 10 min propagation time. In distinction, while the tsunami front from scenario Simple-West appears at a comparable location, its sea surface height anomaly is smaller than the ssha from the previous two scenarios. The synthetic tide gauge stations reveal that scenario Simple-East produces slightly larger wave amplitudes than the other two scenarios on the simpler fault geometry (with an exception at station Húsavík, cf. Fig. 8 and 9). Scenario Simple-East generates a maximum crest-to-valley difference (wave height) of 0.9 m near Ólafsfjörður. Positive amplitudes (i.e., the maximum distance between the highest point of a tsunami wave crest and the ocean at rest) greater than 30 cm can also be observed near Dalvík and Grímsey Island. The Scenario Simple-East tsunami continues propagating towards Akureyri but with locally significantly decreased amplitudes.

Overall, the tsunami scenarios initiated by dynamic rupture scenarios on the complex fault geometry cause smaller tsunamis (Fig. 7, see bottom three rows). In contrast to the scenarios on the simpler fault geometry, now the respective tsunami characteristics are highly dependent on epicentral location. The tsunami in scenario Complex-West arrives at the tip of Iceland's north coast within the first 5 min and then propagates directly into the bay of Eyjafjörður, arriving at Dalvík at around 10 min (Fig. B1). Wave heights for scenario Complex-West exceed ~20 cm at Siglufjörður and Grímsey Island and are the largest among the three scenarios on the complex fault geometry in Ólafsfjörður with ~40 cm.

The tsunami scenario caused by dynamic rupture in the middle of the complex Húsavík-Flatey Fault, in scenario Complex-Middle, affects the town of Húsavík within the first minute. Wave heights reach ~20 cm at several locations across Skjálfandi Bay. The tsunami enters the neighboring fjord Eyjafjörður arriving in Dalvík after approximately 15 min, with amplitudes up to ~10 cm, but again decays before reaching Akureyri. The tsunami generated in scenario Complex-East remains completely bounded by the bay surrounding Húsavík. Consequently, waves only expand within Skjálfandi Bay for about 10 min reaching ssha on the order of ±10 cm. Due to its lower wave heights, this tsunami marginally signals at any of the other synthetic tide gauge stations.

### 3.2.2 3D fully-coupled scenarios

Based on the results from the one-way linked simulations we select those earthquake-tsunami scenarios causing larger wave heights for the computationally more demanding fully-coupled models. A single fully-coupled simulation of joint dynamic

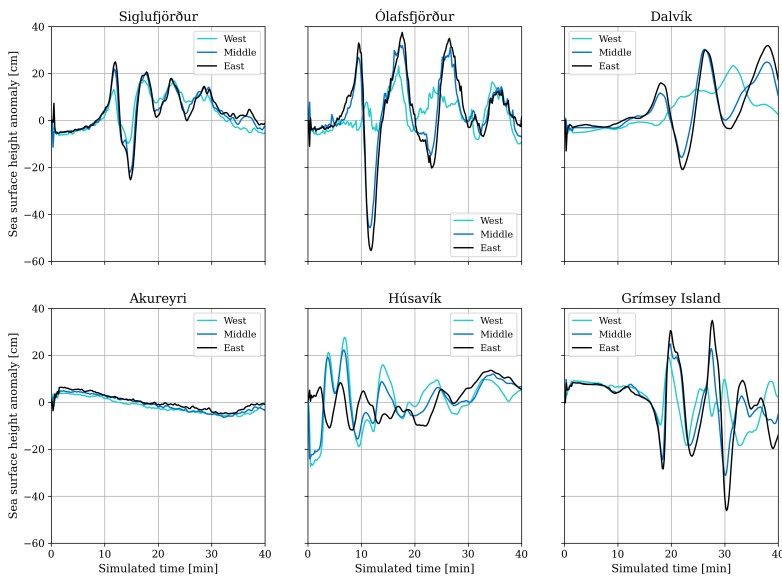

**Figure 8.** Sea surface height anomaly (ssha [cm]) vs simulation time (40 min) for the three one-way-linked scenarios sourced by dynamic rupture simulations on the simpler fault geometry recorded at six synthetic tide gauge stations close to the towns Siglufjörður, Ólafsfjörður, Dalvík, Akureyri, Húsavík (west to east on the mainland) and Grímsey Island.

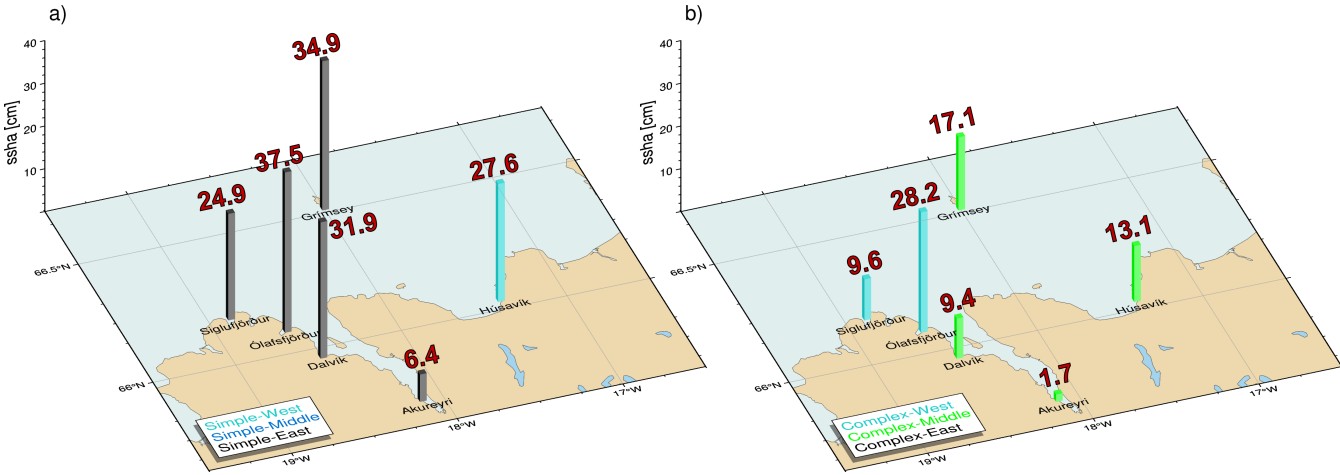

**Figure 9.** Maximum sea surface height anomaly (ssha [cm]) recorded throughout the simulation time of 40 minutes at synthetic tide gauge stations nearby local communities in North Iceland for the one-way linked scenarios based on the simpler fault geometry (a) and the complex fault geometry (b). At each tide gauge, we show the maximum ssha of all three respective scenarios, with bar colors indicating the epicentral location of the scenario causing maximum ssha at a given location.

rupture and tsunami generation (for 3 min of simulated time) requires ∼4 h computational time with 40 nodes (1920 cores), that is, a total of 7680 CPUh, on the Munich supercomputer SuperMUC-NG (https://doku.lrz.de/supermuc-ng-10745965.html). To first order, the fully-coupled tsunami simulations match the seismic and tsunami waveforms obtained using the one-way linked approach (Fig. B2). Seismic waves result in transient motions of the sea surface and affect the ocean response but do not appear to contribute to tsunami generation. However, in the fully-coupled simulations seismic waves within Earth,

acoustic waves within the ocean, and wave conversions superimpose. We show the three 3D fully-coupled dynamic rupture scenarios using the simple fault geometry in Figs. 10, B3, B4 to better understand the dynamic tsunami generation and complex superposition of different wave types. Their interaction is visible in panels (a) and (b), where we illustrate the sea surface height anomaly (ssha) and sea surface vertical velocity (ssvv) after 20 s simulated time. Close to the fault, we see the excited tsunami waves, best visible in panels (a), which start to propagate away from the ruptured fault system. At the same time, faster

propagating acoustic waves already approach the water layer boundaries. We select two profiles approximately perpendicular to the fault system's strike direction of each fully-coupled scenario, with their bathymetry shown in panels (c). Along these two cross-sections, we plot the space-time evolution of ssvv in the respective panels (d) of Figs. 10, B3, B4. They indicate distinct features.

First, we see the propagation of the tsunami at a speed of ∼35 m s$^{-1}$ towards the open ocean. The tsunami waves in all

three scenarios travel 5.6 km in 160 s (cross-section 2 in Figs. 10, B3, B4). A slightly larger – yet comparable – value of ∼41.8 m s$^{-1}$ can be calculated using the relation $\sqrt{g \cdot H}$ for the tsunami velocity, approximating the gravitational acceleration as $g = 10$ m s$^{-2}$ and the average water depth as $H = 175$ m from cross-section 2 in panel (c) of Figs. 10, B3, B4. The tsunami waves visible in cross-section 1 of all panels (d) show a decrease in wave velocity (now ∼20 m s$^{-1}$) as the tsunami front approaches the shoreline, which is located at 0 km. This expected effect is caused by the reduction of the local bathymetry to

less than 40 m depth, evident at a distance of 20 km away from the coast (Figs. 10c, B3c, B4c, cross-section 1).

Second, we observe the complex seismo-acoustic wave excitation and interaction in the initial phase of tsunami generation. The high-amplitude acoustic waves are clearly visible in all three scenarios. The seismic-generated acoustic waves propagate at a speed of $c_0 = 1500$ m s$^{-1}$ and are, therefore, much faster than the oceanic tsunami. Importantly, the ssvv amplitudes caused by the acoustic waves are larger than those corresponding to the actual tsunami.

Next to ocean acoustic waves, we observe normal dispersion, i.e. frequency-dependent wave speeds, of the tsunami (Fig. 11). We use the same two cross-sections as before and show ssha (rows one and three, Fig. B5) and ssvv (rows two and four, Fig. B5) at a simulated time of 2 min for the three scenarios on the simpler fault geometry, given both tsunami modeling techniques. The one-way linked tsunami waveforms (dashed black line) for both cross-sections in the first and third row are rather smooth. We note that accounting for the dispersive effects, which are not considered in the depth-integrated (hydrostatic) shallow water

equations we are solving for, could potentially lead to less smooth synthetics. Their overall trend including the spatial location of peaks and troughs is well matched by the corresponding fully-coupled waveforms (solid blue line). However, a close look into the waveforms indicates additional short-period signals in-between wave crests and troughs. A zoom into the sea surface vertical velocities (Fig. 11, B5), reveals multiple distinct wavefronts reflecting normal dispersion effects. In contrast, anomalous

dispersion, where shorter wavelengths (higher frequencies) propagate faster than longer wavelengths (lower frequencies), can
not be identified in our simulations, which is expected due to the locally shallow ocean (Abrahams et al., 2023).

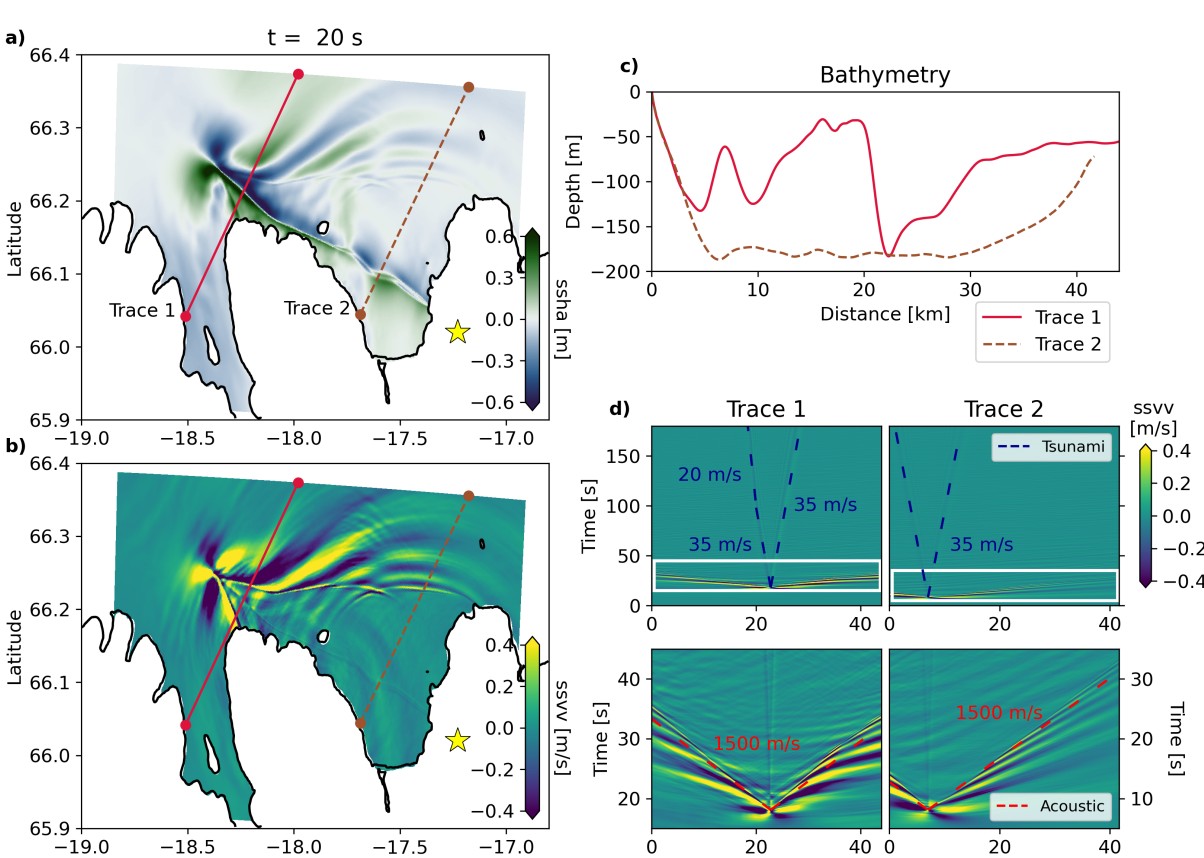

**Figure 10.** 3D fully-coupled earthquake-tsunami scenario Simple-East, with dynamic rupture on the simple fault geometry and a hypocenter in the East (yellow star). Snapshots at $t = 20\ s$ of a) the sea surface height anomalies (ssha) and b) sea surface vertical velocity (ssvv). c) Corresponding bathymetry profiles along the two selected cross-sections stretching from the shoreline (0 km) towards the open ocean. d) Space-time evolution of ssvv along the two cross-sections for the full duration of the fully-coupled simulations (upper row, highlighting the tsunami and the superposition of near-field displacements, seismic and acoustic waves). The white box indicates the zoom on the tsunami generation (lower row, highlighting the fast propagating acoustic waves). Simulation results for the fully-coupled scenarios Simple-Middle and Simple-West are shown in Figs. B3 and B4 respectively.

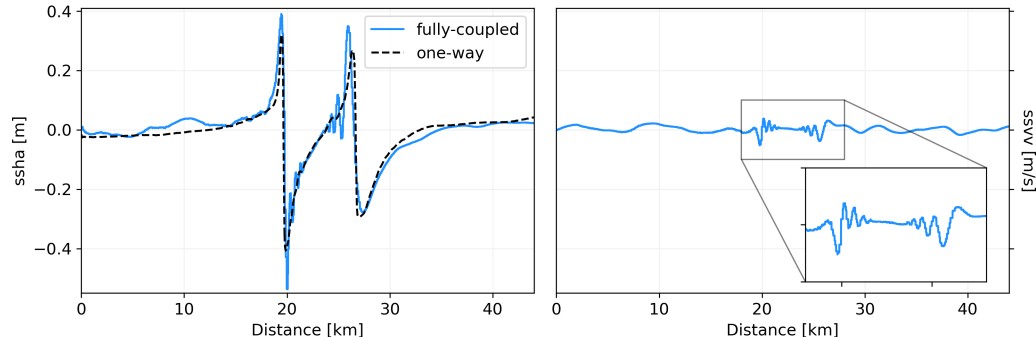

**Figure 11.** a) Sea surface height anomaly (ssha [m]) for scenario Simple-East along cross-section 1 at $t$ =2 min for the fully-coupled (solid blue line) and one-way linked (dashed black line) simulations. The overall trend of the one-way linked waveform, i.e., the spatial location of peaks and troughs, is well matched by the corresponding fully-coupled waveform. b) Sea surface vertical velocity (ssvv [m s$^{-1}$]) for the fully-coupled scenario Simple-East along trace 1 at $t$ =2 min highlighting tsunami normal dispersion. The shoreline is located at 0 km. Fig. B5 shows a comparison of the simulation results for all three scenarios on the simpler fault geometry along both cross-sections.

## 4 Discussion

Submarine ruptures across strike-slip fault systems were long assumed to produce only minor vertical offsets and hence no significant disturbance of the water column. Linked and fully-coupled earthquake dynamic rupture and tsunami modeling for the 2018 $M_W$ 7.5 Sulawesi earthquake in Indonesia suggest that coseismic-induced seafloor displacements critically contributed to the generation of an unexpected and devastating local tsunami in Palu Bay (e.g., Ulrich et al., 2019b; Krenz et al., 2021; Ma, 2022). Widespread liquefaction-induced coastal and submarine landslides likely also played an important role (e.g., Carvajal et al., 2019; Gusman et al., 2019; Pakoksung et al., 2019; Sassa and Takagawa, 2019; Sepúlveda et al., 2020). Our simulations of six earthquake dynamic rupture scenarios show that the Húsavík-Flatey Fault Zone can host tsunamigenic earthquakes. This may have important implications for tsunami hazard assessment of submarine strike-slip fault systems in transform and transtensional tectonic settings worldwide. For example, the North-Alfeo Fault in the Ionian Sea may be capable to generate a $M_W \approx 7$ strike-slip earthquake (Scicchitano et al., 2022), but is often not considered in tsunami modeling.

Our earthquake dynamic rupture scenarios can generate enough vertical seafloor displacements to source a localized tsunami. The scenarios on the simpler fault geometry may be considered worst-case events because the ruptures break over the entire main fault length accumulating large fault slip (equivalent to $\sim M_W$ 7.3). The moment magnitudes of our dynamic rupture models on the complex fault geometry are lower ($M_W$ 6.7 − $M_W$ 7.0) and involve more segmented slip due to rupture-jumping across the highly segmented fault network. Large-scale geometric fault complexity can act as an "earthquake gate" (Oskin et al., 2015; Lozos, 2016; Duan et al., 2019; Liu et al., 2021). An earthquake gate is a mechanical barrier to earthquake rupture (Liu et al., 2022), which has the potential to alter the rupture extent. In the case of the simpler fault geometry, the restraining and releasing bend east of Flatey Island may be considered as such an earthquake gate since this smooth main fault bend does allow

some ruptures to propagate across while terminating others depending on the local pre-stress and dynamic stress evolution (Li et al., 2023). The segmented, more complex fault geometry is more effective in dynamically arresting earthquake rupture (Segall and Pollard, 1980; Wesnousky, 2006). However, large earthquakes are still dynamically possible with rupture capable of jumping across several fault stepovers before eventually terminating.

For all scenarios, we observe pronounced dynamic rake rotation near the surface, which we consider as plausible dynamic mechanism for generating increased coseismic vertical offset. The dynamic deviations from pure right-lateral strike-slip faulting are on the order of $\pm 20°$ and introduce significant shallow dip-slip motion. Thereby, vertical seafloor displacements in our simulations are enhanced, which are critical for tsunami generation. Shallow rake rotation has been inferred for surface-breaking earthquakes using geological slickenlines and simple dynamic rupture models (Spudich et al., 1998; Guatteri and Spudich, 1998; Kearse et al., 2019). Vertical stress changes at the rupture front cause this change in rake angle, which is more pronounced near the surface due to smaller confining stresses (Kearse and Kaneko, 2020). No rake rotation is expected directly atop the hypocenter, which is confirmed in Fig. 6. In Fig. 6, a gradual increase of rake rotation can be observed away from the hypocenter for both unilateral and bilateral rupture scenarios on the simpler fault geometry. More complex patterns of rake rotation result in the scenarios on the complex fault geometry, and we observe a dependence of the spatial distribution of rake rotation on the fault segment length and on hypocenter location. Changes in rake for the right-lateral strike-slip earthquake dynamic rupture scenarios cause mostly uplift in the compressional quadrants. Gaudreau et al. (2023) investigate the 1971 San Fernando thrust faulting earthquake using aerial stereo photographs and discuss a rotation of rake away from the pre-stress direction. He et al. (2022) use InSAR, GNSS, and optical data to study the 2019 Ridgecrest Sequence and report vertical cumulative coseismic surface displacement after the sequence, interpreted as an indication of prominent coseismic rake rotation. Other studies focusing on finite-fault models, allowing for rake variations in their inversions for slip, show rake rotation most prominent at patches, which are near the surface and have a large slip magnitude (Metzger et al., 2017; Feng et al., 2017; Xiao et al., 2022; Hong et al., 2022).

In contrast to the suggested important contribution of off-fault deformation to strike-slip tsunami generation in Palu Bay (Ma, 2022), the effect of off-fault plasticity is likely small in our simulations (Fig. A1). We find that off-fault deformation contributes only about ∼3 % of the total seismic moment. Accounting for the potential existence of additional shallow, weak sediments, which are more prone to off-fault plastic deformation, may increase local uplift (Seno and Hirata, 2007; Ma and Nie, 2019; Wilson and Ma, 2021; Ulrich et al., 2022; Ma, 2023).

Modeling tsunami scenarios for hazard assessment or rapidly after submarine earthquakes often relies on simplifications, such as the negligence of source time-dependency, only considering vertical seafloor deformation without bathymetry effects, solely planar fault geometries, or neglecting tsunami dispersion and acoustic wave effects. Abrahams et al. (2023) introduce non-dimensional parameters allowing quantifying the range of validity for certain modeling assumptions. Our average water depth $H$ can be approximated as ∼200 m, the source width $\sigma_r$ as given by the length of the HFFZ (∼100 km), the source duration $\sigma_t$ of 30 s constrained by the rupture duration (cf. moment rates Fig. 4), the gravitational acceleration $g = 10$ m s$^{-2}$ and acoustic wave speed $c_0 = 1500$ m s$^{-1}$, so we can calculate the three non-dimensional numbers posed by Abrahams et al. (2023) as specified in Table 3.

We see that the shallow water limit is fulfilled ($H/\sigma_r \ll 1$), which justifies using our one-way linked earthquake-tsunami modeling approach. While we use time-dependent seafloor displacements, our source should appear effectively as instantaneous to tsunami waves ($\sqrt{gH} \cdot \sigma_t/\sigma_r \ll 1$) due to the relatively short rupture duration and shallow water depth. This fact explains the similarity in the tsunami propagation and shape of the tsunami wavefronts for the simple fault geometry scenarios (i.e., Fig. 7 after 2 min (second column) and 10 min (third column)), which all break the entire main fault length and lead to

similar fault slip distributions. In contrast, the scenarios on the complex fault geometry differ distinctly in their final fault slip distributions and areas of seafloor displacements depending on the chosen epicenter location. However, to compare the tsunami generation phase, it is indispensable to consider a time-dependent source model for both approaches, the one-way linked and fully-coupled method, for comparability.

   From the average water depth $H$ being much smaller than $c_0 \cdot \sigma_t$ (Table 3) we expect that it is justified to neglect acoustic wave

excitation since their amplitudes should be small. However, our fully-coupled simulations include acoustic wave generation with high vertical velocity amplitudes, larger than the tsunami signals (Figs. 10, B3, B4). Dynamic rupture reaching the Earth's surface can cause strong radiation (Kaneko and Goto, 2022) including the generation of high-frequency seismic waves due to the locally strong deceleration at the rupture front (e.g., Madariaga et al., 2006; Okuwaki et al., 2014; Li et al., 2022). Part of this seismic wave energy is converted to ocean-acoustic waves at the seafloor (e.g., Krenz et al., 2023), as observed during

the 2011 Tohoku-Oki (e.g., Maeda et al., 2013) and the 2003 Tokachi-Oki earthquakes (e.g., Nosov and Kolesov, 2007) using ocean-bottom pressure sensors. Earlier studies found that the conversion between seismic and ocean acoustic waves occurs predominantly at slopes of the seafloor (e.g., Noguchi et al., 2013). Here, however, local bathymetry is generally flat, and the conversion is dominated by dynamic source complexity, such as surface rupture and the associated shallow rake rotation.

   In our study, we compare a simple fault geometry representing the Húsavík-Flatey Fault Zone and a very complex fault

network consisting of 55 individual fault segments. Klinger (2010) and Lefevre et al. (2020) proposed a linear relationship between the thickness of the seismogenic crust (brittle upper crust) and the length of fault segments for strike-slip geometries. This would imply a relatively short average fault segment length given the locking depth of 6 – 10 km for the HFFZ (Metzger and Jónsson, 2014). Therefore, during a large strike-slip earthquake along the HFFZ, rupture may segment into several subevents (Jiao et al., 2021; Klinger, 2022), as resembled in our Complex-Middle scenario. However, Iceland, offering a unique

geologic complexity, is located atop the Mid-Atlantic Ridge and influenced by the underlying mantle plume (Torsvik et al., 2015; Celli et al., 2021) with significantly varying crustal thickness over the last 56 Ma (Hjartarson et al., 2017), potentially altering established scaling relation. Hence, it is important to include different fault structural complexity within earthquake and tsunami simulations to accurately capture plausible earthquake scenarios.

   The three tsunami scenarios sourced by dynamic rupture simulations across the complex fault geometry cause significantly

smaller tsunamis. This is due to lower and more segmented fault slip leading to less vertical seafloor displacements, which are spatially more restricted. The largest total wave height (i.e., crest-to-trough difference) of ∼40 cm is observed at the synthetic tide gauge stations near Grímsey Island for scenario Complex-Middle and near Ólafsfjörður for Complex-West. The tsunami from scenario Complex-East does not have a significant impact on the virtual tide gauges since much of the coseismic ground

displacement occurs onshore. The town Ólafsfjörður is also highly exposed to tsunami signals in the scenarios using the simpler fault geometry with wave heights reaching 0.9 m.

We find that our scenario Simple-East poses the largest impact for coastal communities, except for Húsavík. Here the hypocenter is near the town, which may experience strong ground shaking (Li et al., 2023), but not a large tsunami. However, Húsavík can be affected by scenario Simple-West causing nearly 60 cm crest-to-trough difference. This unilateral rupture nucleating at the western end of the HFFZ builds up energy while propagating towards Húsavík, explaining the larger observations. We find that none of our scenarios endanger the town of Akureyri, which is shielded by the narrow Eyjafjörður. The modeled tsunami does not amplify but loses energy due to multiple reflections within the bay and due to the protection by Hrísey Island.

Ruiz-Angulo et al. (2019) performed a preliminary investigation of the tsunami potential for the Húsavík-Flatey Fault Zone using a uniform fault-slip earthquake dislocation source with a moment magnitude of 7.0, located in the middle of the fault system. They utilized the Okada method (Okada, 1985) with instantaneous sourcing of the tsunami by the final static displacements. Their maximum synthetic crest-to-trough difference of ∼30 cm also occurs at Ólafsfjörður. While this is slightly larger than the maximum crest-to-trough difference of 26 cm which we observe for the scenario Complex-Middle, it is a factor of 2.5 smaller than our scenario Simple-Middle (77 cm).

These differences may be due to our dynamic rupture models including dynamically evolving relatively large shallow fault slip (up to ∼8 m for Simple-East) with no SSD for scenarios on the simpler fault geometry and near-surface rake rotation ($\pm 20°$). This results in higher-than-expected coseismic vertical displacements ($\pm 1$ m). In addition, we include local bathymetry and, to a smaller extent, off-fault plastic deformation, all contributing to the tsunami generation. We do not consider combined earthquake and landslide-induced tsunami scenarios for the HFFZ, which can additionally increase the local tsunami height (Ruiz-Angulo et al., 2019).

We extend recent 3D dynamic rupture models by Li et al. (2023) to tsunami modeling. In these scenarios, relatively high peak fault slip localizes near the free surface while the average fault slip is overall ∼40 % smaller than the peak fault slip (Table 2). Li et al. (2023) show that the synthetic ground motions produced by such dynamic rupture models are in good agreement with the latest regional empirical ground motion model (Kowsari et al., 2020). While the moment magnitudes of some of these models are larger than previous slip-deficit based estimates of the accumulated moment along the HFFZ (e.g, $M_W$ 6.8 ± 0.1, Metzger et al. (2011)) they have comparable moment magnitudes to historic events. Slip-deficit based magnitude estimates are typically relying on several assumptions including in this case a complete stress relaxation of the 1872 $M_W$ 6.5 earthquake and subsequent steady stress accumulation. The dynamic rupture scenarios are consistent with the average fault slip and effective rupture area scaling relations of Mai and Beroza (2000), which have recently been validated for the Southern Iceland Seismic Zone (SISZ, Bayat et al. (2022)). The SISZ is similar in its tectonic and seismic context to the TFZ in Northern Iceland.

Our simulated maximum fault slip occurring within the shallow offshore part of the HFFZ, i.e., the part which is relevant for the subsequent tsunami generation, is comparable to geological observations from earthquakes rupturing along faults with similar length to the HFFZ. Examples of strike-slip ruptures as summarized by Wesnousky (2008) comprise Neo Dani, Japan (length 80 km, max. slip 7.9 m, Matsuda (1974)), Luzon, Philippines (length 112 km, max. slip 6.2 m, Yomogida and Nakata

(1994)), and Landers, California (length 77 km, max. slip 6.7 m, Sieh et al. (1993)). A recent example includes the second event of the devastating Kahramanmaraş earthquake sequence (length 150 km) resulting in up to 8 m fault slip near the surface (Jia et al., 2023). While we use a LSW friction law, considering a rate-and-state dependent friction law would allow to include shallow velocity-strengthening behavior which may decrease slip in the shallowest parts of the fault (e.g., Kaneko et al., 2008).

We exclude inundation in the one-way linked approach to enable a more meaningful comparison with the fully-coupled method. Our fully-coupled simulations are computationally demanding and do not allow to model inundation (e.g., Krenz et al., 2021). We show that the fault geometry in our six one-way linked scenarios can influence the subsequent tsunami generation. Here, future studies may explore potential variations in fault dip, which may further enhance the vertical seafloor displacement during the earthquake rupture. Changes in earthquake source parameters are known to affect the maximum tsunami height (e.g., Burbidge et al., 2015) and resulting inundation (e.g., Gibbons et al., 2022).

Our 3D fully-coupled simulations include unexpectedly high-amplitude acoustic waves, which may serve as a rapid indicator of surface dynamic rupture. A better understanding of such acoustic wave signals may improve tsunami early warning since these can be detected earlier in the far-field, e.g., at ocean bottom pressure sensors, in comparison to the tsunami recorded at conventional DART buoys (Yamamoto, 1982; Cecioni et al., 2014; Mei and Kadri, 2017; Gomez and Kadri, 2021). However, in the near-field, ocean acoustic waves can superimpose onto tsunami signals, impeding early warning efforts. In addition to the seismo-acoustic wave excitation in the fully-coupled simulations, we observe dispersion of tsunami propagation velocity (Tsai et al., 2013). Glimsdal et al. (2013) show that enhanced dispersion effects are expected for earthquakes with magnitude 8 and less as opposed to less dispersive tsunamis caused by the largest earthquakes. We do not account for dispersion in the one-way linked tsunami simulations that are based on solving the non-linear hydrostatic shallow water equations. However, Boussinesq-type tsunami models can account for dispersion (e.g., Madsen et al., 1991; Baba et al., 2015). Accounting for dispersion effects can be important if the resulting series of excited oceanic waves locally interfere constructively and amplify, which has been observed in tsunami scenarios of the South China Sea (Ren et al., 2015) and outer-rise normal faults (Baba et al., 2021). Here, we do not detect significant differences in wave height or tsunami arrival times compared to our one-way linked scenarios, despite dispersion effects. Likely reasons include the relatively shallow water depth and the close proximity of the HFFZ to the coast preventing interferences.

| Source width $\sigma_r$ (m) | Source duration $\sigma_t$ (s) | Instantaneous source $\sqrt{gH} \cdot \sigma_t/\sigma_r \ll 1$ | Negligible acoustic wave excitation $H/(c_0 \cdot \sigma_t) \ll 1$ | Shallow water limit $H/\sigma_r \ll 1$ |
|---|---|---|---|---|
| 100,000 | 30 | Justified | Justified | Justified |

**Table 3.** Non-dimensional parameters for the justification of modeling assumptions as introduced by Abrahams et al. (2023). The parameter $H$ is the average water depth ($\sim$200 m), $c_0$ is the acoustic wave speed of 1500 m s$^{-1}$, and $g$ is the gravitational acceleration.

## 5 Conclusions

We present a suite of realistic earthquake-tsunami scenarios for North Iceland, comparing one-way linked and 3D fully-coupled modeling techniques. Both approaches agree in the resulting tsunamis from strike-slip earthquake dynamic rupture scenarios on the Húsavík-Flatey Fault Zone. We investigate two distinct fault system geometries – a simpler fault geometry with three fault segments and a highly complex fault system composed of 55 fault segments – to represent the 100 km long Húsavík-Flatey Fault Zone striking from onshore to offshore. Our study showcases how dynamic earthquake source mechanisms, including dynamic

rupture rake rotation near the surface (of $\pm20°$) combined with shallow fault slip, cause coseismic vertical displacement in the order of $\pm1$ m and the generation of high-amplitude acoustic waves without strong bathymetric slopes. We find that our earthquake-tsunami scenarios on a less segmented fault system, in particular with a hypocenter in the East near the town of Húsavík, generate the largest wave heights of $\sim$0.9 m near the local community Ólafsfjörður. Húsavík is the only town that is more affected by a scenario with a hypocenter in the West of the HFFZ, causing a maximum tsunami crest-to-trough difference

of $\sim$0.4 m. None of our scenarios regardless of the source complexity, endanger the town Akureyri, which is shielded by its narrow Eyjafjörður Fjord from the coseismically sourced tsunami. 3D fully-coupled scenarios include source dynamics, seismic, acoustic, and tsunami waves and result in complexities not present in our one-way linked simulations. We observe the excitation of tsunami normal dispersion and unexpectedly large acoustic waves, which may serve as a rapid indicator of surface-breaking dynamic rupture. Our findings highlight the importance of considering tsunamigenic strike-slip earthquakes

in tsunami hazard assessment. Accounting for the dynamics of earthquake source effects and fully-coupled tsunami generation may be useful to enhance tsunami hazard assessment and facilitate improvements to early warning systems.

*Code and data availability.* SeisSol is available from GitHub (https://github.com/SeisSol), sam(oa)$^2$-flash from GitLab (https://gitlab.lrz.de/samoa/samoa). The input files are hosted on Zenodo under https://zenodo.org/record/8360914.

*Video supplement.* Supplementary videos showing the propagation of the rupture front together with the seismic wavefield spreading across
the surface are available (https://zenodo.org/record/8360914). Also included are movies for the tsunami propagation for scenario Simple-East based on both the one-way linked and fully-coupled method.

## Appendix A: Earthquake dynamic rupture

a)  b)

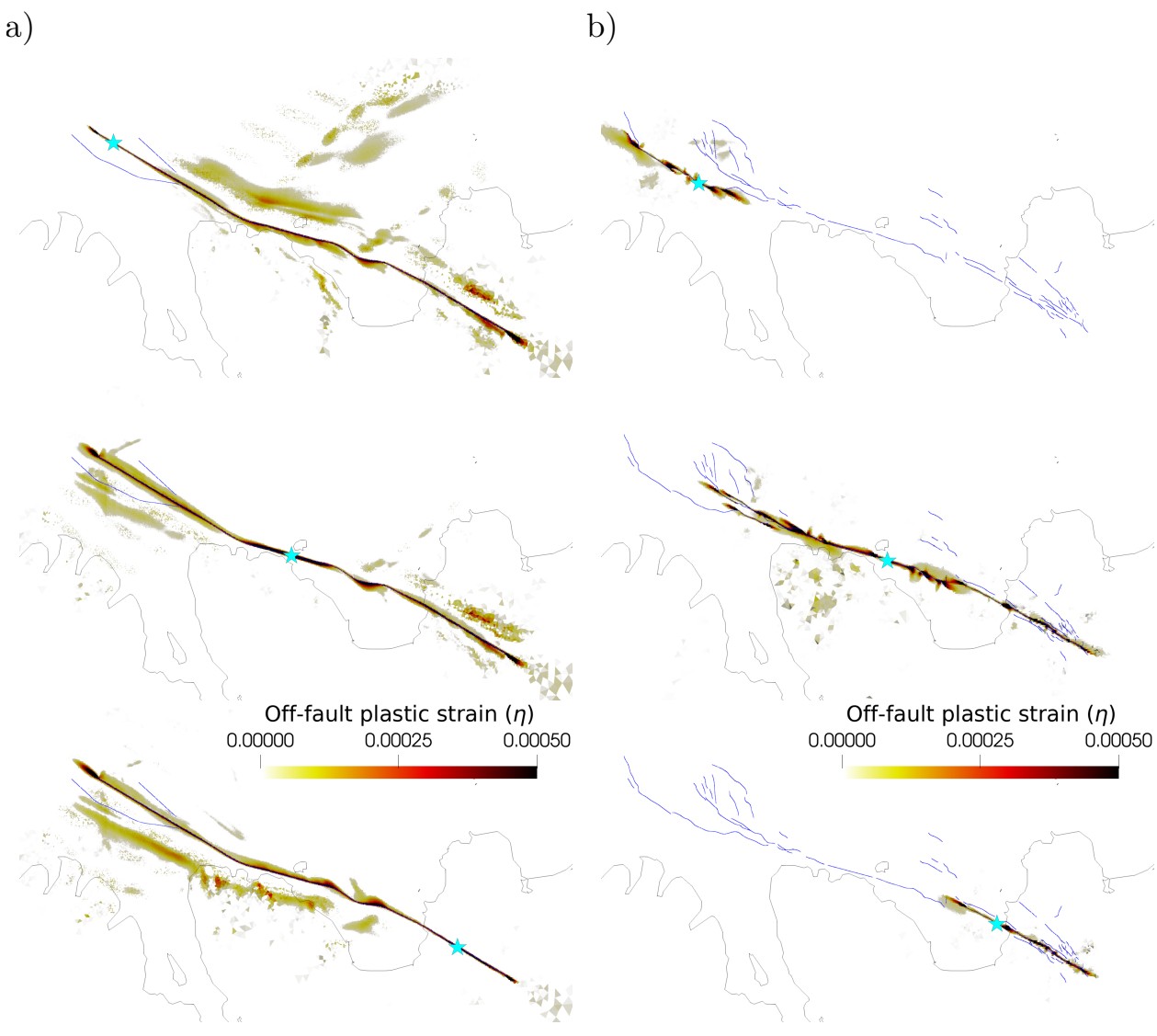

**Figure A1.** Accumulation of off-fault plastic strain ($\eta$) on the free surface for the dynamic rupture simulations on a) the simple fault geometry and b) the complex fault geometry. Cyan stars mark the epicenter locations. The scalar quantity $\eta$ is calculated from the plastic strain rate tensor $\dot{\epsilon}^p$ following Ma (2008) and Wollherr et al. (2018), i.e., at the end time $t$ of the simulation as $\eta(t) = \int_0^t d\eta = \int_0^t \sqrt{\frac{1}{2}\dot{\epsilon}_{ij}^p \dot{\epsilon}_{ij}^p}$ with $\dot{\epsilon}_{ij}$ being the plastic strain increment at one time step.

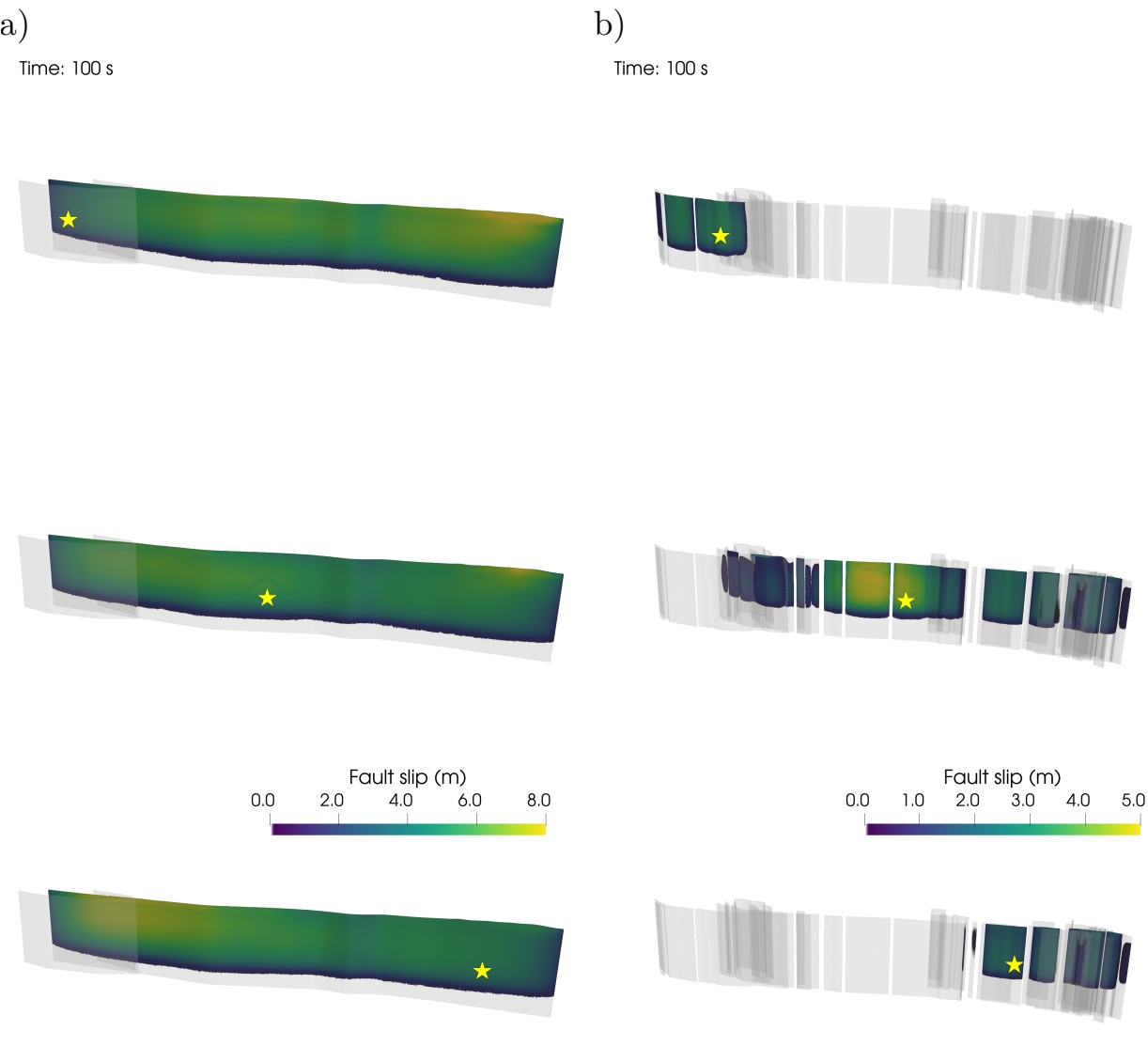

**Figure A2.** Accumulated fault slip of the earthquake dynamic rupture simulations on a) the simple fault geometry and b) the complex fault geometry. Yellow stars mark the hypocenter locations. Note the adjusted scale for fault slip in b) to better perceive differences among the three complex scenarios.

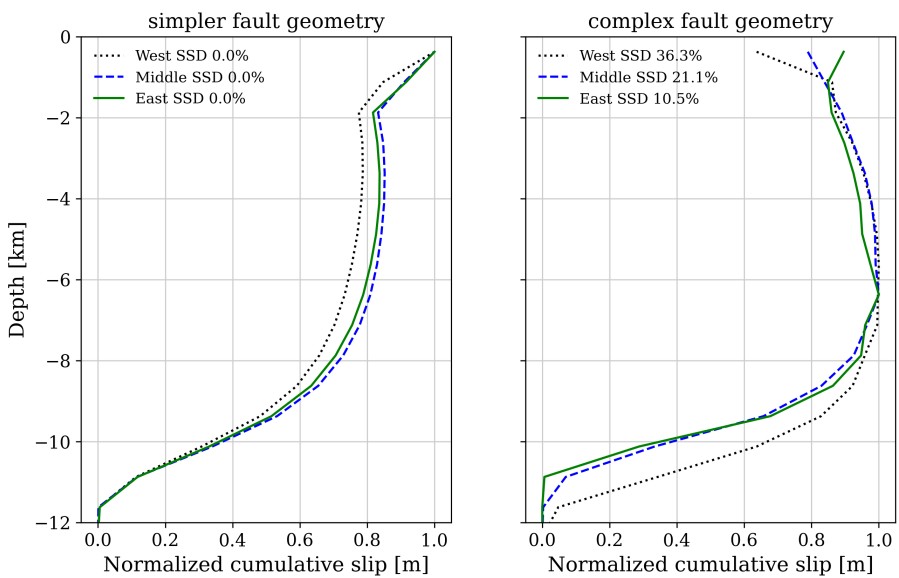

**Figure A3.** Normalized cumulative slip with depth for all six earthquake dynamic ruptures. The amount of shallow slip deficit (SSD) is indicated at the top left for each model on the respective fault geometry. 0 % of SSD represents no near-surface reduction of fault slip, while a higher percentage indicates that coseismic slip in the uppermost crust is less than slip occurring at average depths of the seismogenic layer (i.e., 4 – 6 km, Fialko et al. (2005)). The scenarios on the simpler fault geometry exhibit no SSD with large shallow fault slip, while SSD up to 36.3 % can be observed for dynamic rupture model Complex-West.

## Appendix B:  Tsunami

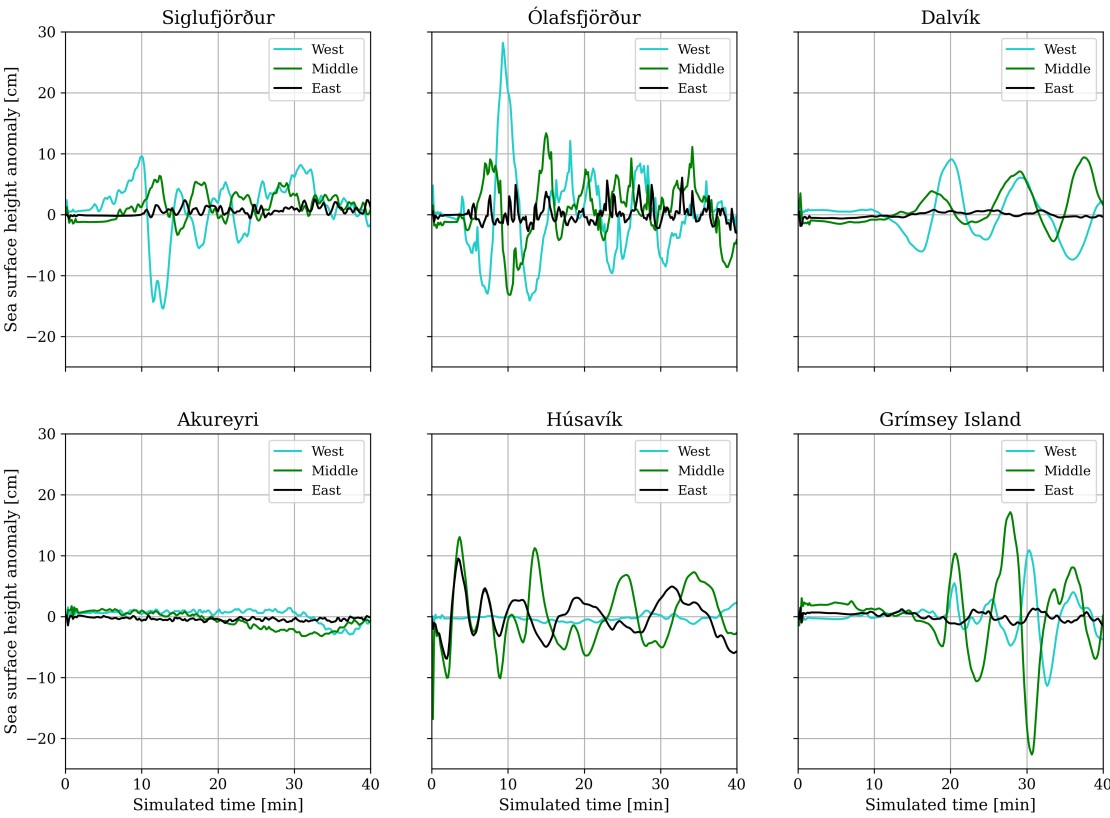

**Figure B1.** Sea surface height anomaly (ssha [cm]) vs simulation time (40 min) for the three one-way-linked scenarios sourced by dynamic rupture simulations on the complex fault geometry recorded at six synthetic tide gauge stations close to the towns Siglufjörður, Ólafsfjörður, Dalvík, Akureyri, Húsavík (west to east on the mainland) and Grímsey Island.

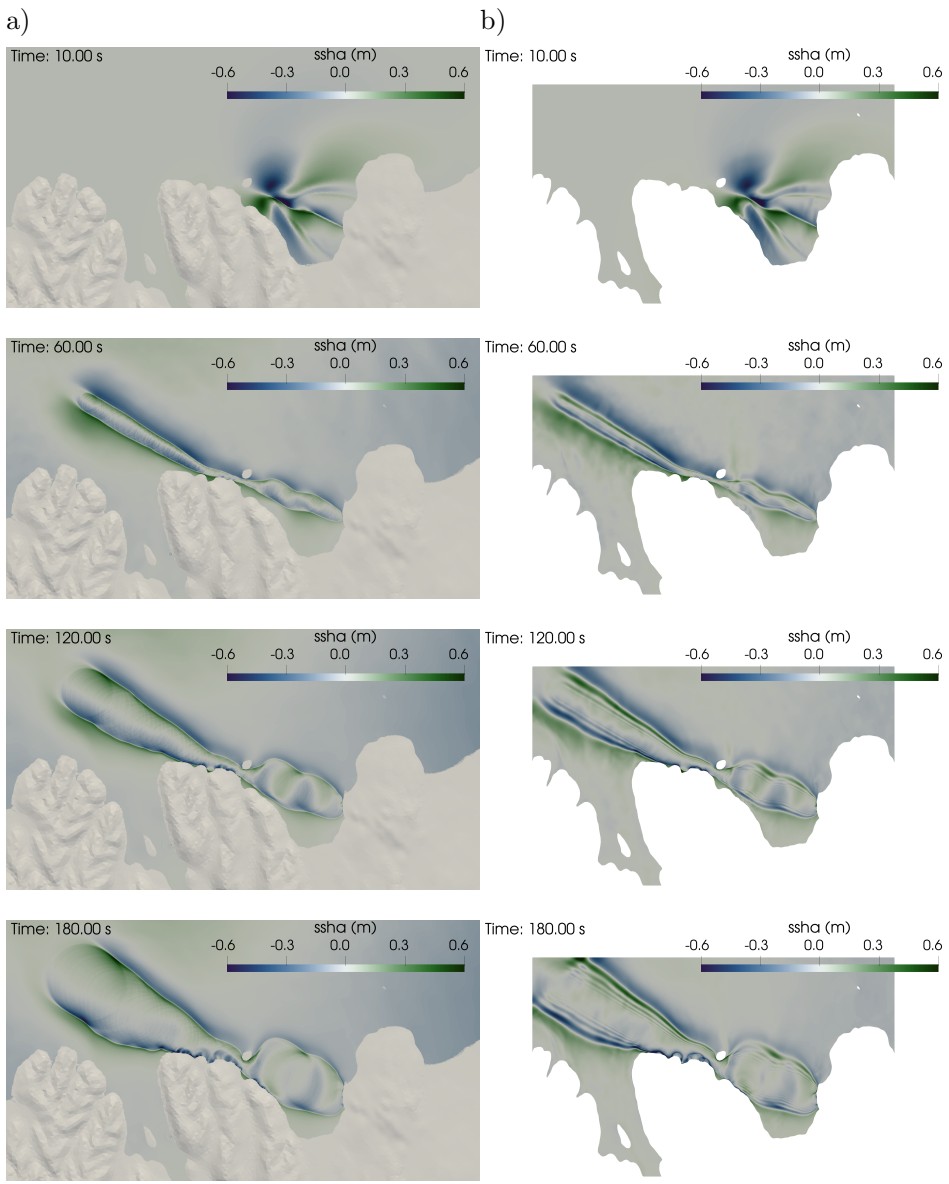

**Figure B2.** Waveform comparison for scenario Simple-East. a) One-way linked simulation. b) Fully-coupled model. Snapshots at 10 s, 1 min, 2 min and 3 min.

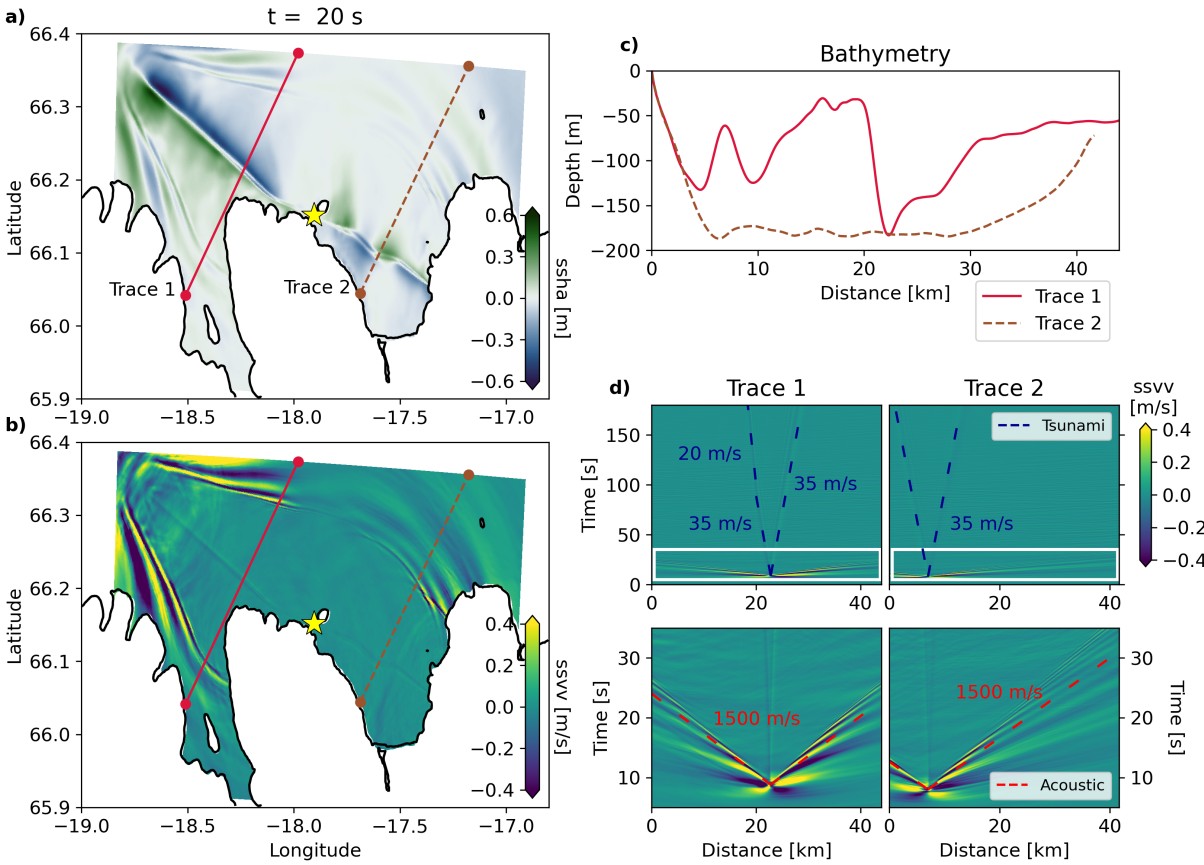

**Figure B3.** 3D fully-coupled earthquake-tsunami scenario Simple-Middle, with dynamic rupture on the simple fault geometry and a hypocenter in the Middle (yellow star). Snapshots at $t = 20\ s$ of a) the sea surface height anomalies (ssha) and b) sea surface vertical velocity (ssvv). c) Corresponding bathymetry profiles along the two selected cross-sections stretching from the shoreline (0 km) towards the open ocean. d) Space-time evolution of ssvv along the two cross-sections for the full duration of the fully-coupled simulations (upper row, highlighting the tsunami and the superposition of near-field displacements, seismic and acoustic waves). The white box indicates the zoom on the tsunami generation (lower row, highlighting the fast propagating acoustic waves).

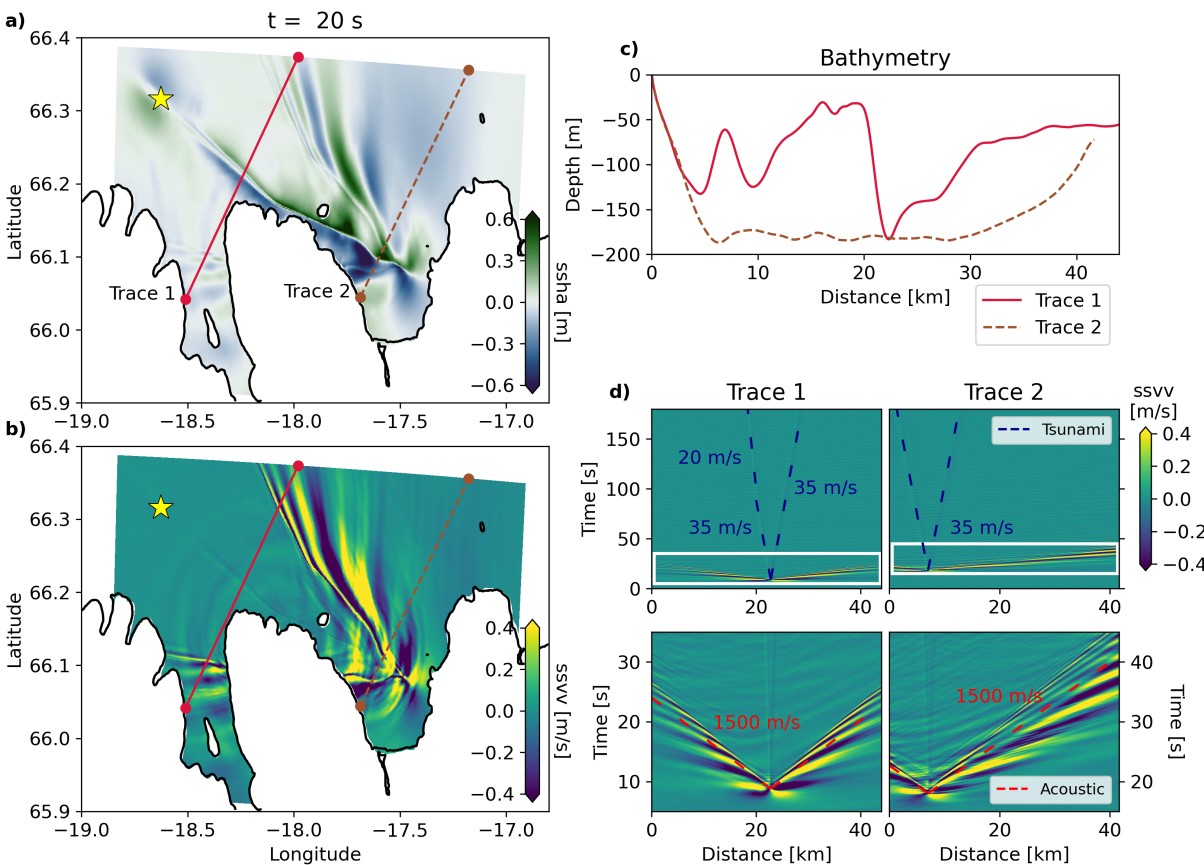

**Figure B4.** 3D fully-coupled earthquake-tsunami scenario Simple-West, with dynamic rupture on the simple fault geometry and a hypocenter in the West (yellow star). Snapshots at $t = 20\ s$ of a) the sea surface height anomalies (ssha) and b) sea surface vertical velocity (ssvv). c) Corresponding bathymetry profiles along the two selected cross-sections stretching from the shoreline (0 km) towards the open ocean. d) Space-time evolution of ssvv along the two cross-sections for the full duration of the fully-coupled simulations (upper row, highlighting the tsunami and the superposition of near-field displacements, seismic and acoustic waves). The white box indicates the zoom on the tsunami generation (lower row, highlighting the fast propagating acoustic waves).

*Author contributions.* FK performed the formal analysis and used the dynamic rupture (DR) models provided by BL for earthquake-tsunami modeling. FK proceeded with the visualization and writing of the original draft. A-AG supervised the project and gave substantial feedback for conceptualizing the research goals. SW and TU helped in the tsunami model setup and provided their expertise in post-processing earthquake-tsunami simulations. CA provided the dataset of relocated seismicity up to 2019, the 3D velocity model for TFZ, and the simplified fault geometry on the basis of her analyses. Both A-AG and BH acquired financial support. BH further supervised the work done at IMO. A-AG, BL, SW, TU and BH all provided comments on the manuscript and helped to review & edit the original draft. Each author contributed to the article and approved the submitted version.

*Competing interests.* The authors declare that they have no conflict of interest.

*Acknowledgements.* This work received funding from the European Union's Horizon 2020 research and innovation programme (TEAR ERC Starting grant agreement No. 852992) and Horizon Europe (ChEESE-2P grant No. 101093038, DT-GEO grant No. 101058129 and Geo-INQUIRE grant No. 101058518). We acknowledge additional funding from the National Science Foundation (grant No. EAR-2121666) and the National Aeronautics and Space Administration (80NSSC20K0495). We thank Lukas Krenz, Lauren Abrahams and Eric Dunham for invaluable discussions and their contributions to the 3D fully-coupled earthquake-tsunami modeling capabilities in SeisSol. We thank all participants of the NorthQuake 2022 workshop (https://hac.is/en/radstefnur/northquake-2022) for insightful presentations and discussions. The computing infrastructure available at the Department of Earth and Environmental Sciences at LMU, Geophysics (Oeser et al., 2006), and at the Leibniz Supercomputing Center (LRZ, projects no. pn68fi, pr63qo, and pn49ha) are highly appreciated. The Generic Mapping Tools (Wessel et al., 2019) and its Python interface PyGMT (Uieda et al., 2022) were used to generate some of the figures. We used two kinds of gridded bathymetric data (Ryan et al., 2009; GEBCO Compilation Group, 2020) and scientific colour maps (Crameri, 2018; Crameri et al., 2020).

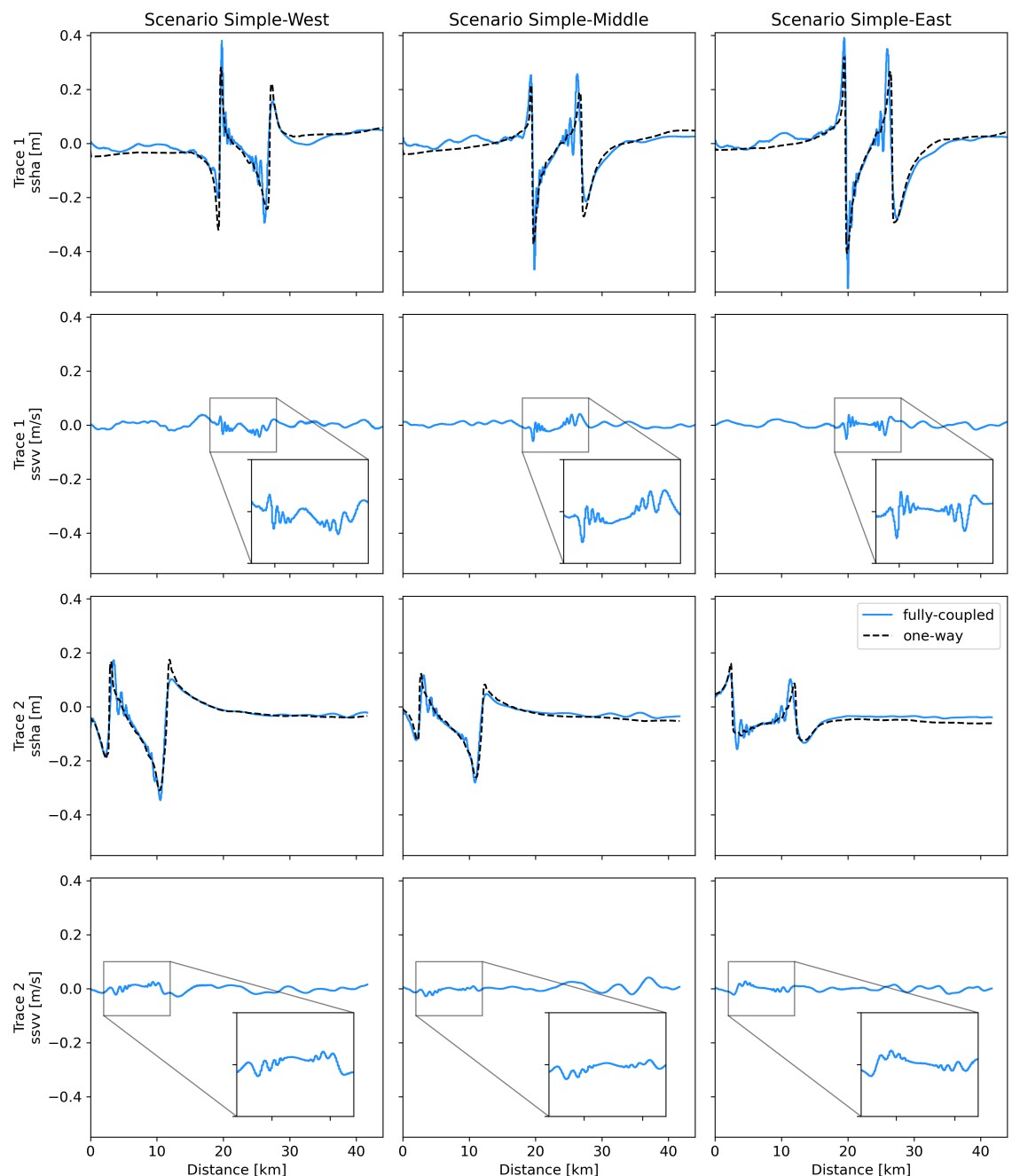

**Figure B5.** Sea surface height anomaly (ssha [m]) along previous two cross-sections (i.e., Figs. 10, B3, B4) at $t = 2$ min for the fully-coupled (solid blue line) and one-way linked (dashed black line) scenarios on the simpler fault geometry in the first and third row. Sea surface vertical velocity (ssvv [m s$^{-1}$]) for all fully-coupled simulations highlighting tsunami normal dispersion in the second and bottom row. The shoreline is located at 0 km.

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
