# Peer review of "Linked and fully-coupled 3D earthquake dynamic rupture and tsunami modeling for the Húsavík-Flatey Fault Zone in North Iceland"

_EGUsphere, 2023_

## Referee Comment (RC1)

**Review of the manuscript of Kutschera et al., "Linked and fully-coupled 3D earthquake dynamic rupture and tsunami modeling for the Húsavík-Flatey Fault Zone in North Iceland"**

**Summary**

Kutschera et al. investigate the tsunamigenic potential partially offshore transform fault in North Iceland by and presenting six dynamic rupture model scenarios based on two fault geometries (simple/complex) and three rupture nucleation points (west/center/east). The physics-based models are constraint by empirical friction laws, pre-stress conditions and fault strengths. The rupture simulations are once one-way linked to the tsunami models, once fully-coupled with them.

The authors find that simpler fault geometries cause significantly larger ruptures, higher displacement and higher tsunami waves. They show that the fully-coupled model scenarios create more detailed wave propagation fields that also include seismic and acoustic waves. Since the latter are very fast, they could potentially be implemented in early warning systems, the authors conclude.

The manuscript presents state-of-the-art tsunami-modeling that provides realistic upper estimates of tsunami heights for nearby coastal towns, a relevant information for the North Icelandic municipalities. The text clearly reflects that the author are experienced experts in dynamic rupture modeling and the tectonic context. The model set up seems robust to me, thus my main comments (see below) mostly address the discussion and interpretation of the model results. I also suggest a few modifications of text and figures to improve the overall reading experience, particularly for modeling laypersons. I hope the authors will find them useful.

**Main comments**

**Large slip caused by simple fault geometry**

A)  You state that the fault accommodates 6-9 mm/yr of slip since the last earthquake ~150 years ago in 1872. The estimated slip deficit would be approximately ~1 m today. This stands in large contrast to your observed coseismic slip of up to 8 m, making me doubting your pre-event model conditions. Can you explain the reason for this large value (e.g. overshooting, low friction parameters, etc.)? Is such high slip even realistic on a 100 km long fault? Could you somehow dampen this slip using other pre-event model conditions?

B)  I am impressed by how the complex fault geometry hinders large earthquakes. Given that the *real* fault geometry is even more complex, I would expect that M7 earthquakes are not even possible, but of course they are! Can you explain this apparent contradiction further? Or is it irrelevant, because the predicted wave heights far from the source are of a similar size (factor ~2)?

**Vertical coseismic displacement**     Your models predict near-surface, vertical coseismic displacements at the rupture end, which is what is also often observed strike-slip earthquakes (cf. Interferograms and geodetic slip models of Ridegcrest, Maduo, Muji, Sarez). It might be not the main scope of this earthquake, but I find this interesting enough to briefly be discussed: Why to the

segments rotate? Is it consistent, does always the same side of a right-lateral fault exhibits up motion, or what does it depend on (nucleation, dip etc.)?

**Figures** Revise the number of figures to improve the ratio between visual presentation and text description. For example, Figure 10-12 and B3-B4 use a lot of space, but are hardly discussed. Maybe there is a better way to present (parts of) them and make your point? Figure 13 uses a full page, only to state that coupled models provide more details.
For a better recognition, please use the same symbols in all figures, e.g. mark the nucleation point in all relevant figures similarly. To better follow your argumentation, please refer more often to the figure under discussion, maybe also mark features under discussion with (more) arrows and labels.

**Adjective-chains** Some sentences contain extremely long chains of adjectives, making the sentence difficult to read, particularly, when not an expert in the domain. Some verbs, "of"s, or commas might help to improve the readability. Examples are "both one-way linked 3D earthquake dynamic rupture and shallow-water equations tsunami simulations" (L70), "time-dependent one-way linked and 3D fully-coupled earthquake-tsunami modeling" (L6), "recently developed physics-based dynamic rupture models" (L73), L79/80, L94/95, "the dynamic rupture initial conditions" (L100), "complex off-shore fault system structure" (L109), "showcased complex fault geometry scenarios" (146), "community dynamic rupture benchmarks" (169).

**Detailed comments and suggestions to improve the reading experience**

L35: northern

L37ff (and also abstract): While introducing the tectonic setting of the TFZ and HFF, make sure that geographic features mentioned in the text also appear in figure labels to make sure the readership can follow your descriptions, for example, by referring already here to Figure 1. I would also simplify the descriptions, for example, Nordurland eystra (L48/237) sounds like national description of the region and I would not use it, Kolbeinsey Ridge (KR and NVZ ares never used anymore), Olafsfjördur should be shown in a map. Please revise the text regarding this. For example, the inset in Figure 1a could be increased and the three lineaments could be schematically marked.

L42/3: plate

L45: This sentence sounds somewhat strange.

L52: Please quantify the term "last glacial maximum"

L59: unit missing (years?). Introduce the abbreviation GNSS and GPS. The term GNSS is more general as it covers all positioning systems.

L82ff: I would not abbreviate the word "Section"

L104: I suggest to delete "of active submarine fault systems" and end the sentence with "a submarine earthquake rupture".

L125: please define *Shmax* in text form not only in line 127 but here.

L155: No brackets needed around Abril et al.

Table 1: Is there a unit missing for Shmax? I would write "150/155" as entry with the asterisk.

L165-167: Repetition of above.

L171: replace "&" with "and"

L173: Please explain "sam(oa)2-flash" already here, not in the consecutive sentence, e.g. by moving "dynamically adaptive, parallel software" one line up. What is the meaning of "parallel software"? That it is able to do parallel computing?

L177: Appendix B is rather short and would add maybe two additional text lines and one equation to the main text. It could be skipped at all (L175-178 already explain the key facts and that its contribution is small) or embedded in the text directly.

L182: delete "so-called"

L188: delete "medium"

L197: Rephrase and start with "The on-fault resolution of 200 m is gradually..."

L210: "A comparison of…" could be deleted, and the figure reference can be moved to the beginning of L212.

L212/Table 27/main comment: 7.9 m slip sounds like too much compared to the estimated slip deficit, if the fault should accommodate 7 mm/yr or so, no, so maybe 1m of slip at most? Can you explain the reason for these high slip values of all models?

L215: Can you explain a bit more? Is it relative to the max. slip observed further down? Is it calculated by the ratio between the area left and right of the curve plotted in Figure A3?

L216/217: Is there a correlation between the spatial pattern observed in off-fault strain (Figure A1) and the SSD, or depth of max. slip? I notice that off-fault strain occurs at constant particular distance to the fault surface trace.

L217: "(Fig. 3b)" (could you zoom into Fig. 3b somewhat to show more details of the flower structure?

L220: delete "are non-negligible" and refer already in this sentence to Figure 5.

L224: refer to Figure 1 here.

L224: Delete "However,"

L227: there is a unneeded bracket left

L228/229: It would help to have the geographic markers under discussion also provided in Figure 5.

L231/232: Given the complexity of the rest of the paper, this explanation of the rake seems to be unnecessary for the targeted audience. You could start the next sentence instead with "Earthquakes on a vertically dipping, right-lateral fault system, such as the HFFS, predominantly exhibit slip rake angles of 180 deg. However…" Also, the sketch in Figure 5 should be updated to show a vertical fault.

L233-235/L329-332/main comment: This agrees very well with InSAR offset observations (unfortunately rarely shown, for example, Ridgecrest, He et al., 2022, Figure 2k, https://doi.org/10.1029/2021JB022779), or slip models that do not constrain the rake angle (e.g. Sarez earthquake, Metzger et al., 2017, Figure S8, http://dx.doi.org/10.1002/2017TC004581, Muji earthquake, Feng et al., 2017, Figure 6, https://doi.org/10.1785/0220170019, or Maduo earthquake,

Xiao et al., 2022, https://doi.org/10.1007/s12583-022-1637-x, Hong et al., 2022, Figure 6, https://doi.org/10.1785/0120210250)

L236: Add reference to Figure 1.

L239: Delete "3D"

L241: I do not understand the benefit of the auxiliary wording of "synthetic tide gauge stations placed near". Could it not be deleted (also later) and speak of "predicted sea surface height anomaly at the coastal towns of" instead?

L282: What is SuperMUC-NG? Maybe just say "on our computational infrastructure" or so.

L285ff: Technically, the order of mentioning a), b), c), d) should be alphabetically, so exchange the labels b and c.

L294: Revise this sentence, there is a verb missing, I think.

L295: "cross-section to in Figs."

L3010304: It would be instructive to mark these waveforms directly in subfigures d) and refer to the them here (instead of in the appendix).

L294-L304: Given the length of the text discussing Figures 10-12 and also the information content of the figures (to the untrained eye, at least, subfigures b) and d) look very similar in all cases), you might consider moving two of them to the supporting material?

L300: "Figs. 10b, 11b, 12b, cross-section 1)"

L305-314: You show a panel of 12 figures to only conclude that the waveforms contain more short-period signal. If there is not more to gain from this plot, I would only show two subplots as pars-pro-toto.

L316: "Submarine" instead of "Submerged"?

L317: Rearrange the Sulawesi sentence: "Linked and fully-coupled...modeling of the 2018 … earthquake imply that coseismic…"

L320: No need to repeat what you did, just start with "Our simulations of six rupture scenarios show that the Husavik-Flatey…

L325: Repetition of what you just said in L321, also better say "host a local" instead of "source a localized".

L332: Replace "In difference" with "Unlike" or "Oppositely to"

L334: Refer to Figure A1 here.

L340: Is there a "so" missing between the wave speed and "we can calculate"?

L340-350: I do not see the benefit of Table 3 as everything is already mentioned here in the text.

L348: Refer to the respective subfigures in Figure 7 here

L353ff: I do not understand: Do you imply that after submarine earthquakes at shallow depths we should be more concerned about acoustic waves rather than tsunami waves?

L394: See my comment for L233-235 resp. L329-332: Yes, the 8 m of slip seems to be too large, but vertical offsets also exist at the end of strike-slip ruptures.

L411: Delete "on-average" or replace it with "relatively"

L415: I cannot guess what kind a "sizeable tsunami" is, maybe you find another term?

L416: Be more specific what you mean by "wto distinct fault system", e.g. by saying, "a simpler (X segments) and a more complex (55 segments) fault geometry".

L419: To me 8 m of fault slip seems too unrealistic to point out in the conclusion. It would implicate 800 years of complete slip deficit, no?

**Figure/Caption comments**

**Figure 1**      I do not understand the meaning of the percentage of SSD, please explain better, also the meaning of "shallow" in km.

**Figure 5**      The font is rather small, can you increase it? Also increase the size of the hypocentral star. Use the same symbol (size and color) in Figure 5, 6, 7 10-12.

**Figure 7**      The figure readability would improve if towns are directly labeled/numbered in the subfigure, instead of describing their location in the caption. Add the hypocenter, similar to Figure 5.

**Figure 6**      The sketch of the rakes should be adapted to show a vertical fault (incl. slip direction), not a dipping one.

**Figure 8/B1**   Remove the overarching titles

**Figure 9**      Increase the font label size. Could you rework the choice of colors (or background color of the legend), such that the individual colors are readable and distinguishable?

**Figure 10-12**  It would be instructive to non-experts to label the different wave fronts in subfigure d). Flip label b) and c). It took me until reading the conclusions that acoustic waves are "booms" (right?) and not real waves, because in your Figures the color-map is "sea surface vertical velocity [ssvv]"? Please clarify.

**Figure B2**     The color map is flipped between a) and b).

**Supporting Text**

When I tested the rupture wavefield videos, some of the started only at the second half of the the video time. Please check.

---

## Referee Comment (RC2)

Comment to the authors

Authors: Kutschera *et al.*

Manuscript #: egusphere-2023-1262

The manuscript submitted by Dr. Kutschera and the colleagues investigates the potential tsunami hazard due to strike-slip earthquakes in the Húsavík-Flatey Fault Zone in North Iceland. Their proposal of tsunami scenarios based on two different tsunami modeling approaches, one-way linked tsunami model and the 3D fully-coupled (earthquake–tsunami) model, are informative to understand how similar and different these results are. The authors also try the simulation with differently assumed simple and complex fault geometries, which yielded different results. They suggest the importance of consideration of detailed fault geometry for tsunami scenarios and that strong fast-arriving acoustic waves may be used to notice the tsunami potential in advance.

The manuscript is very well-prepared and deliver useful information in an appropriate way. I here list up numbers of comments, but these are just for more concise and fair descriptions. I hope these will be helpful for the authors to improve the manuscript. After this minor revision, I can recommend the publication of this article.

[Moderate comments]

1. When the authors discuss on the 2018 Palu tsunami, only the strike-slip earthquake itself is attributed to the huge tsunami, but there is still possibility of the landslide origins. It would be fairer to mention previous studies which proposed the landslide tsunami origins somewhere in introduction or discussion (for example, around Lines 33–33, 318–320, or 396–397).

2. When the authors compare the two tsunami modeling approach, a one-way linked or 3D fully-coupled ones, it sounds that the one-way linked approach is somewhat underrated. For example, the dispersion property can be included into the one-way linked model by using an appropriate dispersive tsunami model, but in the present manuscript, it sounds that the dispersion can incorporated only by the 3D fully-coupled model (I know that this is not the authors' intention). The non-dispersion is due to the shallow-water wave model. Please keep fairness to compare the one-way linked and the 3D fully-coupled model.

3. I think both two tsunami models you use here do not consider tsunami runup or inundation to coasts.

Is my understanding correct? The classical one-way linked model can easily incorporate the runup/inundation, as done many models such as Baba et al. (2015). How about the 3D fully-coupled model? I guess it requires sophisticated modeling of a moving land/ocean boundary. Please discuss on this somewhere (maybe in Discussion), because the runup/inundation can be also important aspects for the tsunami hazard assessment. (I understand that this would be out of your scope, and comparison at tide gauges without inundation is useful enough. I comment on this because it would be very informative for tsunami researchers.)

ref) Baba, T., Takahashi, N., Kaneda, Y., Ando, K., Matsuoka, D., & Kato, T. (2015). Parallel Implementation of Dispersive Tsunami Wave Modeling with a Nesting Algorithm for the 2011 Tohoku Tsunami. *Pure and Applied Geophysics*, *172*(12), 3455–3472. https://doi.org/10.1007/s00024-015-1049-2

[Minor comments]
Lines 37–47: The tectonic setting is better to described as a figure. it is difficult to understand the tectonics only from the text. I recommend that the authors put a figure with a broader area including the main tectonic setting, with notations of TFZ, NVZ, and the fault motion direction with average velocity of the plate boundary. These will be helpful for readers who are not familiar with the region.

Line 50: *The strongest historically recorded M 7 event in 1755 caused extensive damage and may have generated a series of oceanic waves (i.e., a tsunami) that hit the coastline (Stefansson et al., 2008; Þorgeirsson, 2011; Ruiz-Angulo et al., 2019)*
Would you please explain this part more clearly? Did the referenced studies show any evidence of tsunamis (such as historically documented record or tsunami sediment), or just speculated based on the large fault offset found on seafloor?

Line 59: *calculate a recurrence interval of 32 $\pm$24*
What are the unit? Year?

Figure 1. See my comments on Lines 37–47. This figure needs to include the information of more broad tectonic setting.

Lines 74–81:

The detailed information, on such as SeisSol, GMM, or parameter differences, does not fit here in Introduction. I recommend that the authors introduce what they will show just more briefly here and move the details to in Sect 2 or 3.1. It will guide readers more smoothly to the details.

Line 90: *We model one-way linked (cf. Sect. 2.4) and fully-coupled (cf. Sect. 2.5) tsunami scenarios (Abrahams et al., 2023)*

"One-way linked and fully-coupled tsunami scenarios" sounds strange to me. If we say "tsunami scenarios", it would be possible tsunami *cases* by different source models, as you say in the following paragraph like "We use six earthquake scenarios." Please consider replacing the "tsunami scenarios" by "tsunami models", "tsunami approaches" or "tsunami computation methods".

Line 131–132: *It also agrees with assuming a 90°dipping fault system.*

How about "It also agrees with our assumption of a 90° dipping fault system"?

Line 153: *6 to 10 km is the inferred locking depth for the HFFZ (Metzger and Jónsson, 2014).*

Based on what is the locking depth estimated? Please clalify.

Figures 3:

Three panels in the figure are shown too small. Please consider showing the panels vertically in a column., or "a) on the top row, and b) and c) on the bottom row.

Line 165:

Here you explain SeisSol. You may remove the description of SeisSol in Introduction and explain the details here.

Line 233: *However, we observe dynamic rake rotation (°æ20°) near the surface during the rupture (Fig. 6).*

It is interesting that there happens a rake orientation. Could you please explain how this happens?

Figure 6, A1, A2:

The panels are very small. Please enlarge the figures, by showing them vertically, three panels of a) in 1st column, and those of b) in the 2nd column. The same problem happens in Fig. A1 and A2. Please consider revising them, as well.

Figure 7:

Why are the Simple cases at t=120 and 600s show very broad ssha ( faint green and blue)? This looks weird, and is very contrasted to the complex cases. Please check they are modelled correctly and, if true, explain what they represent.

Figure B2:

The color bar is flipped. Please revise it.

Lines 305–310 (Related to my moderate comment 2):

As you know, the non-dispersive character is not because of a one-way linked tsunami model, but just because of the "linear long-wave model" employed. It would be fair to note that the dispersion can be simulated in a one-way approach as well.

Lines 333–335:

*In difference to a proposed dominance of off-fault deformation in strike-slip tsunami generation in Palu Bay (Ma, 2022), the effect of offfault plasticity is likely small in our simulations. We find that off-fault deformation contributes only about ~3 % of the total seismic moment.*

Could you please mention possible reasons of the difference? Is it because of fault geometry, rupture model, or other parameters?

---

## Author Comment (AC1)

**Rebuttal letter Solid Earth egusphere-2023-1262**

Linked and fully-coupled 3D earthquake dynamic rupture and tsunami modeling for the Húsavík-Flatey Fault Zone in North Iceland

We thank the Editor and two anonymous Referees for their positive evaluation and constructive comments. We carefully addressed all comments by adding clarifications or corrections to the manuscript and by improving Figs. 1, 2, 3, 5, 6, 7, 8, 9, 10, former Fig. 11 (now B3), former Fig. 12 (now B4), former Fig. 13 (now split into the new Fig. 12 and Fig. B5) and Figs. A1, A2, B1, B2. We provide point-to-point responses in the following.

Additionally, based on a comment by the editorial support team, we used the Coblis - Color Blindness Simulator and revised the color schemes of Figs. 1, 2 as well as the linestyle of the traces in Figs. 10, B3, B4 (former Figs. 10, 11, 12).

Our responses are given in **green**, with changes to the manuscript in *italic*. We include the comments of the **Editor received via email in dark red**, of **referee #1 in black** and those of **referee #2 in blue**.

We would like to thank you again for the positive feedback and thoughtful reviews. We hope the revised manuscript will be well received.

Sincerely,
Fabian Kutschera, on behalf of all co-authors

**Editor**

We have now received the second review for your manuscript and I have, therefore, been able to terminate the current online discussion phase for the article. The third reviewer was solicited by the system in error, and has been cancelled. You can now complete your replies to comments from reviewers. In particular, please address in detail the main comments from reviewer 1 regarding the large slip caused by simple fault geometries.

Thank you very much for the update and taking care of handling our manuscript. We address both reviews and your comment in the following. Please see our responses to reviewer 1 regarding the large slip caused by dynamic ruptures on the simpler fault geometry in R1.1 and RM1.56.

**RC1 - Referee #1**

Our responses are given in **green**, with changes to the manuscript in *italic*. We include the comments of **referee #1 in black**.

**Summary**

Kutschera et al. investigate the tsunamigenic potential partially offshore transform fault in North Iceland by and presenting six dynamic rupture model scenarios based on two fault geometries (simple/complex) and three rupture nucleation points (west/center/east). The physics-based models are constraint by empirical friction laws, pre-stress conditions and fault strengths. The rupture simulations are once one-way linked to the tsunami models, once fully-coupled with them.

The authors find that simpler fault geometries cause significantly larger ruptures, higher displacement and higher tsunami waves. They show that the fully-coupled model scenarios create more detailed wave propagation fields that also include seismic and acoustic waves. Since the latter are very fast, they could potentially be implemented in early warning systems, the authors conclude.

The manuscript presents state-of-the-art tsunami-modeling that provides realistic upper estimates of tsunami heights for nearby coastal towns, a relevant information for the North Icelandic municipalities. The text clearly reflects that the author are experienced experts in dynamic rupture modeling and the tectonic context. The model set up seems robust to me, thus my main comments (see below) mostly address the discussion and interpretation of the model results. I also suggest a few modifications of text and figures to improve the overall reading experience, particularly for modeling laypersons. I hope the authors will find them useful.

We appreciate the thoughtful review of our manuscript. In the following, we first address your main comments, and then respond to your detailed comments and suggestions.

**Main comments**

Our responses are given in **green**, with changes to the manuscript in *italic*. We include the comments of **referee #1 in black**.

**Large slip caused by simple fault geometry:**

A) You state that the fault accommodates 6-9 mm/yr of slip since the last earthquake ~150 years ago in 1872. The estimated slip deficit would be approximately ~1 m today. This stands in large contrast to your observed coseismic slip of up to 8 m, making me doubting your pre-event model conditions. Can you explain the reason for this large value (e.g. overshooting, low friction parameters, etc.)? Is such high slip even realistic on a 100 km long fault? Could you somehow dampen this slip using other pre-event model conditions?

**R1.1:** We understand your concern regarding the significant amount of slip observed in some of our simulations as compared to the estimated slip deficit.
Unfortunately,  the historical earthquake record provides us with a limited snapshot of the long-term seismic cycle of the fault system and slip deficit estimates rely on a number of assumptions. For example, while the last significant (M>6) historical earthquakes occurred roughly 150 years ago, it is not certain whether these events released all of the accumulated slip.

We are extending the 3D dynamic rupture models by Li et al. (2023) to tsunami modeling. In their Figure 11, reproduced below, they show that the synthetic ground motions produced by such dynamic rupture models are in good agreement with the latest regional empirical ground motion model (Kowsari et al., 2020). While the moment magnitudes of some of our earthquake scenarios are larger than previous estimates of the accumulated moment along the HFFZ (e.g, Mw 6.8±0.1, Metzger et al., 2011), the dynamic rupture earthquake scenarios have comparable moment magnitudes to historic events. The rupture scenarios are also consistent with the scaling relation of Mai and Beroza (2000). Li et al. (2023) show this in Figure S3 in Supporting Information S1, also reproduced below. The scaling relations of Mai and Beroza (2000) have recently been validated for the Southern Iceland Seismic Zone (SISZ) (Bayat et al., 2022). The SISZ is similar in its tectonic and seismic context to the Tjörnes Fracture Zone (TFZ) in Northern Iceland.

Our dynamic rupture scenarios are inherently complex, involving highly nonlinear interactions between frictional failure of complex fault systems, seismic wave propagation, a 3D subsurface model, topo-bathymetry, and fault-to-fault interaction. These dynamic factors can lead to large variability of co-seismic slip dynamics and lead to local variations in slip (e.g., potentially also overshooting) that may not be fully captured by a "simpler" slip deficit calculation based solely on the slip rate.

In addition to the peak slip, we now also report the average fault slip in L214ff and Table 2. Note that the peak slip is localized only near the surface, while the average fault slip is ~40 % smaller.

L214ff reads now:

*"Scenario Simple-East produces the largest maximum fault slip localized at the offshore section of the fault system (7.90 m) with an average fault slip of 4.93 m (Table 2). We consider those parts of the fault which experience at least 0.01 m coseismic slip for computing the average fault slip."*

| | simple fault geometry | | | complex fault geometry | | |
|---|---|---|---|---|---|---|
| Hypocenter | West | Middle | East | West | Middle | East |
| $M_W$ | 7.34 | 7.33 | 7.34 | 6.74 | 7.07 | 6.68 |
| Avg. fault slip [m] **New** | 5.03 | 4.80 | 4.93 | 2.14 | 1.97 | 1.51 |
| Max. fault slip [m] | 10.34 | 8.11 | 7.90 | 3.50 | 5.23 | 2.74 |
| Max. fault slip offshore [m] | 6.93 | 6.58 | 7.90 | 3.50 | 5.23 | 2.74 |
| Max. peak slip rate [m s$^{-1}$] | 15.05 | 14.93 | 15.14 | 10.44 | 11.59 | 8.66 |
| Max. peak slip rate offshore [m s$^{-1}$] | 13.53 | 12.58 | 15.14 | 10.44 | 11.59 | 8.62 |
| Max. seafloor uplift [m] (after Tanioka filter) | 0.75 | 1.05 | 0.95 | 0.56 | 0.44 | 0.23 |
| Max. seafloor subsidence [m] (after Tanioka filter) | -0.74 | -0.79 | -0.76 | -0.66 | -0.79 | -0.42 |

*"Table 2. Key results of our here considered six earthquake dynamic rupture scenarios. Note that we only report the maximum offshore coseismic vertical displacements (i.e., seafloor offsets) in the table because the onshore vertical displacements do not contribute to the tsunami generation."*

We updated the manuscript and added a section discussing the large fault slip (L430-439), which reads now:

*"We extend recent 3D dynamic rupture models by Li et al. (2023) to tsunami modeling. In these scenarios, relatively high peak fault slip localizes near the free surface while the average fault slip is overall ~40 % smaller than the peak fault slip (Table 2). Li et al. (2023) show that the synthetic ground motions produced by such dynamic rupture models are in good agreement with the latest regional empirical ground motion model (Kowsari et al., 2020). While the moment magnitudes of some of these models are larger than previous slip-deficit based estimates of the accumulated moment along the HFFZ (e.g, Mw 6.8±0.1, Metzger et al., 2011) they have comparable moment magnitudes to historic events. Slip-deficit based magnitude estimates are typically relying on several assumptions including in this case a complete stress relaxation of the 1872 Mw = 6.5 earthquakes and subsequent steady stress accumulation. The dynamic rupture scenarios are also consistent with the scaling relation of Mai and Beroza (2000), which have recently been validated for the Southern Iceland Seismic Zone (SISZ, Bayat et al., 2022). The SISZ is similar in its tectonic and seismic context to the TFZ in Northern Iceland."*

[Figure]

Figure 11 from Li et al. (2023) showing: "(a)–(e) Comparison of the synthetic ground motion from earthquake scenarios across Model-C and ground motion models in terms of spectral acceleration (SA[1.0 s], in g). See the caption of Figure 10 for further details. (e) Mean attenuation relationship for the four rupture scenarios across Model-C." Their Model-C corresponds to our simple fault geometry model.

[Figure]

Figure S3 from Li et al. (2023) showing: "(a) Moment magnitude vs effective rupture area for the earthquake scenarios using Models B and C (triangles) compared with scaling relationships of P. Mai & Beroza (2000). (b) Moment magnitude vs average fault slip and (c) moment magnitude vs effective rupture length."

B) I am impressed by how the complex fault geometry hinders large earthquakes. Given that the *real* fault geometry is even more complex, I would expect that M7 earthquakes are not even possible, but of course they are! Can you explain this apparent contradiction further? Or is it irrelevant, because the predicted wave heights far from the source are of a similar size (factor ~2)?

**R1.2:** The bilateral ("Middle") dynamic rupture scenario on the complex fault geometry produced a moment magnitude of 7.07. Large earthquakes are still dynamically possible even on complex fault geometries but under a narrower range of initial conditions (as discussed in Li et al. (2023)). During the earthquake dynamic rupture complex fault geometry may serve as a natural barrier preventing the rupture from jumping from one segment to another. We extend our discussion as follows:

L337-344 reads now:
*"Large-scale geometric fault complexity can act as an "earthquake gate" (Lozos, 2016; Duan et al., 2019; Liu et al., 2021). An earthquake gate is a mechanical barrier to earthquake rupture (Liu et al., 2022), which has the potential to alter the rupture extent. In the case of the simpler fault geometry, the restraining and releasing bend east of Flatey Island may be considered as such an earthquake gate since this smooth main fault bend does allow some ruptures to propagate across while terminating others depending on the local prestress and dynamic stress evolution (Li et al., 2023). The segmented, more complex fault geometry is more effective in dynamically arresting earthquake rupture (Segall & Pollard, 1980; Wesnousky, 2006). However, large earthquakes are still dynamically possible with rupture capable of jumping across several fault stepovers before eventually terminating."*

We think the fault geometric complexity is not irrelevant since it has an influence on the final moment magnitude, as we see in our dynamic rupture simulations with varying hypocenter locations. Also, the resulting difference in the predicted tsunami heights is significant. Please see the following excerpts:

L270ff:
*"Overall, the tsunami scenarios initiated by dynamic rupture scenarios on the*

*complex fault geometry cause smaller tsunamis (Fig. 7, see bottom three rows). In contrast to the scenarios on the simpler fault geometry, now the respective tsunami characteristics are highly dependent on epicentral location."*

L405ff:
*"The three tsunami scenarios sourced by dynamic rupture simulations across the complex fault geometry cause significantly smaller tsunamis. This is due to lower and more segmented fault slip leading to less vertical seafloor displacements, which are spatially more restricted."*

**Vertical coseismic displacement:**

Your models predict near-surface, vertical coseismic displacements at the rupture end, which is what is also often observed strike-slip earthquakes (cf. Interferograms and geodetic slip models of Ridegcrest, Maduo, Muji, Sarez). It might be not the main scope of this earthquake, but I find this interesting enough to briefly be discussed: Why to the segments rotate? Is it consistent, does always the same side of a right-lateral fault exhibits up motion, or what does it depend on (nucleation, dip etc.)?

**R1.3:** Thank you for pointing this out. Your additional references (see minor comment RM1.31) have been very helpful, and we included them in the now updated part of the discussion (see below). We also point out that the rake rotation is caused by vertical stress changes induced by slip in the uppermost part of the fault, which has a lower confining stress (Spudich et al., 1998; Guatteri & Spudich, 1998; Kearse et al., 2019; Kearse & Kaneko, 2020).

L345-362 reads now:
*"For all scenarios, we observe pronounced dynamic rake rotation near the surface, which we consider a plausible dynamic mechanism for generating increased coseismic vertical offset. The dynamic deviations from pure right-lateral strike-slip faulting are on the order of ±20° and introduce significant shallow dip-slip motion. Thereby, vertical seafloor displacements in our simulations are enhanced, which are critical for tsunami generation. Shallow rake rotation has been inferred for surface-breaking earthquakes using geological slickenlines and simple dynamic rupture models (Spudich et al., 1998; Guatteri and Spudich, 1998; Kearse et al., 2019). Vertical stress changes at the rupture front cause this change in rake angle, which is more pronounced near the surface due to smaller confining stresses (Kearse and Kaneko, 2020). No rake rotation is expected directly atop the hypocenter, which is confirmed in Fig. 6. In Fig. 6, a gradual increase of rake rotation can be observed away from the hypocenter for both unilateral and bilateral rupture scenarios on the simpler fault geometry. More complex patterns of rake rotation result in the scenarios on the complex fault geometry, and we observe a dependence of the spatial distribution of rake rotation on the fault segment length and on hypocenter location. Changes in rake for the right-lateral strike-slip earthquake dynamic rupture scenarios cause mostly uplift in the compressional quadrants. Gaudreau et al. (2023) investigate the 1971 San Fernando thrust faulting earthquake using aerial stereo photographs and discuss a rotation of rake away from the prestress direction. He et al. (2022) use InSAR, GNSS, and optical data to study the 2019*

*Ridgecrest Sequence and report vertical cumulative coseismic surface displacement after the sequence, interpreted as an indication of prominent coseismic rake rotation. Other studies focusing on finite-fault models, allowing for rake variations in their inversions for slip, show rake rotation most prominent at patches, which are near the surface and have a large slip magnitude (Metzger et al., 2017; Feng et al., 2017; Xiao et al., 2022; Hong et al., 2022)."*

**Figures**

Revise the number of figures to improve the ratio between visual presentation and text description. For example, Figure 10-12 and B3-B4 use a lot of space, but are hardly discussed. Maybe there is a better way to present (parts of) them and make your point? Figure 13 uses a full page, only to state that coupled models provide more details.

For a better recognition, please use the same symbols in all figures, e.g. mark the nucleation point in all relevant figures similarly. To better follow your argumentation, please refer more often to the figure under discussion, maybe also mark features under discussion with (more) arrows and labels.

**R1.4:** Thank you very much for your suggestions on how the figures and their representation could be improved. We followed our recommendation and moved previous Figs. 11 and 12 into the appendix (now Figs. B3, B4, see our responses RF1.7 and F8). We also included the annotations of the different wave types in the subfigure (d) of the new Figs. 10, B3, B4 (see RF1.7 and F8). Additionally, we changed the marker of the hypocenter. Throughout the manuscript, we now consistently use a star, which has the color yellow as long as it does not interfere with the color scheme of the figure itself (please see our responses RF1.2, RF1.3, F3, F4, F5, and F8).

We moved the former Fig. 13 to the appendix as well. Instead, we created a "pars pro toto" comparison as suggested (see RM1.42), which we show in the new Fig. 11 (please see our response F9).

**Adjective-chains**

Some sentences contain extremely long chains of adjectives, making the sentence difficult to read, particularly, when not an expert in the domain. Some verbs, "of"s, or commas might help to improve the readability. Examples are "both one-way linked 3D earthquake dynamic rupture and shallow-water equations tsunami simulations" (L70), "time-dependent one- way linked and 3D fully-coupled earthquake-tsunami modeling" (L6), "recently developed physics- based dynamic rupture models" (L73), L79/80, L94/95, "the dynamic rupture initial conditions" (L100), "complex off-shore fault system structure" (L109), "showcased complex fault geometry scenarios" (146), "community dynamic rupture benchmarks" (169).

**R1.5:** Thank you for pointing out our long adjective chains. We carefully revised the manuscript and made the following changes:

L5ff:
*"Here, we investigate physics-based scenarios combining simulations of 3D earthquake dynamic rupture and seismic wave propagation with tsunami generation and propagation. We present time-dependent modeling of one-way linked and 3D fully-coupled earthquakes and tsunamis for the ~100 km long Húsavík-Flatey Fault Zone in North Iceland."*

L72-78:
*"In this study, we investigate the tsunami potential of the HFFZ using two techniques to couple earthquake and tsunami models.. First, we apply a one-way linked approach, that links the time-dependent seafloor deformation from 3D earthquake dynamic rupture with a subsequent tsunami simulation based on solving the shallow-water equations  (Ulrich et al., 2019b; Madden et al., 2020; Wirp et al., 2021; Ulrich et al., 2022; van Zelst et al., 2022). Second, we show 3D fully-coupled earthquake-tsunami models, which simulate seismic (i.e., elastic), ocean gravity (i.e., tsunami) and compressional ocean acoustic waves simultaneously and self-consistently (Lotto and Dunham, 2015; Krenz et al., 2021; Abrahams et al., 2023). We extend six recent dynamic rupture scenarios (Fig. 2) from a suite of physics-based dynamic rupture models (Li et al., 2023)."*

L101ff:
*"Here, the initial conditions of the dynamic rupture models, including fault geometries, pre-stress, and fault strength, are constrained by seismic, geodetic, and bathymetry observations as briefly summarized in the following sections."*

L110f:
*"provide detailed insight on the complexity of the structure of the off-shore fault system."*

L148:
*"in comparison to their chosen scenarios using the complex fault geometry"*

L171f:
*"has been verified in community benchmarks for dynamic rupture earthquake simulations"*

L205f:
*"All initial conditions of the dynamic rupture models are kept the same as in the respective linked scenarios."*

**Minor comments**

Our responses are given in **green**, with changes to the manuscript in *italic*. We include the comments of **referee #1 in black**.

| Referee #1 | Revision |
|---|---|
| L35: northern | RM1.1: Northern → northern |
| L37ff (and also abstract): While introducing the tectonic setting of the TFZ and HFF, make sure that geographic features mentioned in the text also appear in figure labels to make sure the readership can follow your descriptions, for example, by referring already to Figure 1. I would also simplify the descriptions, for example, Nordurland eystra (L48/237) sounds like national description of the region and I would not use it, Kolbeinsey Ridge (KR and NVZ are never used anymore), Olafsfjördur should be shown in the map. Please revise the text regarding this. For example, the inset in Figure 1a could be increased and the three lineaments could be schematically marked. | RM1.2: We have added Olafsfjördur to the map (Fig. 1b). Furthermore, we updated the main text referring earlier to Fig. 1 and included KR and NVZ in Fig. 1a. The sentence *"The region of Norðurland eystra has experienced several large earthquakes in the past"* has been updated to *"North Iceland has experienced several large earthquakes in the past"*.
Please also see our response in F1. |
| L42/3: plate | RM1.3: Capitalization changed. Reads now:
*"Earthquake faulting in the TFZ is driven by eastward spreading of the Eurasian plate with an average velocity of ~18 mm/yr relative to the North American plate (Stefansson et al., 2008; Demets et al., 2010)."* |
| L45: This sentence sounds somewhat strange. | RM1.4: Lines 37-47 have been rephrased. Reads now:
*"Here, we focus on the ~100 km long Húsavík-Flatey Fault Zone (HFFZ, Fig. 1), the largest strike-slip fault in Iceland, which is part of the Tjörnes Fracture Zone (TFZ). The TFZ is a complex transcurrent fault system composed of three main lineaments. It links the Kolbeinsey Ridge (KR) as part of the Mid-Atlantic Ridge offshore North of Iceland (Eyjafjarðaráll Rift Zone) to its manifestation on land in the Northern Volcanic Zone (NVZ), which is characterized by volcanic systems and extensional faulting (Sæmundsson, 1974; Einarsson, 1991; Geirsson et al., 2006; Einarsson, 2008; Stefansson et al., 2008; Einarsson & Brandsdóttir, 2021).* |

| | |
|---|---|
| | *Earthquake faulting in the TFZ is driven by eastward spreading of the Eurasian plate with an average velocity of ~18 mm/yr relative to the North American plate (Stefansson et al., 2008; Demets et al., 2010). The HFFZ strikes from offshore to onshore and is characterized by right-lateral (dextral) strike-slip faulting, a faulting mechanism which appears frequently subparallel to the adjacent active rift zones of Iceland (Karson et al., 2018). It poses the largest threat to coastline communities such as the town of Húsavík, which is located atop the Húsavík-Flatey Fault Zone at the eastern side of Skjálfandi Bay."* |
| L52: Please quantify the term "last glacial maximum" | RM1.5: Changed. L53 reads now:
*"during the last ~12,000 years"* |
| L59: unit missing (years?). Introduce the abbreviation GNSS and GPS. The term GNSS is more general as it covers all positioning systems. | RM1.6: Changed. L59-62 reads now:
*"De Pascale (2022) calculate a recurrence interval of 32±24 years for a magnitude 6 event. Recent velocities obtained from Global Navigation Satellite System (GNSS) measurements – using more than 100 continuous and campaign-style GNSS stations in total – are close to zero near the fault, indicating that the HFFZ may be fully locked (Barreto et al., 2022)."* |
| L82ff: I would not abbreviate the world "Section" | RM1.7: We followed the Solid Earth submission guidelines (https://www.solid-earth.net/submission.html#manuscriptcomposition, 5th point).
"The abbreviation "Sect." should be used when it appears in running text and should be followed by a number unless it comes at the beginning of a sentence." |
| L104: I suggest to delete "of active submarine fault systems" and end the sentence with "a submarine earthquake rupture". | RM1.8: Changed. Reads now:
*"The fault geometry plays an important role in the potential for tsunami generation caused by submarine earthquake rupture."* |
| L125: please define Shmax in text form not only in line 127 but here. | RM1.9: Updated. Reads now:
*"Based on the three best quality criteria from the world stress map project (Zoback et al., 1989; Zoback, 1992; Sperner et al., 2003; Heidbach et al., 2007, 2010), they choose the maximum horizontal stress, SHmax (cf. Table 1), to set up a homogeneous regional stress field (Ziegler et al., 2016). This is consistent with previous estimates of SHmax from Angelier* |

| | |
|---|---|
| | *et al., 2004 and agrees with the local transtensional deformation pattern (Garcia and Dhont, 2004)."* |
| L155: No brackets needed around Abril et al. | RM1.10: Deleted parentheses. |
| Table 1: Is there a unit missing for Shmax? I would write "150/155" as entry with the asterisk. | RM1.11: We added degrees for the SHmax unit and followed your recommendation for the entry. |
| L165-167: Repetition of above. | RM1.12: We shortened this sentence but want to partially leave it in the text as a transition and for better clarification. Reads now:
*"We use the scientific open-source software package SeisSol (https://github.com/SeisSol/SeisSol, http://www.seissol.org) to simulate six earthquake dynamic rupture scenarios on the HFFZ on two fault system geometries (Sect. 2.1)."* |
| L171: replace "&" with "and" | RM1.13: Changed. |
| L173: Please explain "sam(oa)2-flash" already here, not in the consecutive sentence, e.g., by moving "dynamically adaptive, parallel software" one line up. What is the meaning of "parallel software"? That it is able to do computing in parallel? | RM1.14: Lines 175-178 have been updated. We also added "computing" for a better understanding. Reads now:
*"The one-way linked workflow uses the time-dependent seafloor displacement output from SeisSol to initialize sea surface perturbations within sam(oa)$_2$-flash (https://gitlab.lrz.de/samoa/samoa), a dynamically adaptive software for parallel computing (Meister et al., 2016). It solves the non-linear hydrostatic shallow water equations and has been linked to SeisSol in previous work (Ulrich et al., 2019; Madden et al., 2020; Wirp et al., 2021)."* |
| L177: Appendix B is rather short and would maybe add two additional text lines and one equation to the main text. It could be skipped at all (L175-178 already explain the key facts and that its contribution is small) or embedded in the text directly. | RM1.15: We followed your suggestion and decided to skip the first part of "Appendix B: Tsunami" regarding the Tanioka filter. It is not part of our key results and as you have noticed, we mention it already in the main text. Now, Appendix B contains solely the five supplementary figures supporting the tsunami simulation results. |
| L182: delete "so-called" | RM1.16: Deleted. |
| L188: delete "medium" | RM1.17: First appearance of "medium" deleted. |
| L197: Rephrase and start with "The on-fault | RM1.18: Rephrased. Reads now: |

| | |
|---|---|
| resolution of 200 m is gradually…" | RM1.19: *"The on-fault resolution of 200 m is gradually coarsened away from the HFFZ to a maximum size of 5 km at the edges of the elastic medium."* |
| L210: "A comparison of …" could be deleted, and the figure reference can be moved to the beginning of L212. | RM1.20: Changed. Reads now:
*"All earthquake dynamic ruptures on the simpler fault geometry break over the entire main fault length and generate larger maximum slip (Fig. A2)."* |
| L211/Table 2/main comment: 7.9 m slip sounds like too much compared to the estimated slip deficit, if the fault should accommodate 7 mm/yr or so, no, so maybe 1 m of slip at most? Can you explain the reason for these high slip values of all models? | RM1.21: Please see our answer to your main comment in our response R1.1. |
| L215: Can you explain a bit more? Is it relative to the max. slip observed further down? Is it calculated by the ratio between the area left and right of the curve plotted in Figure A3? | RM1.22: We have added another sentence to better explain the term 'shallow slip deficit (SSD)' within the text. L217-222 reads now:
*"Furthermore, the three earthquake dynamic rupture simulations on the simple fault geometry cause significant shallow fault slip resulting in a negligible shallow slip deficit (SSD, Fig. A3). The SSD ratio is defined as the ratio of near-surface slip to slip at seismogenic depths (e.g., Fialko et al., 2005; Marchandon et al., 2021). A higher percentage of SSD indicates that fault slip occurring at depth is larger compared to slip in the uppermost part of the fault. This is the case for all three scenarios on the complex fault geometry".* |
| L216/217: Is there a correlation between the spatial pattern observed in off-fault strain (Figure A1) and the SSD, or depth of max. slip? I notice that off-fault strain occurs at constant particular distance to the fault surface trace: | RM1.23: Investigating the correlation between shallow slip deficit (SSD) and off-fault plastic strain was not one of our key research questions. However, this might be a very interesting point to further investigate. It seems as if with larger near-surface slip we get higher strain values, however, we think the width of the strain accumulation and the flower structure might be more controlled by fault bends itself (i.e., restraining and releasing bends) (Ma, 2022), which can be seen in Fig. 12. |
| L217: "(Fig. 3b)" could you zoom into Fig. 3b somewhat to show more details of the flower structure? | RM1.24: We now show a zoom into the flower structure in Fig. 3b. Please see our response F2. |

| | |
|---|---|
| L220: delete "are non-negligible" and refer already in this sentence to Figure 5. | RM1.25: Deleted. Reads now:
*"The coseismic earthquake displacements reach up to ~1 m of seafloor uplift and up to ~0.8 m of subsidence (Fig. 5, Table 2) for ruptures on the simpler fault geometry with nucleations in the East and Middle of the HFFZ."* |
| L224: refer to Figure 1 here. | RM1.26: Inserted. Reads now:
*"This has a significant impact on seismic hazard assessment, in particular for the town of Húsavík, which is located directly above the HFFZ (Fig. 1)."* |
| L224: Delete "However," | RM1.27: Deleted. |
| L227: there is a unneeded bracket left | RM1.28: Deleted. |
| L228/229: It would help to have the geographic markers under discussion also provided in Figure 5. | RM1.29: We inserted a North arrow as well as a scale (25 km). Please check our response F3. |
| L231/232: Given the complexity of the rest of the paper, this explanation of the rake seems to be unnecessary for the target audience. You could start the next sentence instead with "Earthquakes on a vertically dipping, right-lateral fault system, such as the HFFZ, predominantly exhibit slip rake angles of 180 deg. However, …" Also, the sketch in Figure 5 should be updated to show a vertical fault. | RM1.30: Thank you for the feedback. Reads now:
*"Earthquakes on a vertically dipping, right-lateral fault system, such as the HFFZ, predominantly exhibit rake angles of 180°. However, we observe dynamic rake rotation (±20°) near the surface during the rupture (Fig. 6)."* |
| L232-235/L329-332/main comment: This agrees very well with InSAR offset observations (unfortunately rarely shown, for example, Ridgecrest, He et al., 2022, Figure 2k, https://doi.org/10.1029/2021JB022779), or slip models that do not constrain the rake angle (e.g., Sarez earthquake, Metzger et al., 2017, Figure S8, https://doi.org/10.1002/2017TC004581, Muji earthquake, Feng et al., 2017, Figure 6, https://doi.org/10.1785/0220170019, or Maduo earthquake, Xiao et al., 2022, https://doi.org/10.1007/s12583-022-1637-x, Hong et al., 2022, Figure 6, https://doi.org/10.1785/0120210250) | RM1.31: We want to thank you for pointing this out. It is of great value for our discussion. We address this in depth in our answer R1.3. |
| L236: Add reference to Figure 1 | RM1.32: Added at the end of the sentence. |
| L239: Delete "3D" | RM1.33: Deleted. |
| L241: I do not understand the benefit of the | RM1.34: We use the wording "synthetic tide |

| | |
|---|---|
| auxiliary wording of "synthetic tide gauge stations placed near". Could it not be deleted (also later) and speak of "predicted sea surface height anomaly at the coastal towns of" instead? | gauge stations" to emphasize that no real tide gauges exist at the locations we investigate. In general, we try to avoid the word "predicted", as it might be misunderstood by the general public. |
| L282: What is SuperMUC-NG? | RM1.35 We restructured the sentence and include the information that SuperMUC-NG is a high-performance computer.
Reads now:
*"A single fully-coupled simulation of joint dynamic rupture and tsunami generation (for 3 min of simulated time) requires ~4 h computational time with 40 nodes (1920 cores), that is, a total of 7680 CPUh, on the Munich supercomputer SuperMUC-NG (https://doku.lrz.de/supermuc-ng-10745965.html)."* |
| L285ff: Technically, the order of mentioning a), b), c), d) should be alphabetically, so exchange the labels b and c. | RM1.36: We changed this accordingly. Please see our answers RF1.7 and F8. |
| L294: Revise this sentence, there is a verb missing, I think. | RM1.37: Revised. Reads now:
*"First, we see the propagation of the tsunami at a speed of ~35 m/s towards the open ocean."* |
| L295: "cross-section to in Figs." | RM1.38: Reads now:
*"The tsunami waves in all three scenarios travel 5.6 km in 160 s (cross-section 2 in Figs. 10, B3, B4)."* |
| L3010304: It would be instructive to mark these waveforms directly in the subfigures d) and refer to them here (instead of in the appendix). | RM1.39: We changed this accordingly. Please see our answers R1.4 and F8. |
| L294-304: Given the length of the text discussion Figures 10-12 and also the information content of the figures (to the untrained eye, at least, subfigures b) and d) look very similar in all cases), you might consider moving two of them to the supporting material? | RM1.40: We changed this accordingly. Please see our answers R1.4 and F8. |
| L300: "Figs. 10b ,11b, 12b, cross-section 1)" | RM1.41: We revised this and followed your suggestion to flip b) and c) in Figs. 10, B3 and B4. Please see F8.
Reads now:
*"(Figs. 10c, B3c, B4c, cross-section 1)"* |
| L305-314: You show a panel of 12 figures to only conclude that the waveforms contain more short-period signal. If there is not | RM1.42: We changed this accordingly. Please see our detailed comment F9. |

| | |
|---|---|
| more to gain from this plot, I would only show two subplots as pars-pro-toto. | |
| L316: "Submarine" instead of "Submerged"? | RM1.43: Changed to *"submarine"*. |
| L317: Rearrange the Sulawesi sentence: "Linked and fully-coupled … modeling of the 2018 … earthquake imply that … coseismic…" | RM1.44: Rearranged. Please see our response R2.1. |
| L320: No need to repeat what you did, just start with "Our simulations of six rupture scenarios show that the Husavik-Flatey…" | RM1.45: Changed. Reads now: *"Our simulations of six earthquake dynamic rupture scenarios show that the Húsavík-Flatey Fault Zone can host tsunamigenic earthquakes."* |
| L332: Replace "In difference" with "Unlike" o "Oppositely to | RM1.46: We changed the sentence. Please see RM1.45. |
| L334: Refer to Figure A1 here. | RM1.47: Inserted reference. Reads now: *"In contrast to the suggested important contribution of off-fault deformation to strike-slip tsunami generation in Palu Bay (Ma, 2022), the effect of off-fault plasticity is likely small in our simulations (Fig. A1)."* |
| L340: Is there a "so" missing between the wave speed and "we can calculate"? | RM1.48: We added a comma and the word *"so"* for a better readability. Reads now: *"Our average water depth H can be approximated as ˜200 m, the source width σr as given by the length of the HFFZ (˜100 km), the source duration σt of 30 s constrained by the rupture duration (cf. moment rates Fig. 4), the gravitational acceleration g = 10 m s−2, and acoustic wave speed c0 = 1500 m s−1 , so we can calculate the three non-dimensional numbers posed by Abrahams et al., 2023 as specified in Table 3."* |
| L340-350: I do not see the benefit of Table 3 as everything is already mentioned here in the text. | RM1.49: We understand your concern. However, some readers might only skim through the text and find it easier to take a look at a concise table for an overview. Therefore, we decided to not take the table out. |
| L348: Refer to the respective subfigures in Figure 7 here | RM1.50: Reads now: *"This fact explains the similarity in the tsunami propagation and shape of the tsunami wavefronts for the simple fault geometry scenarios (i.e., Fig. 7 after 2 min (second column) and 10 min (third column)), which all break the entire main* |

| | |
|---|---|
| | *fault length and lead to similar fault slip distributions."* |
| L353ff: I do not understand: Do you imply that after submarine earthquakes at shallow depths we should be more concerned about acoustic waves rather than tsunami waves? | RM1.51: We did not intend to imply this. We meant to say that the vertical velocity amplitudes of the acoustic waves are larger than the amplitudes from the tsunami. A better understanding of how these different types of waves superimpose can help future efforts of tsunami early warning if proper instrumentation is available.

L387f reads now:
*"However, our fully-coupled simulations include acoustic wave generation with high vertical velocity amplitudes, larger than the tsunami signals (Figs. 10, 11, 12)."*

L447-451 reads now:
*"A better understanding of such acoustic wave signals may improve tsunami early warning since these can be detected earlier in the far-field, e.g., at ocean bottom pressure sensors, in comparison to the tsunami recorded at conventional DART buoys (Yamamoto, 1982; Cecioni et al., 2014; Mei and Kadri, 2017; Gomez and Kadri, 2021). However, in the near-field, ocean acoustic waves can superimpose onto tsunami signals, impeding early warning efforts."* |
| L394: See my comment for L233-235 resp. L329-332: Yes, the 8 m of slip seems to be large, but vertical offsets also exists at the end of strike-slip ruptures. | RM1.52: Thank you again for pointing this out. Please see our answers to the comment at the beginning of the document. |
| L411: Delete "on-average" or replace with "relatively" | RM1.53: Replaced with *"relatively"*. |
| L415: I cannot guess what kind a "sizable tsunami" is, maybe you find another term? | RM1.54: We deleted the word *"sizable"*, as it is not needed at this point. Lines 423ff provide the necessary information about the tsunami height. |
| L416: Be more specific what you mean by "two distinct fault system", e.g., by saying "a simpler (X segments) and a more complex (55 segments) fault geometry". | RM1.55: Reads now:
*"We investigate two distinct fault system geometries – a simpler fault geometry with three fault segments and a highly complex fault system composed of 55 fault segments – to represent the 100 km long Húsavík-Flatey Fault Zone striking from onshore to offshore."* |

| L419: To me 8 m of fault slip seems too unrealistic to point out in the conclusion. It would implicate 800 years of complete slip deficit, no? | RM1.56: Please have a look at our response R1.1. |
|---|---|

**Figures/Caption comments**

Our responses are given in **green**, with changes to the manuscript in *italic*. We include the comments of **referee #1 in black**.

**Figure 1**

"I do not understand the meaning of the percentage of SSD, please explain better, also the meaning of "shallow" in km.

RF1.1: We assume you are referring to Fig. A3, which shows the normalized cumulative fault slip with depth for all six earthquake dynamic ruptures. We address this in our answer F12, which includes the updated figure caption as well as the figure itself.

Additionally, we updated the main text, which we address in RM1.22.

**Figure 5**

The font is rather small, can you increase it? Also increase the size of the hypocentral star. Use the same symbol (size and color) in Figure 5, 6, 7 10-12.

RF1.2: We address this in our answers F3, F4, F5 and F8.

**Figure 7**

The figure readability would improve if towns are directly labeled/numbered in the subfigure, instead of describing their location in the caption. Add the hypocenter, similar to Figure 5.

RF1.3: We address this in our answer F5.

**Figure 6**

The sketch of the rakes should be adapted to show a vertical fault (incl. slip direction), not a dipping one.

RF1.4: Please have a look at our answer F4.

**Figure 8/B1**

Remove the overarching titles

RF1.5: We address this in our answer F6.

**Figure 9**

Increase the font label size. Could you rework the choice of colors (or background color of the legend), such that the individual colors are readable and distinguishable?

RF1.6: We address this in our answer F7.

**Figure 10-12**

It would be instructive to non-experts to label the different wave fronts in subfigure d). Flip label b) and c). It took me until reading the conclusions that acoustic waves are "booms" (right?) and not real waves, because in your Figures the color-map is "sea surface vertical velocity [ssvv]"? Please clarify.

RF1.7: We address the improvements of the figures in our answer F8.

To clarify your question about acoustic waves, we made the following changes:

L72-77 (also see R1.5) reads now:
*"In this study, we investigate the tsunami potential of the HFFZ using two techniques to couple earthquake and tsunami models.. First, we apply a one-way linked approach, that links the time-dependent seafloor deformation from 3D earthquake dynamic rupture with a subsequent tsunami simulation based on solving the shallow-water equations (Ulrich et al., 2019b; Madden et al., 2020; Wirp et al., 2021; Ulrich et al., 2022; van Zelst et al., 2022). Second, we show 3D fully-coupled earthquake-tsunami models, which simulate seismic (i.e., elastic), ocean gravity (i.e., tsunami), and compressional ocean acoustic waves simultaneously and self-consistently (Lotto and Dunham, 2015; Krenz et al., 2021; Abrahams et al., 2023)."*

L195-198 reads now:
*"Within the water layer, we set the rigidity equal to zero ($\mu = 0$) and we prescribe an ocean acoustic wave speed of ~1500 m s−1. Acoustic waves (compressive sound waves) are modeled everywhere within the water layer while tsunamis waves, treated as surface gravity waves, are modeled driven by gravity forces acting as restoring forces trying to restore equilibrium at the sea surface (Krenz et al., 2021)."*

**Figure B2**

The color map is flipped between a) and b).

RF1.8: Thank you for pointing this out. We address this in our answer F13.

**Supporting Text**

Our responses are given in **green**, with changes to the manuscript in *italic*. We include the comments of **referee #1 in black**.

When I tested the rupture wavefield videos, some of the started only at the second half of the the video time. Please check.

We apologize for the inconvenience. We additionally uploaded all videos in *.avi* format, which hopefully resolves the issue (https://zenodo.org/record/8360914).

**RC2 - Referee #2**

Our responses are given in **green**, with changes to the manuscript in *italic*. We include the comments of **referee #2 in blue**.

**Summary**

The manuscript submitted by Dr. Kutschera and the colleagues investigates the potential tsunami hazard due to strike-slip earthquakes in the Húsavík-Flatey Fault Zone in North Iceland. Their proposal of tsunami scenarios based on two different tsunami modeling approaches, one-way linked tsunami model and the 3D fully-coupled (earthquake–tsunami) model, are informative to understand how similar and different these results are. The authors also try the simulation with differently assumed simple and complex fault geometries, which yielded different results. They suggest the importance of consideration of detailed fault geometry for tsunami scenarios and that strong fast-arriving acoustic waves may be used to notice the tsunami potential in advance.

The manuscript is very well-prepared and deliver useful information in an appropriate way. I here list up numbers of comments, but these are just for more concise and fair descriptions. I hope these will be helpful for the authors to improve the manuscript. After this minor revision, I can recommend the publication of this article.

We appreciate the thoughtful review of our manuscript. We address all comments and suggestions in the following.

**Moderate comments**

1. When the authors discuss on the 2018 Palu tsunami, only the strike-slip earthquake itself is attributed to the huge tsunami, but there is still possibility of the landslide origins. It would be fairer to mention previous studies which proposed the landslide tsunami origins somewhere in introduction or discussion (for example, around Lines 33–33, 318–320, or 396–397).

   **R2.1:** We agree with your comment and now mention the possibility of a landslide induced tsunami originating from the 2018 Palu, Sulawesi, Earthquake.

   L325-329 reads now:
   *"Linked and fully-coupled earthquake dynamic rupture and tsunami modeling for the 2018 Mw 7.5 Sulawesi earthquake in Indonesia suggest that coseismic-induced seafloor displacements critically contributed to the generation of an unexpected and devastating local tsunami in Palu Bay (e.g., Ulrich et al., 2019b; Krenz et al., 2021; Ma, 2022). Widespread liquefaction-induced coastal and submarine landslides likely also played an important role (e.g., Carvajal et al., 2019; Gusman et al., 2019; Pakoksung et al., 2019; Sassa & Takagawa, 2019; Sepúlveda et al., 2020)."*

   Likewise, for the HFFZ, we updated L429f:
   *"We do not consider combined earthquake and landslide-induced tsunami scenarios for the HFFZ, which can additionally increase the local tsunami height (Ruiz-Angulo et al., 2019)."*

2. When the authors compare the two tsunami modeling approach, a one-way linked or 3D fully-coupled ones, it sounds that the one-way linked approach is somewhat underrated. For example, the dispersion property can be included into the one-way linked model by using an appropriate dispersive tsunami model, but in the present manuscript, it sounds that the dispersion can incorporated only by the 3D fully-coupled model (I know that this is not the authors' intention). The non-dispersion is due to the shallow-water wave model. Please keep fairness to compare the one-way linked and the 3D fully- coupled model.

   **R2.2:** We apologize, this was not our intention. The one-way linked tsunami modeling approach is a well-established framework. Fully-coupled tsunami modeling requires higher computational cost for large-scale fully-coupled runs (Krenz et al., 2021) and is not accounting for inundation. The reviewer is right that tsunami dispersion can also be included for nonlinear shallow water equations using for example Boussinesq-type approximations (e.g., Madsen et al., 1991; Baba et al., 2015). As detailed in Abrahams et al., 2023 fully-coupled simulations can be complementary to the one-way linked approach, e.g., to accurately capture larger amplitude ocean acoustic and seismic waves that are superimposed on tsunami waves in the source region during the initial tsunami generation phase.

   We updated L451-456. Reads now:

*"In addition to the seismo-acoustic wave excitation in the fully-coupled simulations, we observe dispersion of tsunami propagation velocity (Tsai et al., 2013). Glimsdal et al. (2013) show that enhanced dispersion effects are expected for earthquakes with magnitude 8 and less as opposed to less dispersive tsunamis caused by the largest earthquakes. We do not account for dispersion in the one-way linked tsunami simulations that are based on solving the non-linear hydrostatic shallow water equations. However, Boussinesq-type tsunami models can account for dispersion (e.g., Madsen et al., 1991; Baba et al., 2015).*

Additionally, we changed "the" to "our" in L474f:
*"3D fully-coupled scenarios include source dynamics, seismic, acoustic, and tsunami waves and result in complexities not present in our one-way linked simulations."*

Another update can be found in our response RM2.14.

3. I think both two tsunami models you use here do not consider tsunami runup or inundation to coasts. Is my understanding correct? The classical one-way linked model can easily incorporate the runup/inundation, as done many models such as Baba et al. (2015). How about the 3D fully-coupled model? I guess it requires sophisticated modeling of a moving land/ocean boundary. Please discuss on this somewhere (maybe in Discussion), because the runup/inundation can be also important aspects for the tsunami hazard assessment. (I understand that this would be out of your scope, and comparison at tide gauges without inundation is useful enough. I comment on this because it would be very informative for tsunami researchers.)

**R2.3:** This is correct and a good point for future work regarding tsunami hazard assessment. We include this in our discussion about our current limitations.

L441-446 reads now:
*"We exclude inundation in the one-way linked approach to enable a more meaningful comparison with the fully-coupled method. Our fully-coupled simulations are* computationally demanding and *do not allow to model inundation (e.g., Krenz et al., 2021). We show that the fault geometry in our six one-way linked scenarios can influence the subsequent tsunami generation. Here, future studies may explore potential variations in fault dip, which may further enhance the vertical seafloor displacement during the earthquake rupture. Changes in earthquake source parameters are known to affect the maximum tsunami height (e.g., Burbidge et al., 2015) and resulting inundation (e.g., Gibbons et al., 2022)."*

**Minor comments**

Our responses are given in **green**, with changes to the manuscript in *italic*. We include the comments of **referee #2 in blue**.

| Referee #1 and #2 | Revision |
|---|---|
| Lines 37–47: The tectonic setting is better to described as a figure. it is difficult to understand the tectonics only from the text. I recommend that the authors put a figure with a broader area including the main tectonic setting, with notations of TFZ, NVZ, and the fault motion direction with average velocity of the plate boundary. These will be helpful for readers who are not familiar with the region. | RM2.1: We updated Fig. 1 accordingly. Please see our response F1. |
| Line 50: *The strongest historically recorded M 7 event in 1755 caused extensive damage and may have generated a series of oceanic waves (i.e., a tsunami) that hit the coastline (Stefansson et al., 2008; Þorgeirsson, 2011; Ruiz-Angulo et al., 2019)*
 Would you please explain this part more clearly? Did the referenced studies show any evidence of tsunamis (such as historically documented record or tsunami sediment), or just speculated based on the large fault offset found on seafloor? | RM2.2: We clarified this in L49-52. Reads now:
 *"The largest M7 event in 1755 caused extensive damage and historic reports indicate that a tsunami hit the coastline and overturned boats (Stefansson et al., 2008; Þorgeirsson, 2011; Ruiz-Angulo et al., 2019). Likewise, such reports include records that the events in 1872 caused rapid sea level changes resulting in a series of waves, i.e., a tsunami-like behavior."* |
| Line 59: *calculate a recurrence interval of 32±24*
 What are the unit? Year? | RM2.3: We forgot the unit. Please see our response RM1.6. |
| Lines 74–81:
 The detailed information, on such as SeisSol, GMM, or parameter differences, does not fit here in Introduction. I recommend that the authors introduce what they will show just more briefly here and move the details to in Sect 2 or 3.1. It will guide readers more smoothly to the details. | RM2.4:

 We removed the part about SeisSol as suggested (please also see RM2.9), but decided to keep the citation of Li et al. (2023), as we are building up on their dynamic rupture models.

 You can find the update in our response R1.5. |
| Line 90: *We model one-way linked (cf. Sect. 2.4) and fully-coupled (cf. Sect. 2.5) tsunami scenarios (Abrahams et al., 2023)*
 "One-way linked and fully-coupled tsunami scenarios" sounds strange to me. If we say "tsunami scenarios", it would be possible tsunami cases by different source models, | RM2.5: We changed this part accordingly. Reads now:
 *"We present one-way linked (cf. Sect. 2.4) and fully-coupled (cf. Sect. 2.5) tsunami models"* |

| | |
|---|---|
| as you say in the following paragraph like "We use six earthquake scenarios." Please consider replacing the "tsunami scenarios" by "tsunami models", "tsunami approaches" or "tsunami computation methods". | |
| Line 131–132: *It also agrees with assuming a 90∘dipping fault system.* How about "It also agrees with our assumption of a 90° dipping fault system"? | RM2.6: Changed. L133f reads now: *"It also agrees with our assumption of a 90° dipping fault system."* |
| Line 153: *6 to 10 km is the inferred locking depth for the HFFZ (Metzger and Jónsson, 2014).* Based on what is the locking depth estimated? Please clalify. | RM2.7: Clarified. L154ff reads now: *"6 to 10 km is the inferred locking depth for the HFFZ (Metzger & Jónsson, 2014), which was estimated by the combined analysis of InSAR time-series and GNSS data and a back-slip model, which describes the interseismic locking by applying continuous slip at depth in reversed slip direction (e.g., Savage, 1983; Metzger et al., 2013; Wang et al., 2015)."* |
| Figures 3: Three panels in the figure are shown too small. Please consider showing the panels vertically in a column., or "a) on the top row, and b) and c) on the bottom row. | RM2.8: We updated Fig. 3 based on your suggestions, please see F2. |
| Line 165: Here you explain SeisSol. You may remove the description of SeisSol in Introduction and explain the details here. | RM2.9: We removed the description of SeisSol from the Introduction. Please see our responses RM2.9 and R1.5. |
| Line 233: *However, we observe dynamic rake rotation (20°) near the surface during the rupture (Fig. 6).* It is interesting that there happens a rake orientation. Could you please explain how this happens? | RM2.10: We significantly extended our discussion of the rake rotation. Please see our response R1.3. |
| Figure 6, A1, A2: The panels are very small. Please enlarge the figures, by showing them vertically, three panels of a) in 1st column, and those of b) in the 2nd column. The same problem happens in Fig. A1 and A2. Please consider revising them, as well. | RM2.11: We followed your suggestion and updated Figs. 5, 6, A1 and A2. Please view our updates in F3, F4, F10 and F11. |
| Figure 7: Why are the Simple cases at t=120 and 600s show very broad ssha ( faint green and blue)? This looks weird, | RM2.12:  Thank you for pointing this out. The faint blue colors in Fig. 7 (see F5) are due to the static displacement, as visible in Fig. 5 (see F3). This is also present for our |

| | |
|---|---|
| and is very contrasted to the complex cases. Please check they are modelled correctly and, if true, explain what they represent. | complex cases, however, at a much smaller scale compared to the scenarios on the simpler fault geometry.

We include this in the figure caption, please see our update in F5. |
| Figure B2:
The color bar is flipped. Please revise it. | RM2.13: You are absolutely correct, we fixed this. Please see RF1.8 and F13. |
| Lines 305–310 (Related to my moderate comment 2):
As you know, the non-dispersive character is not because of a one-way linked tsunami model, but just because of the "linear long-wave model" employed. It would be fair to note that the dispersion can be simulated in a one-way approach as well. | RM2.14: Please see our detailed response R2.2.

Additionally, we updated L315ff. Reads now:
*"The one-way linked tsunami waveforms (dashed black line) for both cross-sections in the first and third row are rather smooth. We note that accounting for the dispersive effects, which are not considered in the depth-integrated (hydrostatic) shallow water equations we are solving for, could potentially lead to less smooth synthetics."* |
| Lines 333–335:
*In difference to a proposed dominance of off-fault deformation in strike-slip tsunami generation in Palu Bay (Ma, 2022), the effect of offfault plasticity is likely small in our simulations. We find that off-fault deformation contributes only about ~3 % of the total seismic moment.*
Could you please mention possible reasons of the difference? Is it because of fault geometry, rupture model, or other parameters? | RM2.15:
We updated L364-368. Reads now:
*"In contrast to the suggested important contribution of off-fault deformation to strike-slip tsunami generation in Palu Bay (Ma, 2022), the effect of off-fault plasticity is likely small in our simulations (Fig. A1). We find that off-fault deformation contributes only about ~3 % of the total seismic moment. Accounting for the potential existence of shallow, weak sediments, which are more prone to off-fault plastic deformation, may increase local uplift (Seno & Hirata, 2007; Ma & Nie, 2019; Wilson & Ma, 2021; Ulrich et al., 2022)."* |

**Updated Figures**

**Figure 1, 2**

F1: Based on a comment by the editorial support team, we used the Coblis - Color Blindness Simulator (https://www.color-blindness.com/coblis-color-blindness-simulator) and revised the color schemes of Figs. 1, 2.

In panel a), we marked the Kolbeinsey Ridge (KR) as part of the Mid-Atlantic Ridge offshore North of Iceland (Eyjafjarðaráll Rift Zone), which connects the Tjörnes Fracture Zone (TFZ) to its manifestation on land in the Northern Volcanic Zone (NVZ).

In panel b), we included a schematic tectonic overview of the TFZ. It shows the three lineaments, the indication of right-lateral strike-slip faulting for the HFFZ, the overall direction of plate motion as well as the location and names of the volcanic centers at the Grimsey Oblique Rift for completeness.

[Figure]

*"Figure 1. Overview of the Tjörnes Fracture Zone (TFZ), which connects the Kolbeinsey Ridge (KR) as part of the Mid-Atlantic Ridge offshore North of Iceland (Eyjafjarðaráll Rift Zone) to its manifestation on land in the Northern Volcanic Zone (NVZ). Yellow circles represent relocated seismicity from 1993 to 2019 (Abril et al., 2018, 2019). a) The here used "simple" fault geometry of the Húsavík-Flatey Fault Zone (HFFZ), which has three segments, is shown as red lines (Li et al., 2023) . Historic large earthquakes with M ≥ 6 are indicated as blue stars (Ambraseys and Sigbjörnsson, 2000; Stefansson et al., 2008; Þorgeirsson, 2011; Jónsson, 2019). b) The used "complex" fault geometry of the HFFZ (Li et al., 2023), which includes 55 fault segments (shown as red lines) together with major towns in the region of Norðurland eystra. The inset at the bottom right shows a schematic tectonic overview of the TFZ with the average plate motion (Stefansson et al., 2008; Demets et al., 2010) and the overall regional tectonic setting in North Iceland, including the location and names of the volcanic centers at the Grímsey Oblique Rift."*

[Figure]

*"Figure 2. Overview over the six 3D dynamic rupture earthquake scenarios based on Li et al. (2023). Arrows indicate the three varied epicenter locations. Each dynamic rupture scenario is nucleated at a hypocentral depth of 7 km. We show the on-fault measured moment magnitude and the equivalent centroid moment tensor solutions (constructed after Ulrich et al., 2022) representing overall strike-slip faulting mechanisms of the dynamic rupture scenarios. a) shows the three dynamic rupture models on the simple fault system geometry with varying epicentral locations and b) are the three scenarios on the complex fault system geometry."*

**Figure 3**

F2: We now show a zoom into the flower structure in Fig. 3b and restructured the layout based on comment RM2.8. Additionally, we include the 3D water layer in panel d).

[Figure]

*"Figure 3. a) Snapshot at t = 10 s of the simulated seismic wavefield for the earthquake dynamic rupture nucleating in the East of the simple fault geometry. b) Accumulated off-fault plastic strain (η) at the end of simulation Simple-East forming a shallow flower structure. The zoom into the flower structure at the bottom right additionally shows the incorporated static mesh refinement near the fault. c) Mesh of the fully-coupled earthquake-tsunami simulation with the distinction between the elastic medium (Earth) and the acoustic medium (Ocean). d) Vertically exaggerated 3D water layer of the fully-coupled mesh with a maximal length (E-W) of 86 km, maximal width (N-S) of 52 km and maximal depth (Z) of 430 m."*

**Figure 5**

F3: We increased the size of the hypocentral star in Figs. 5, 6, 7, A1, A2 and Figs. 10, B3, B4 (formerly Figs. 10-12) and consistently use a star to mark the hypocenter  (based on comment RF1.2). We try to stick to a yellow filling. However, due to partially overlapping color maps impeding the visibility and interfering with the Color Blindness Simulator, we use different colored stars in Figs. 5 and A1.

The top-left panel now includes a length scale and a North arrow for better spatial recognition. Based on comment RM2.11, we changed the figure layout from 2x3 to 3x2.

a)  b)

[Figure]

*"Figure 5. Uplift and subsidence from the surface displacements of earthquake dynamic rupture simulations after accounting for local bathymetry using the Tanioka filter (Tanioka and Satake, 1996) on a) the simple fault geometry and b) the complex fault geometry. Black stars mark the epicenter locations."*

**Figure 6**

F4: Based on comment RM2.11, we changed the figure layout from 2x3 to 3x2. Additionally, we improved the schematic illustration of the rake rotation (see comment RF1.4) and now use strike-slip beachballs (with varying degree of rake) instead of two blocks.

[Figure]

*"Figure 6. Dynamic rake rotation in the dynamic rupture simulations on a) the simple fault geometry and b) the complex fault geometry. Yellow stars mark the hypocenter locations. A rake of 180 degrees indicates pure right-lateral strike-slip faulting."*

**Figure 7**

F5: We added the epicenters as yellow stars on the map (see RF1.2) and numbered the towns in the subfigure (top left), while providing their names in the updated figure caption (see RF1.3).

We also updated the figure caption, addressing RM2.12.

[Figure]

*"Figure 7. Sea surface height anomaly (ssha) of all six one-way linked earthquake-tsunami scenarios at 10 s (first column), 2 min (second column), and 10 min (third column) simulation*

*times. The yellow star in the first column marks the epicenter of each scenario. The red points in the top-left panel indicates the position of synthetic tide gauges near the coastal towns (1) Siglufjörður, (2) Ólafsfjörður, (3) Dalvík, (4) Akureyri, (5) Húsavík (west to east on the mainland) and (6) Grímsey Island. The faint distant blue and green coloring is due to the static vertical displacement as shown in Fig. 5, which is more pronounced for scenarios on the simpler fault geometry."*

**Figure 8, B1**

F6: We removed the overarching titles in Figs. 8, B1 (see comment RF1.5) and updated the order of the subfigures to be better in agreement with Fig. 7. Please view the updated figure below:

[Figure]

*"Figure 8. Sea surface height anomaly (ssha [cm]) vs simulation time (40 min) for the three one-way-linked scenarios sourced by dynamic rupture simulations on the simpler fault geometry recorded at six synthetic tide gauge stations close to the towns Siglufjörður, Ólafsfjörður, Dalvík, Akureyri, Húsavík (west to east on the mainland) and Grímsey Island."*

[Figure]

*"Figure B1. Sea surface height anomaly (ssha [cm]) vs simulation time (40 min) for the three one-way-linked scenarios sourced by dynamic rupture simulations on the complex fault geometry recorded at six synthetic tide gauge stations close to the towns Siglufjörður, Ólafsfjörður, Dalvík, Akureyri, Húsavík (west to east on the mainland) and Grímsey Island."*

**Figure 9**

F7: We increased the font label size as requested in comment RF1.6 and now use the color white as the background color of the legend, which improves the recognizability of the different scenarios.

[Figure]

*"Figure 9. Maximum sea surface height anomaly (ssha [cm]) recorded throughout the simulation time of 40 minutes at synthetic tide gauge stations nearby local communities in North Iceland for the one-way linked scenarios based on the simpler fault geometry (a) and the complex fault geometry (b). At each tide gauge, we show the maximum ssha of all three respective scenarios, with bar colors indicating the epicentral location of the scenario causing maximum ssha at a given location."*

**Figure 10, B3, B4**

F8: We flipped labels b) and c) as suggested (see RM1.36 and RF1.7) and included the annotations directly in subfigure d) (see R1.4). Also, we plotted a white box in the upper two panels of subfigure d) to show the zoom of the lower two panels of d). We further moved Figs. 11 and 12 into the appendix (now B3 and B4) and, therefore, deleted previous figures (formerly B3, B4, B5) from the appendix because they seem redundant otherwise.

[revised manuscript text omitted]

F12: We updated the figure caption based on comment RF1.1 to provide more information about the meaning of the percentage of SSD.

[Figure]

*"Figure A3. Normalized cumulative slip with depth for all six earthquake dynamic ruptures. The amount of shallow slip deficit (SSD) is indicated at the top left for each model on the respective fault geometry. 0 % of SSD represents no near-surface reduction of fault slip, while a higher percentage indicates that coseismic slip in the uppermost crust is less than slip occurring at average depths of the seismogenic layer (i.e., 4 – 6 km, Fialko et al. (2005)). The scenarios on the simpler fault geometry exhibit no SSD with large shallow fault slip, while SSD up to 36.3 % can be observed for dynamic rupture model Complex-West."*

**Figure B2**

F13: Thank you for bringing to our attention that initially the color scale was flipped (see RF1.8 and RM2.13). We have inverted the color map in b) such that it now matches a) as well as the previous figures showing the sea surface height anomaly (ssha).

[revised manuscript text omitted]

---

## Author Response (AR2)

**Rebuttal letter Minor Revision Solid Earth egusphere-2023-1262**

Linked and fully-coupled 3D earthquake dynamic rupture and tsunami modeling for the Húsavík-Flatey Fault Zone in North Iceland

We thank the handling editor for his positive feedback. We include the additional comment of the **Editor in dark red** and address his comments in **green**, with changes to the manuscript in *italic*. We hope the revised manuscript will be well received.

Sincerely,
Fabian Kutschera, on behalf of all co-authors

**Editor**

Thank you for your rigorous response to reviewer comments, which have in general thoroughly addressed all points raised by the reviewers. The clarity of your reply is commendable and appreciated.

Please consider the following additional suggestions from the handling editor before final consideration for publication in Solid Earth. Minor further additions re. the main concerns raised by reviewer 1 for coseismic slip distances:

- Add a couple of sentences/short section discussing observed coseismic slips from other strike-slip faults globally to add confidence that modelled coseismic slips fall within the range naturally observed for the modelled earthquake magnitudes and observed length of the Húsavík-Flatey Fault Zone.

We followed this suggestion and added to L450-457:
*"Our simulated maximum fault slip occurring within the shallow offshore part of the HFFZ, i.e., the part which is relevant for the subsequent tsunami generation, is comparable to geological observations from earthquakes rupturing along faults with similar length to the HFFZ. Examples of strike-slip ruptures as summarized by Wesnousky (2008) comprise Neo Dani, Japan (length 80 km, max. slip 7.9 m, Matsuda (1974)), Luzon, Philippines (length 112 km, max. slip 6.2 m, Yomogida & Nakata (1994)), and Landers, California (length 77 km, max. slip 6.7 m, Sieh et al. (1993)). A recent example includes the second event of the devastating Kahramanmaras earthquake sequence (length 150 km) resulting in up to 8 m fault slip near the surface (Jia et al., 2023). While we use a LSW friction law, considering a rate-and-state dependent friction law would allow to include shallow velocity-strengthening behavior which may decrease slip in the shallowest parts of the fault (e.g., Kaneko et al., 2008)."*

- Add further description and justification (i.e. references) for the major modelled physical parameters that specifically can impact coseismic fault slip distances (e.g. Cplast or others) to section 2 (model setup).

*We added to L162-165:*
*"Our assumed lower limit of the locking depth (~10 km, see Li et al. (2023)) is shallower in comparison to the locking depths of most continental strike-slip faults (Vernant, 2015). Consequently, this can result in an overshoot of fault length scaling relations (Mai and Beroza, 2000; Shaw, 2013). However, it is in agreement with oceanic transform faults (Abercrombie and Ekström, 2001), where the warmer temperature of the lithosphere at the Mid-Atlantic Ridge controls slip at depth."*

L171-174 reads now:
*"Similarly to the model parameterization in Li et al. (2023), the bulk friction is set to resemble the fault static coefficient of friction (μs = 0.6) and assumed to be constant in the elastic solid*

*medium. Bulk cohesion is depth-dependent and varies in dependence of the velocity model. It is calculated as a function of our 3D rigidity model as Cplast = $10^{-4}\mu$ [Pa], which is following the low cohesion model of Roten et al. (2014)."*

And L447ff reads now:
*"The dynamic rupture scenarios are consistent with the **average fault slip and effective rupture area** scaling relations of Mai and Beroza (2000), which have recently been validated for the Southern Iceland Seismic Zone (SISZ, Bayat et al. (2022))."*

- Briefly explain your rationale for using higher R0 values in this study compared to Li et al. (2023), i.e. is this simply to investigate the maximum potential a Tsunami may have (?), and clarify the sentence in lines 156-158 - do you mean "in comparison to values used in Li et al. (2023)"?

Thank you for this question, we now clarify in L147-152:
*"We here choose a slightly higher R0 = 0.9 for all three dynamic rupture simulations on the complex fault geometry in comparison to the scenarios shown by Li et al. (2023) using the complex fault geometry (R0 = 0.85). R0 itself is difficult to directly obtain from observations and we constrain it using a few dynamic rupture trial-and-error simulations (Ulrich et al., 2019a). The change in R0 results in a ~20 % average increase in vertical displacements. Based on the large parameter space explored in the suite of HFFZ dynamic rupture simulations of Li et al. (2023), our chosen models represent end-member earthquake-tsunami scenarios in terms of large uplift."*

We also clarified in L80ff:
*"The **simple fault geometry rupture models of Li et al. (2023)** coincide with historically and physically plausible earthquake magnitudes, stress drop, rupture speed, and slip distributions, and produce ground motions that have been verified against empirical Ground Motion Models (GMMs) calibrated for Iceland (Kowsari et al., 2020)."*

Upon satisfactory addressing of these suggestions, I would be pleased to consider the manuscript for publication in this Solid Earth Special Issue.

Best,

Jordan J.J. Phethean

We hope that we satisfactorily addressed these suggestions. We would like to thank you again for your comments and handling our manuscript.

Sincerely,
Fabian Kutschera, on behalf of all co-authors

**Rebuttal letter references**

Abercrombie, R. E., & Ekström, G. (2001). Earthquake slip on oceanic transform faults. *Nature*, *410*(6824), Article 6824. https://doi.org/10.1038/35065064

Bayat, F., Kowsari, M., & Halldorsson, B. (2022). A new 3-D finite-fault model of the Southwest Iceland bookshelf transform zone. *Geophysical Journal International*, *231*(3), 1618–1633. https://doi.org/10.1093/gji/ggac272

Jia, Z., Jin, Z., Marchandon, M., Ulrich, T., Gabriel, A.-A., Fan, W., Shearer, P., Zou, X., Rekoske, J., Bulut, F., Garagon, A., & Fialko, Y. (2023). The complex dynamics of the 2023 Kahramanmaraş, Turkey, Mw 7.8-7.7 earthquake doublet. *Science*. https://doi.org/10.1126/SCIENCE.ADI0685

Kowsari, M., Sonnemann, T., Halldorsson, B., Hrafnkelsson, B., Snæbjörnsson, J., & Jónsson, S. (2020). Bayesian inference of empirical ground motion models to pseudo-spectral accelerations of south Iceland seismic zone earthquakes based on informative priors. *Soil Dynamics and Earthquake Engineering*, *132*, 106075. https://doi.org/10.1016/J.SOILDYN.2020.106075

Li, B., Gabriel, A.-A., Ulrich, T., Abril, C., & Halldórsson, B. (2023). Physics-based dynamic rupture models, fault interaction and ground motion simulations for the segmented Húsavík-Flatey Fault Zone, Northern Iceland. *Journal of Geophysical Research: Solid Earth*.

Mai, P. M., & Beroza, G. C. (2000). Source Scaling Properties from Finite-Fault-Rupture Models. *Bulletin of the Seismological Society of America*, *90*(3), 604–615. https://doi.org/10.1785/0119990126

Matsuda, T. (1974). Surface faults associated with Nobi (Mino-Owari) earthquake of 1891, Japan. *Spec. Rep. Earthq. Res. Inst.*, *13*, 85–126.

Roten, D., Olsen, K. B., Day, S. M., Cui, Y., & Fäh, D. (2014). Expected seismic shaking in Los Angeles reduced by San Andreas fault zone plasticity. *Geophysical Research Letters*,

*41*(8), 2769–2777. https://doi.org/10.1002/2014GL059411

Shaw, B. E. (2013). Earthquake Surface Slip‐Length Data is Fit by Constant Stress Drop and is Useful for Seismic Hazard Analysis. *Bulletin of the Seismological Society of America*, *103*(2A), 876–893. https://doi.org/10.1785/0120110258

Sieh, K., Jones, L., Hauksson, E., Hudnut, K., Eberhart-Phillips, D., Heaton, T., Hough, S., Hutton, K., Kanamori, H., Lilje, A., Lindvall, S., McGill, S. F., Mori, J., Rubin, C., Spotila, J. A., Stock, J., Thio, H. K., Treiman, J., Wernicke, B., & Zachariasen, J. (1993). Near-Field Investigations of the Landers Earthquake Sequence, April to July 1992. *Science*, *260*(5105), 171–176.

Ulrich, T., Gabriel, A. A., Ampuero, J. P., & Xu, W. (2019a). Dynamic viability of the 2016 Mw 7.8 Kaikōura earthquake cascade on weak crustal faults. *Nature Communications*, *10*(1). https://doi.org/10.1038/s41467-019-09125-w

Vernant, P. (2015). What can we learn from 20years of interseismic GPS measurements across strike-slip faults? *Tectonophysics*, *644–645*, 22–39. https://doi.org/10.1016/j.tecto.2015.01.013

Wesnousky, S. G. (2008). Displacement and Geometrical Characteristics of Earthquake Surface Ruptures: Issues and Implications for Seismic-Hazard Analysis and the Process of Earthquake Rupture. *Bulletin of the Seismological Society of America*, *98*(4), 1609–1632. https://doi.org/10.1785/0120070111

Yomogida, K., & Nakata, T. (1994). Large slip velocity of the surface rupture associated with the 1990 Luzon Earthquake. *Geophysical Research Letters*, *21*(17), 1799–1802. https://doi.org/10.1029/94GL00515